# TWEEDIE MOMENT PROJECTED DIFFUSIONS FOR INVERSE PROBLEMS

## ABSTRACT

Diffusion generative models unlock new possibilities for inverse problems as they allow for the incorporation of strong empirical priors into the process of scientific inference. Recently, diffusion models received significant attention for solving inverse problems by posterior sampling, but many challenges remain open due to the intractability of this sampling process. Prior work resorted to Gaussian approximations to conditional densities of the reverse process, leveraging Tweedie's formula to parameterise its mean, complemented with various heuristics. In this work, we leverage higher order information using Tweedie's formula and obtain a finer approximation with a principled covariance estimate. This novel approximation removes any time-dependent step-size hyperparameters required by earlier methods, and enables higher quality approximations of the posterior density which results in better samples. Specifically, we tackle noisy linear inverse problems and obtain a novel approximation to the gradient of the likelihood. We then plug this gradient estimate into various diffusion models and show that this method is optimal for a Gaussian data distribution. We illustrate the empirical effectiveness of our approach for general linear inverse problems on toy synthetic examples as well as image restoration using pretrained diffusion models as the prior. We show that our method improves the sample quality by providing statistically principled approximations to diffusion posterior sampling problem.

## 1    INTRODUCTION

Solving inverse problems is one of the most crucial goals in scientific inference (Tarantola, 2005; Stuart, 2010). Diffusion models (Song et al., 2020; Ho et al., 2020) recently emerged as a strong alternative to classical Bayesian approaches to solve inverse problems to build samplers for complicated and implicit conditional distributions. With their flexibility, these models replace handcrafted priors on the latent signal with pretrained and strong empirical priors. For example, given a latent signal $\mathbf{x}_0$ (say a face image), we can *train* a diffusion model to sample from the prior $p(\mathbf{x}_0)$. The idea for solving inverse problems is to leverage extraordinary modelling power of diffusion models to learn samplers for priors and couple this with a given likelihood $p(\mathbf{y}|\mathbf{x}_0)$ for a given data $\mathbf{y}$ to sample from posterior densities for inverse problems. However, designing a diffusion model for the posterior comes with its own challenges due to intractability. Despite its challenges, this approach has been recently taking off with lots of activity in the field, e.g. for compressed sensing (Bora et al., 2017; Kadkhodaie & Simoncelli, 2021), projecting score-based stochastic differential equations (SDEs) (Song et al., 2021b), gradient-based approaches (Daras et al., 2022; Chung et al., 2022b), magnetic reasonance imaging (MRI) by approximating annealed Langevin dynamics with approximate scores (Jalal et al., 2021), image restoration (Kawar et al., 2022), score-based models as priors but with a normalising flow approach (Feng et al., 2023), variational approaches (Mardani et al., 2023; Feng & Bouman, 2023). Most relevant ideas to us, which we will review in detail in Section 5, use Tweedie's formula (Efron, 2011; Laumont et al., 2022) to approximate the smoothed likelihood, e.g. Diffusion posterior sampling (DPS) (Chung et al., 2022a; 2023) and pseudo-guided diffusion models (ΠGDM) (Song et al., 2023). Similar approaches are also exploited using singular-value decomposition (SVD) based approaches (Kawar et al., 2021). In this work, we develop an approach which builds a tighter approximation to optimal formulae for approximating the scores of the posterior diffusion model.

This paper is devoted developing novel methods to solving inverse problems, given a latent (target) signal $\mathbf{x}_0 \in \mathbb{R}^{d_x}$, noisy observed data $\mathbf{y} \in \mathbb{R}^{d_y}$, a known linear observation map $\mathbf{H}$, and a *pretrained* diffusion prior. The main tool we use is Tweedie's formula to obtain both the mean and the covariance for approximating diffused likelihoods, to be used for building the final posterior score approximation. This is as opposed to previous works which only utilised first moment approximations using Tweedie's formula (Chung et al., 2022a; Song et al., 2023). We show that utilising covariance approximation, with moment projections, provides a principled scheme with improved performance, which we term *Tweedie Moment Projected Diffusions* (TMPD).

To demonstrate our method briefly, Figure 1 demonstrates a sampling scenario of a *Gaussian random field* (GRF) whose mean and variance entries are plotted under "Analytic" column[1]. We demonstrate the approximations under this setting provided by our method (TMPD) and its diagonal (cheaper) version (DTMPD), compared with ΠGDM (Song et al., 2023), and DPS (Chung et al., 2022a). The figure demonstrates the optimality our method: Our first and second moment approximations become exact in this case. This results in a drastic performance improvement stemming from the statistical optimality of our method for near-Gaussian settings and also unlocks a possible line for theoretical research for understanding similar diffusion models for inverse problems.

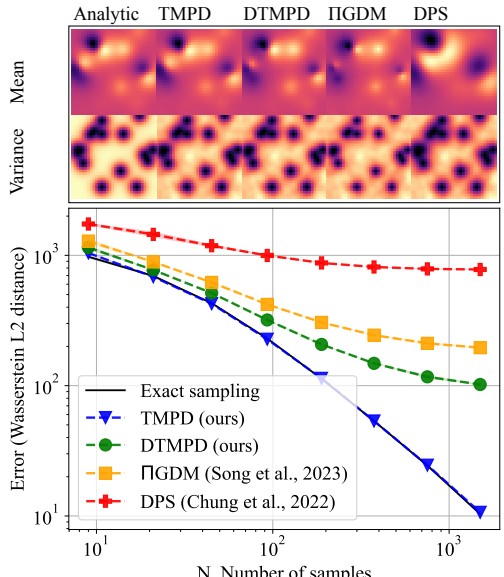

Figure 1: Error to target posterior for a *Gaussian random field*. (Top row) visualisation of the empirical mean and variance of the 1500 samples that were used to compute this error against the analytical moments. (Bottom) Wasserstein distances of different methods w.r.t. sample size. For details, see Appendix E.2.

In what follows, we will first introduce the technical background in Section 2 and then describe TMPD in detail in Section 3. We will then provide some theoretical results about our method in Section 4 and provide a discussion to closely related work in literature in Section 5. Finally, Section 6 will present experiments on Gaussian mixtures, image inpainting and super-resolution, demonstrating quantitative and qualitative improvements provided by TMPD.

## 2 TECHNICAL BACKGROUND

There are two main diffusion modelling paradigms, namely, score-based models (SGMs) (Song et al., 2020) and denoising diffusion probabilistic models (DDPM) (Ho et al., 2020). In both, the goal is to sample from a target distribution $p_0 := p_{\text{data}}$. To that end, an artificial path $p_t$ is introduced, with the property that $p_t$ will approach $\mathcal{N}(0, I)$ for large $t$, i.e., $p_t \to \mathcal{N}(0, I_d)$ as $t \to \infty$. Then, one learns to reverse this process, so that one can sample from a standard normal distribution and transform these samples to samples from $p_{\text{data}}$. In the SDE paradigm, the interpolation parameter $t$ will take continuous values in $t \in [0, T]$, whereas in the DDPM setting, $t$ is discrete. However, the DDPM setting can be seen as discretization of the SDE (Song et al., 2020).

In SGMs, a stochastic differential equation (SDE) is used to noise the data. A very common choice is to use the time-rescaled Ornstein-Uhlenbeck process:

$$\mathrm{d}\mathbf{x}_t = -\frac{1}{2}\beta(t)\mathbf{x}_t\mathrm{d}t + \sqrt{\beta(t)}\mathrm{d}\mathbf{w}_t, \quad \mathbf{x}_0 \sim p_0 = p_{\text{data}}. \tag{1}$$

---

[1]We only plot the variances for visualisation while the GRF has a full covariance.

The corresponding reverse SDE is then given by

$$d\mathbf{z}_t = \frac{1}{2}\beta(T-t)\mathbf{z}_t dt + \beta(T-t)\nabla_{\mathbf{z}_t}\log p_{T-t}(\mathbf{z}_t)dt + \sqrt{\beta(T-t)}d\bar{\mathbf{w}}_t, \quad \mathbf{z}_0 \sim p_T. \quad (2)$$

The transition kernels for the time-changed SDE can be derived from the transition kernels of a normal OU process $p_t(\mathbf{x}_t|\mathbf{x}_0) = \mathcal{N}(\mathbf{x}_t; \sqrt{\alpha_t}\mathbf{x}_0, v_t\mathbf{I}_{d_x})$ where $\alpha_t := \exp\left(-\int_0^t \beta(s)ds\right)$ and $v_t := 1 - \alpha_t$. A parameterisation that performs well in practice is $\beta(t) = \beta_{\min} + t(\beta_{\max} - \beta_{\min})$. In the diffusion modelling literature, the OU process is also sometimes called variance preserving SDE. However, this is not the only SDE that is suitable for the forward process. See Appendix C for details on a time rescaled Brownian motion (Variance Exploding (Song et al., 2020) SDE).

There are two usual approximations to solve equation 2. First, we do not know $p_T$, since it is a noised version of the distribution $p_{\text{data}}$. However for $T$ large enough, we can approximate $p_T \approx p_{\text{ref}} = \mathcal{N}(0, I_d)$. We also do not have $\nabla \log p_{T-t}$ which we need for the drift in equation 2. This can be circumvented by approximating the drift using score-matching techniques (Hyvärinen, 2005; Ho et al., 2020). These methods construct an estimate of the score function by solving the score matching problem in the form of $\mathbf{s}_\theta(\mathbf{x}_t, t) \approx \nabla_{\mathbf{x}_t}\log p_t(\mathbf{x}_t)$. This score can also be used in the setting of DDPM (Ho et al., 2020).

### 2.1 CONDITIONAL SAMPLING FOR THE LINEAR INVERSE PROBLEM

In the preceding section we introduced diffusion models as a method to sample from a target distribution $p_{\text{data}}$. We now suppose that we have access to measurements, or observations $\mathbf{y} \in \mathbb{R}^{d_y}$ of $\mathbf{x}_0 \in \mathbb{R}^{d_x}$:

$$\mathbf{y} = \mathbf{H}\mathbf{x}_0 + \mathbf{u}, \quad \mathbf{u} \sim \mathcal{N}(0, \sigma_y^2\mathbf{I}_{d_y}). \quad (3)$$

We would then be interested in sampling from the conditional distribution of $\mathbf{x}_0$ given $\mathbf{y}$, i.e., $p_{\text{data}}(\ \cdot\ |\mathbf{y})$. To that end, we have to modify the reverse SDE. Optimally, we would like to sample from the reverse SDE to the forward SDE started in $p_{\text{data}}(\ \cdot\ |\mathbf{y})$, instead of the one started in $p_{\text{data}}$.

Optimally, we would want to replace the score $\nabla_{\mathbf{z}_t}\log p_{T-t}(\mathbf{z}_t)$ in equation 2 with the posterior score $\nabla_{\mathbf{z}_t}\log p_{T-t|\mathbf{y}}(\mathbf{z}_t|\mathbf{y})$. Written in terms of the forward process, this coincides with

$$\nabla_{\mathbf{x}_t}\log p_{t|\mathbf{y}}(\mathbf{x}_t|\mathbf{y}) = \nabla_{\mathbf{x}_t}\log p_t(\mathbf{x}_t) + \nabla_{\mathbf{x}_t}\log p_{\mathbf{y}|t}(\mathbf{y}|\mathbf{x}_t). \quad (4)$$

The term $p_{\mathbf{y}|t}(\mathbf{y}|\mathbf{x}_t)$ is given by the integral

$$p_{\mathbf{y}|t}(\mathbf{y}|\mathbf{x}_t) = \int p_{\mathbf{y}|0}(\mathbf{y}|\mathbf{x}_0)p_{0|t}(\mathbf{x}_0|\mathbf{x}_t)d\mathbf{x}_0, \quad (5)$$

which involves a marginalization over $\mathbf{x}_0$. The term $p_{0|t}(\mathbf{x}_0|\mathbf{x}_t)$ however is only defined implicitly through running the diffusion model, making the above integral difficult to evaluate. One way around this is to train a neural network to directly approximate $\nabla \log p_{t|\mathbf{y}}(\mathbf{x}_t|\mathbf{y})$ (Batzolis et al., 2021). Alternatively, if one already has access to an approximation of $\nabla_{\mathbf{x}_t}\log p_t(\mathbf{x}_t)$, one can train an auxiliary network to approximate the term $\nabla \log p_{\mathbf{y}|t}(\mathbf{y}|\mathbf{x}_t)$ in equation 4, (Song et al., 2020; Dhariwal & Nichol, 2021). However, these methods can be time and training-data intensive, as one needs to retrain networks for each conditional task as well have access to paired training data from the joint distribution $(\mathbf{x}_0, \mathbf{y})$. Alternatively, one could indeed try to do a Monte-Carlo approximation of the score corresponding to equation 5. But this needs evaluating the probability flow ODE together with its derivative (Song et al., 2020, Section D.2) for each sample, which is prohibitive, also suffers from high variance (Mohamed et al., 2020, Section 3).

## 3 TWEEDIE MOMENT PROJECTED DIFFUSIONS

In this section, we first introduce *Tweedie moment projections* in Section 3.1 below. Our method relies on the approximation $p_{0|t}(\mathbf{x}_0|\mathbf{x}_t) \approx \mathcal{N}\left(\mathbf{x}_0; \mathbf{m}_{0|t}(\mathbf{x}_t), \mathbf{C}_{0|t}(\mathbf{x}_t)\right)$ to make the sampling process tractable. In that case, since the conditional distribution of $\mathbf{y}$ given $\mathbf{x}_0$ is also Gaussian, we can *compute the integral in equation 5 analytically* — $p_{\mathbf{y}|t}(\mathbf{y}|\mathbf{x}_t)$ will just be another Gaussian in that case, its mean and covariance being determined through $\mathbf{m}_{0|t}$, $\mathbf{C}_{0|t}$, $\mathbf{H}$ and $\sigma_y$. In particular, we can then use this Gaussian to approximate $\nabla \log p_{\mathbf{y}|t}(\mathbf{y}|\mathbf{x}_t)$, since the score of a Gaussian is available in closed form.

### 3.1 Tweedie moment projections

Instead of just approximating the variance of $p_{0|t}(\mathbf{x}_0|\mathbf{x}_t)$ heuristically, we approximate it by projecting on to the closest Gaussian distribution using Tweedie's formula for the second moment. Our approximation at this stage consists of two main steps: (1) Find the mean and covariance of $p_{0|t}(\mathbf{x}_0|\mathbf{x}_t)$ using Tweedie's formula, and (2) approximate this density with a Gaussian using the mean and covariance of $p_{0|t}(\mathbf{x}_0|\mathbf{x}_t)$ (moment projection). Due to this approximation, we will refer to resulting methods as *Tweedie Moment Projected Diffusions* (TMPD). We will first introduce Tweedie's formula for the mean and covariance and then describe the moment projection.

**Proposition 1** (Tweedie's formula). *Let $\mathbf{m}_{0|t}$ and $\mathbf{C}_{0|t}$ be the mean and the covariance of $p_{0|t}(\mathbf{x}_0|\mathbf{x}_t)$, respectively. Then given the marginal density $p_t(\mathbf{x}_t)$, the mean is given as*

$$\mathbf{m}_{0|t} = \mathbb{E}[\mathbf{x}_0|\mathbf{x}_t] = \frac{1}{\sqrt{\alpha_t}}(\mathbf{x}_t + v_t \nabla_{\mathbf{x}_t} \log p_t(\mathbf{x}_t)).$$

*Then the covariance $\mathbf{C}_{0|t}$ is given by*

$$\mathbf{C}_{0|t} = \mathbb{E}\left[ (\mathbf{x}_0 - \mathbf{m}_{0|t})(\mathbf{x}_0 - \mathbf{m}_{0|t})^\top \mid \mathbf{x}_t \right] = \frac{v_t}{\alpha_t}(\mathbf{I}_{d_x} + v_t \nabla^2 \log p_t(\mathbf{x}_t)) = \frac{v_t}{\sqrt{\alpha_t}} \nabla_{\mathbf{x}_t} \mathbf{m}_{0|t}. \quad (6)$$

The proof is an adaptation of Meng et al. (2021, Theorem 1), see Appendix A.1. While $\mathbf{m}_{0|t}$ and $\mathbf{C}_{0|t}$ gives us the moments of the density $p_{0|t}(\mathbf{x}_0|\mathbf{x}_t)$, we do not have the exact form of this density. At this stage, we employ *moment projection*, i.e., we choose the closest Gaussian in Kullback-Leibler (KL) divergence to a distribution with same moments as formalised next.

**Proposition 2** (Moment projection). *Let $p_{0|t}(\mathbf{x}_0|\mathbf{x}_t)$ be a distribution with mean $\mathbf{m}_{0|t}$ and covariance $\mathbf{C}_{0|t}$. Let $\hat{p}_{0|t}(\mathbf{x}_0|\mathbf{x}_t)$ be the the closest Gaussian in KL divergence to $p_{0|t}(\mathbf{x}_0|\mathbf{x}_t)$, i.e.,*

$$\hat{p}_{0|t}(\mathbf{x}_0|\mathbf{x}_t) = \arg \min_{q \in \mathcal{Q}} D_{\mathrm{KL}}(p_{0|t}(\mathbf{x}_0|\mathbf{x}_t)||q),$$

*where $\mathcal{Q}$ is the family of multivariate Gaussian distributions. Then*

$$\hat{p}_{0|t}(\mathbf{x}_0|\mathbf{x}_t) = \mathcal{N}(\mathbf{x}_0; \mathbf{m}_{0|t}, \mathbf{C}_{0|t}).$$

This is a well-known moment matching result, see, e.g., Bishop (2006, Section 10.7). Merging Propositions 1 and 2 leads to the following *Tweedie moment projection*:

$$p_{0|t}(\mathbf{x}_0|\mathbf{x}_t) \approx \mathcal{N}\left( \mathbf{x}_0; \mathbf{m}_{0|t}, \frac{v_t}{\sqrt{\alpha_t}} \nabla_{\mathbf{x}_t} \mathbf{m}_{0|t} \right), \quad (7)$$

where $\mathbf{m}_{0|t}$ is given in Proposition 1. In the next section, we demonstrate how to use this approximation to obtain approximate likelihoods.

### 3.2 Tweedie Moment Projected Likelihood Approximation

We next use the approximation in equation 7 to compute the following integral *analytically*

$$p_{\mathbf{y}|t}(\mathbf{y}|\mathbf{x}_t) = \int p_{\mathbf{y}|0}(\mathbf{y}|\mathbf{x}_0) p_{0|t}(\mathbf{x}_0|\mathbf{x}_t) \mathrm{d}\mathbf{x}_t = \mathcal{N}\left( \mathbf{y}; \mathbf{H}\mathbf{m}_{0|t}, \mathbf{H}\frac{v_t}{\sqrt{\alpha_t}} \nabla_{\mathbf{x}_t} \mathbf{m}_{0|t} \mathbf{H}^\top + \sigma_\mathbf{y}^2 \mathbf{I}_{d_y} \right) \quad (8)$$

which leads to the approximation

$$f^\mathbf{y}(\mathbf{x}_t) := \nabla_{\mathbf{x}_t} \mathbf{m}_{0|t} \mathbf{H}^\top (\mathbf{H}\frac{v_t}{\sqrt{\alpha_t}} \nabla_{\mathbf{x}_t} \mathbf{m}_{0|t} \mathbf{H}^\top + \sigma_\mathbf{y}^2 \mathbf{I}_{d_y})^{-1} (\mathbf{y} - \mathbf{H}\mathbf{m}_{0|t}) \approx \nabla_{\mathbf{x}_t} \log p_{\mathbf{y}|t}(\mathbf{y}|\mathbf{x}_t), \quad (9)$$

where $\nabla_{\mathbf{x}_t}$ only operates on $\mathbf{m}_{0|t}$ in the above equation. We note that this approximation is exact for the case $p_{\mathrm{data}}$ is a Gaussian. However, otherwise, there are several approximations behind equation 9. In particular, the approximation in equation 9 treats $\mathbf{C}_{0|t}$ *constant* w.r.t. $\mathbf{x}_t$ when computing the gradient (which is the case if $p_{\mathrm{data}}$ is Gaussian). For non-Gaussian $p_{\mathrm{data}}$, this results in computationally efficient sampler, as otherwise the resulting terms can be expensive to compute.

### 3.3 ALGORITHMS

Plugging the approximation in equation 9 into the reverse SDE in equation 2 together with the prior score as described in Section 2.1 results in a TMPD for conditional sampling to solve inverse problems. The SDE we will approximate numerically to sample from the conditional distribution is given by

$$d\mathbf{z}_t = \frac{1}{2}\beta(T-t)\mathbf{z}_t dt + \beta(T-t)(\nabla_{\mathbf{z}_t}\log p_{T-t}(\mathbf{z}_t) + f^{\mathbf{y}}_{T-t}(\mathbf{z}_t))dt + \sqrt{\beta(T-t)}d\bar{\mathbf{w}}_t \quad (10)$$

where $\mathbf{z}_0 \sim p_T$ and $f^{\mathbf{y}}_t(\mathbf{z}_t) \approx \nabla \log p_{\mathbf{y}|T-t}(\mathbf{y}|\mathbf{z}_t)$ is our approximation to the data likelihood, given by equation 9. We call this SDE the *TMPD SDE*.

We have two options to convert TMPD SDE into implementable methods: (1) score-based samplers which we abbreviate also as TMPD as they are Euler-Maruyama discretizations of the TMPD SDE akin to Song et al. (2020), (2) denoising diffusion models (TMPD-D). The denoising diffusion approach is derived from approximate reverse Markov chains and is the approach of DDPM and DPS methods (Ho et al., 2020; Chung et al., 2022a). We note that the Gaussian projection can be used in this discrete setting, assuming that the conditional density is available analytically as in Ho et al. (2020), and can be written as $p_{n|0}(\mathbf{x}_n|\mathbf{x}_0) = \mathcal{N}(\mathbf{x}_n; \sqrt{\alpha_n}\mathbf{x}_0, v_n\mathbf{I}_{d_x})$. The idea is to update the unconditional mean $\mathbf{m}_{0|n}(\mathbf{x}_n)$ of the density $p_{0|n}(\mathbf{x}_0|\mathbf{x}_n)$ with a Bayesian update: $p(\mathbf{x}_0|\mathbf{x}_n, \mathbf{y}) \propto p(\mathbf{y}|\mathbf{x}_n)p_{0|n}(\mathbf{x}_0|\mathbf{x}_n)$. Given a similar formulation as above, assuming we have a readily available approximation $p_{0|n}(\mathbf{x}_0|\mathbf{x}_n) \approx \mathcal{N}(\mathbf{x}_0; \mathbf{m}_{0|n}, \mathbf{C}_{0|n})$ and a likelihood similar to equation 8 where $t$ can be replaced by $n$, we can compute the moments of $p(\mathbf{x}_0|\mathbf{x}_n, \mathbf{y})$ analytically, which we denote $\mathbf{m}^{\mathbf{y}}_{0|n}$ and $\mathbf{C}^{\mathbf{y}}_{0|n}$. The Bayes update for Gaussians gives (Bishop, 2006)

$$\mathbf{m}^{\mathbf{y}}_{0|n} = \mathbf{m}_{0|n} + \mathbf{C}_{0|n}\mathbf{H}^\top(\mathbf{H}\mathbf{C}_{0|n}\mathbf{H}^\top + \sigma^2_y\mathbf{I}_{d_x})^{-1}(\mathbf{y} - \mathbf{H}\mathbf{m}_{0|n}), \quad (11)$$

Incorporating equation 11 for $n = N-1, \ldots, 0$ into the usual Ancestral sampling (Ho et al., 2020) steps leads to Algorithm 1, termed TMPD-Denoising (TPMD-D). The update in equation 11 can be used in any discrete sampler such as denoising diffusion implicit models (DDIM) (Song et al., 2021a).

### 3.4 COMPUTATIONALLY CHEAPER APPROXIMATION OF MOMENT PROJECTION

We show in our experiments promising results for TMPD that motivate the exploration of less computationally expensive approximations to the full Jacobian. In particular, we empirically study a computationally inexpensive method that is applicable to inpainting and super-resolution, below.

To make the computational cost of TMPD smaller, we can make an approximation of the Gaussian Projection that requires fewer Jacobian-vector products and does not require linear solves. One approximation that we found useful for sampling from high dimensional diffusion models, e.g., high resolution images, is denoted here as diagonal Gaussian Projection (DTMPD). Instead of the full second moment, DTMPD uses the diagonal of the second moment $\nabla_{\mathbf{x}_t}\mathbf{m}_{0|t} \approx \text{diag}(\nabla_{\mathbf{x}_t}\mathbf{m}_{0|t})$. Intuitively, this approximation will perform well empirically since it is a similar approximation to $\Pi$GDM that assumes dimensional independence of the distribution $p(\mathbf{x}_0|\mathbf{x}_t)$. A further approximation approximates the diagonal of the Jacobian by the row sum of the Jacobian which only requires one vector jacobian product and brings the memory and time complexity of DTMPD down to that of $\Pi$GDM. We discuss a justification of this approximation in Ap. E.1. We use this approximation in the image experiments and find that in practice it is only $(1.5 \pm 0.1)\times$ slower than $\Pi$GDM and DPS across all of our experiments (Sec. 6), with significantly better sample quality for noisy inverse problems.

## 4 THEORETICAL GUARANTEES

Because of the approximations, our method, as well as $\Pi$GDM Song et al. (2023) and DPS (Chung et al., 2022a) do not sample the exact posterior for general prior distributions. Therefore, one cannot hope for these methods to sample the true posterior and a priori it is not even clear how the sampled distribution relates to the true posterior. Without further justification, such methods should only be interpreted as *guidance methods*, where paths are guided to regions where a given observation $\mathbf{y}$ is more likely, not as posterior sampling methods.

We justify our approximation by showing that the TMPD-SDE in equation 10 is able to sample the *exact posterior* in the Gaussian case. One can see that this is in contrast to $\Pi$GDM and DPS in our numerical experiments or by explicitly evaluating their approximations on simple one-dimensional examples.

**Proposition 3** (Gaussian data distribution). *Assume that $p_{\text{data}}$ is Gaussian. Then, the posterior score expression using equation 9 is exact, i.e., if there are no errors in the initial condition and drift approximation $s_\theta(\mathbf{x}_t, t) = \nabla_{\mathbf{x}_t} \log p_t(\mathbf{x}_t)$, the TMPD will sample $p_{\text{data}}(\cdot|\mathbf{y})$ at its final time.*

The proof is given in Appendix B.1. However, most distributions will not be Gaussian. The following Theorem generalizes the above proposition to *non-Gaussian distributions*, as long as they have a density with respect to a Gaussian. We study how close our sampled measure will be to the true posterior distribution and give explicit bounds on the total variation distance in terms of the regularity properties of the density:

**Theorem 1** (General data distribution). *Assume that the data distribution $p_{\text{data}}$ can be written as*

$$p_{\text{data}}(\mathbf{x}_0) = \exp(\Phi(\mathbf{x}_0))\mathcal{N}(\mathbf{x}_0; \boldsymbol{\mu}_0, \boldsymbol{\Sigma}_0), \tag{12}$$

*for some $\boldsymbol{\mu}_0$ and $\boldsymbol{\Sigma}_0$. We furthermore assume that for some $M \geq 1$, it holds that $1/M \leq \exp(\Phi(\mathbf{x})) \leq M$ and $\|\nabla_{\mathbf{x}}\Phi(\mathbf{x})\| \leq L$ for all $\mathbf{x} \in \mathbb{R}^{d_x}$. Then*

$$\|p_{\text{data}}(\cdot|\mathbf{y}) - q_T(\cdot)\|_{\text{TV}} \leq C(1 + T^{1/2})\left((M^{5/2} - 1)(L^{1/2} + 1) + L^{1/2}\right),$$

*where $q_t$ denotes the law of the corresponding reverse-time process for equation 10 at time $t$ and the constant $C$ that only depends on $\mathbf{y}, \mathbf{H}, \sigma_y, \boldsymbol{\mu}_0$ and $\boldsymbol{\Sigma}_0$.*

See Appendix B.2 for a proof.

**Remark 1.** We note that this analysis can be extended to account for (i) discretization error, (ii) the error in the initial distribution of the reverse SDE, (iii) the score approximation error using the techniques from Chen et al. (2023). We think this is a fruitful direction for our work and we leave it for future work.

## 5 RELATED WORK

In this section, we review two closely related methods which we use as benchmarks and summarise the relationship between methods.

The first work is by Chung et al. (2022a), abbreviated DPS-D here[2] where $p_t(\mathbf{x}_0|\mathbf{x}_t)$ is approximated by a Dirac delta (point) distribution centred at $\mathbf{m}_{0|t}$ computed by Tweedie's formula, within the integral given in equation 5. The mean is chosen the same way as our method. Authors choose a Dirac centred at this mean, in our framework this corresponds to choosing a zero covariance, i.e.,

$$\mathbf{m}_{0|t}^{\text{DPS-D}} = \mathbf{m}_{0|t} \qquad \text{and} \qquad \mathbf{C}_{0|t}^{\text{DPS-D}} = 0. \tag{13}$$

In the work Song et al. (2023), abbreviated $\Pi$GDM-D here[3] the same estimator for the mean is chosen. However, the variance is set to a multiple of the identity, corresponding to choices

$$\mathbf{m}_{0|t}^{\Pi\text{G}} = \mathbf{m}_{0|t} \qquad \text{and} \qquad \mathbf{C}_{0|t}^{\Pi\text{G}} = r_t^2 \mathbf{I}_{d_x} \tag{14}$$

The choice of $r_t$ is such that it matches the variance of the reverse SDE if $p_{\text{data}}$ would be a standard normal distribution. Since they employ a different forward SDE to ours (a standard Brownian motion, also called variance exploding SDE), $r_t$ is set to be equal to $((v_t)^{-1} + 1)^{-1}$. In our case, with the OU/variance preserving SDE as a forward process, $r_t$ would be equal to $v_t$.

Another relevant very recent work by Finzi et al. (2023) arrives at the approximation matching ours in the context of modelling physical constraints. However, we focus on general linear inverse problems outside physical domain in this paper together with a novel theoretical result. Also we note the work of Stevens et al. (2023) who consider maximum-a-posteriori (MAP) approach to find the moments of $p_{0|t}$.

---

[2]Since the authors run DDPM-type sampler, we re-abbreviate DPS as DPS-D in this work (as DPS approximation itself can be also run with Euler-Maruyama schemes).

[3]Since the authors run DDIM-type sampler, we re-abbreviate $\Pi$GDM as $\Pi$GDM-D in this work (as $\Pi$GDM approximation itself can also be run with Euler-Maruyama schemes).

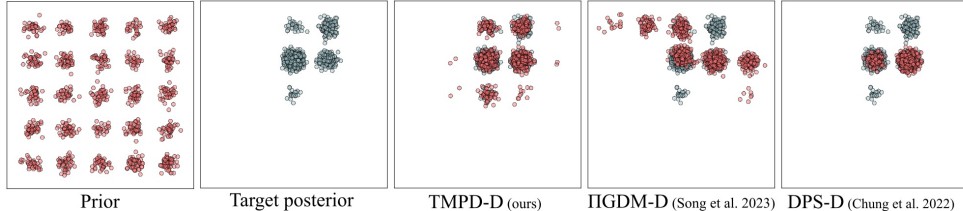

Figure 2: We display the first two dimensions of the GMM inverse problem for one of the measurement models tested ($\mathbf{H}, \sigma_y = 0.1, (d_x, d_y) = (80, 1)$). The blue dots represent samples from the target posterior, while the red dots correspond to samples generated by each of the algorithms used (the names of the algorithms are given at the bottom of each column).

## 6 EXPERIMENTS

In this section, we demonstrate our results as well as the peformance of other approximations to the likelihood provided in Chung et al. (2022a); Song et al. (2023). In particular, we perform comparisons for two of our methods TMPD (an SGM using our approximation) and TMPD-D (a DDPM sampler using equation 11). We compare these to DPS (an SGM sampler using the posterior approximation in equation 13), DPS-D (Chung et al., 2022a) (a DDPM-type sampler using equation 13), ΠGDM (Song et al., 2023) (an SGM sampler using the posterior approximation in equation 14), and finally ΠGDM-D (a DDIM-type sampler using equation 14, but in our experiments a DDPM-type sampler since we set the DDIM hyperparameter $\eta = 1.0$ which is defined in Algorithm 1 by Song et al. (2021a) who show that this is equivalent to a DDPM-type sampler).

The code to run all of our experiments is available at (link).

### 6.1 GAUSSIAN MIXTURE MODEL

We now demonstrate a non-linear SDE example and follow the Gaussian mixture model example of Cardoso et al. (2023) where the data distribution $p_0(\mathbf{x}_0)$ is a mixture of 25 Gaussian distributions. The means and variances of the components of the mixture are given in Appendix E.3. In this case, for each choice of observation $\mathbf{y}$, observation map $\mathbf{H}$ and measurement noise standard deviation $\sigma_y$, the target posterior can be computed explicitly (see Appendix E.3).

To investigate the performance of posterior sampling methods, for each pair of dimensions and observation noise $(d_x, d_y, \sigma_y) \in \{8, 80, 800\} \times \{1, 2, 4\} \times \{10^{-2}, 10^{-1}, 10^0\}$ we randomly generate multiple measurement models $(\mathbf{y}, \mathbf{H}) \in \mathbb{R}^{d_y} \times \mathbb{R}^{d_y \times d_x}$, and equally weight each component of the Gaussian mixture. Further details are given in Appendix E.3. We chose to control the dimension to gain insight into the performance of posterior sampling methods under varying dimensions. We chose to control and the noise level since the different posterior sampling methods have accuracy that depends on the signal to noise ratio. Through randomly varying the observation model, we gain an insight into

Table 1: Sliced Wasserstein for the GMM case. The full table is in Ap. E.3.

| $d_x$ | 8 | 8 | 8 | 80 | 80 | 80 | 800 | 800 | 800 |
|---|---|---|---|---|---|---|---|---|---|
| $d_y$ | 1 | 2 | 4 | 1 | 2 | 4 | 1 | 2 | 4 |
| TMPD-D | 1.6 | **0.7** | **0.3** | 2.7 | **1.0** | **0.3** | 3.1 | **1.4** | **0.4** |
| DTMPD-D | 1.8 | 3.3 | **0.4** | 2.8 | 3.2 | 0.7 | 3.7 | 3.5 | 0.7 |
| DPS-D | 4.7 | 1.8 | 0.7 | 5.6 | 3.2 | 1.2 | 5.8 | 3.5 | 1.4 |
| ΠGDM-D | 2.6 | 2.1 | 3.8 | 3.2 | 2.8 | 0.6 | 3.5 | 3.1 | **0.4** |
| TMPD-D | **1.4** | **0.9** | **0.3** | **2.3** | **1.2** | **0.4** | **2.9** | **1.3** | **0.4** |
| DTMPD-D | 1.8 | 2.7 | 0.5 | 2.6 | 3.2 | 0.8 | 3.4 | 3.4 | 0.8 |
| DPS-D | 4.7 | 1.5 | 0.8 | 5.1 | 3.1 | 1.0 | 5.7 | 3.1 | 1.3 |
| ΠGDM-D | 2.2 | 1.6 | 3.8 | 2.9 | 2.7 | 0.6 | 3.3 | 2.7 | 0.4 |
| TMPD-D | **0.9** | **0.9** | **0.6** | **1.5** | **1.1** | **0.9** | **1.5** | **1.2** | 0.9 |
| DTMPD-D | **0.9** | 1.7 | 0.9 | 1.4 | 2.1 | 0.9 | 1.4 | 2.0 | 1.1 |
| DPS-D | 5.2 | 3.5 | 2.5 | 6.9 | 3.9 | 1.7 | 6.8 | 4.7 | 0.9 |
| ΠGDM-D | 1.5 | 2.3 | 1.8 | **1.6** | 1.4 | **0.9** | 2.0 | 2.0 | **0.6** |

the performance of the posterior sampling methods with different levels of posterior multimodality. This example is interesting because it allows us to study the behaviour of our methods on non-linear problems in high dimensions whilst having access to the target posterior with which to compare (usually, obtaining a 'ground-truth' posterior is not feasible for non-linear problems).

To compare the posterior distribution estimated by each algorithm with the target posterior distribution, we use the sliced Wasserstein (SW) distance defined in Appendix E.3. We use $10^4$ slices for the

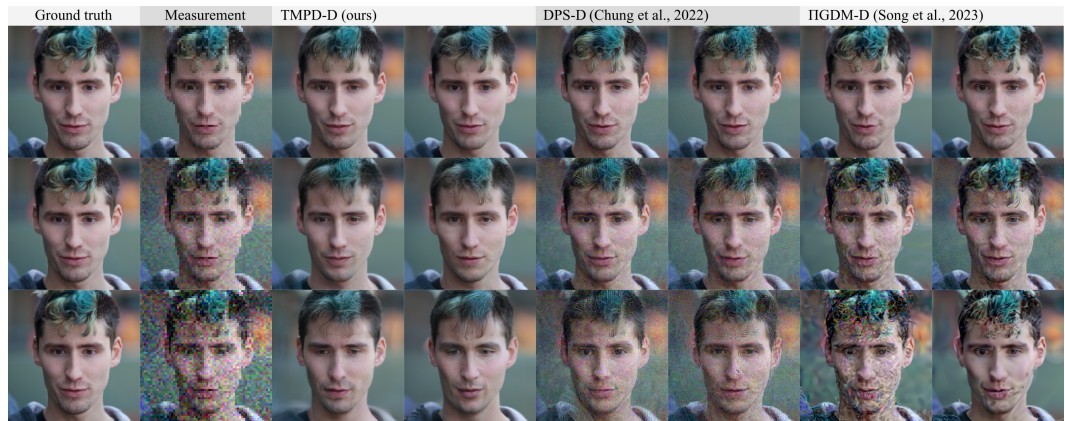

Figure 3: Illustration of the robustness of our method with increasing Gaussian noise for a $4\times$ super-resolution ($256 \times 256 \rightarrow 64 \times 64$) problem. The top row measurement has $\sigma_y = 0.01$, the middle row $\sigma_y = 0.05$ and the bottom row $\sigma_y = 0.1$. For more details and examples, see Appendix E.4.

Table 2: Summary of results for increasingly noisy observations for inpainting (IP) and super-resolution (SR) problems on CIFAR-10 1k validation set. (Top rows) $\sigma_y = 0.01$, (middle rows) $\sigma_y = 0.05$, (bottom rows) $\sigma_y = 0.1$. The full table is in Ap. E.4.

| | $2\times$ 'nearest' SR | | | $4\times$ 'bicubic' SR | | | 'Box' mask IP | | | 'Half' mask IP | | |
|---|---|---|---|---|---|---|---|---|---|---|---|---|
| Method | FID↓ | PSNR↑ | SSIM↑ | FID↓ | PSNR↑ | SSIM↑ | FID↓ | PSNR↑ | SSIM↑ | FID↓ | PSNR↑ | SSIM↑ |
| TMPD-D | **34.5** | 23.7 | 0.806 | **38.2** | 19.9 | 0.541 | **34.2** | 23.4 | 0.778 | **38.5** | 16.0 | 0.577 |
| DPS-D | 43.4 | 24.7 | 0.832 | 54.5 | 20.0 | 0.541 | 39.9 | 24.2 | 0.800 | 43.6 | 16.2 | 0.595 |
| ΠGDM-D | 38.7 | 24.3 | 0.819 | 42.7 | 19.6 | 0.529 | 40.5 | 24.2 | 0.804 | 43.1 | 16.2 | 0.604 |
| TMPD-D | **37.0** | 21.9 | 0.737 | **36.2** | 18.4 | 0.432 | **36.5** | 22.2 | 0.722 | **41.9** | 15.3 | 0.573 |
| DPS-D | 101 | 21.8 | 0.704 | 126 | 18.0 | 0.422 | 101 | 21.7 | 0.681 | 107 | 15.9 | 0.507 |
| ΠGDM-D | 109 | 20.7 | 0.657 | 120 | 17.3 | 0.402 | 90.7 | 21.4 | 0.673 | 93.1 | 15.4 | 0.506 |
| TMPD-D | **37.2** | 20.6 | 0.642 | **35.9** | 17.2 | 0.346 | **38.1** | 21.1 | 0.650 | **41.1** | 15.2 | 0.485 |
| DPS-D | 140 | 19.1 | 0.561 | 155 | 16.7 | 0.350 | 86.4 | 19.6 | 0.579 | 93.2 | 14.7 | 0.398 |
| ΠGDM-D | 161 | 17.4 | 0.499 | 217 | 14.5 | 0.281 | 130 | 18.3 | 0.524 | 128 | 14.1 | 0.397 |

SW distance and compare 1000 samples of TMPD-D, ΠGDM-D and DPS-D in Tables 1 obtained using 1000 denoising steps and 1000 samples of the true posterior distribution.

Table 1 indicates the Central Limit Theorem (CLT) 95% confidence intervals obtained by considering 20 randomly selected measurement models (**H**) for each setting ($d_x, d_x, \sigma_y$). Figure 2 shows the first two dimensions of the estimated posterior distributions corresponding to the configurations $(80, 1)$ from Table 1 for one of the randomly generated measurement model (**H**, $\sigma_y = 0.1$). These illustrations give us insight into the behaviour of the algorithms and their ability to accurately estimate the posterior distribution. We see that TMPD-D estimates the target posterior well compared to ΠGDM-D and DPS-D. TMPD-D covers all of the modes, whereas ΠGDM-D and DPS-D do not.

We perform the same experiment using 1000 samples of TMPD, DTMPD, DPS and ΠGDM, obtained using 1000 Euler-Maruyama time-steps, and results are shown in Appendix E.3.

## 6.2 NOISY OBSERVATION INPAINTING AND SUPER-RESOLUTION

We consider inpainting and super-resolution problems on the FFHQ ($256 \times 256 \times 3$) (Karras et al., 2019) and CIFAR-10 ($32 \times 32 \times 3$) (Krizhevsky et al., 2009) datasets. We compare TMPD to ΠDGM and DPS. We also compare score-based diffusion models with their denoising-diffusion counterparts (denoted with suffix, -D). We use pretrained denoising networks that are available here for all the methods. The methods TMPD, ΠGDM and DPS all have the same numerical solver of their respective reverse-SDE, and TMPD-D, ΠGDM-D and DPS-D all use DDPM since DDIM and DDPM are equivalent algorithms with our chosen hyperparameter $\eta = 1.0$. Therefore, the sampling

methods being compared only differ in the $\nabla_{\mathbf{x}_t} \log p_{\mathbf{y}|t}(\mathbf{y}|\mathbf{x}_t)$ term in their reverse-SDE, and so this experiment allows us to study the behaviour of our method on high-dimensional inpainting and super-resolution compared to the different approximations of the smoothed likelihood. For quantitative comparison, we focus on three widely used distribution distances, Fréchet Inception Distance (FID), Kernel Inception Distance (KID) and Learned Perceptual Image Patch Similarity (LPIPS) distance, to evaluate consistency with the whole dataset by using summary statistics from the CIFAR-10 dataset, which are provided by Song et al. (2020). We also evaluate observation data consistency using various distances between a sampled image and ground truth image, mean-squared-error (MSE), peak signal-to-noise-ratio (PSNR) and structural similarity index measure (SSIM).

A summary of the results for CIFAR-10 sampled using VE DDPM are shown in Table 2, which reports FID scores that effectively evaluate consistency with the prior and two distortion metrics, PSNR and SSIM that evaluate observation data consistency to ground truth images. The complete results are in Table 5 for VE DDPM and Table 6 for score-based VE SDE. Images are normalized to the range $[0, 1]$ and it is on this scale that we add Gaussian measurement noise with standard deviation $\sigma_{\mathbf{y}} \in \{0.01, 0.05, 0.1\}$. For inpainting, we use 'box' and 'half' mask on images on CIFAR-10. For 'half'-type inpainting, we mask out a $16 \times 16$ right half region of the image; for box-type inpainting, we mask out an $8 \times 8$ box region following Cardoso et al. (2023). We illustrate the scalability of our method to FFHQ and robustness to large noise in Fig. 3. For CIFAR-10, We use a downsampling ratios of $2\times$ on each axis ($32 \times 32 \rightarrow 16 \times 16$) with a nearest-neighbour downsampling method and $4\times$ ($32 \times 32 \rightarrow 8 \times 8$) with bicubic downsampling. For FFHQ dataset we use a downsampling ratio of 4 ($256 \times 256 \rightarrow 64 \times 64$) with nearest-neighbour. Whereas no hyperparameters are required for our method, we chose the DPS scale hyperparameter by optimising LPIPS, MSE, PSNR and SSIM on a validation set of 128 images (see Fig. 4 for an example). For more experimental details including illustration of samples used to generate the tables can be found in Appendix E.4.

# 7 DISCUSSIONS, LIMITATIONS AND FUTURE WORK

In this paper, we introduced TMPD, a diffusion modelling approach to solve inverse problems and sample from conditional distributions using unconditional diffusion models. On various tasks, TMPD achieves competitive quality with other methods that aim to solve the noisy, linear inverse problem while avoiding the expensive, problem-specific training of conditional models. We can use problem-agnostic diffusion models to solve certain problems that would be cost-ineffective to address individually with conditional diffusion models, leading to a much wider set of applications. The ability to handle measurement noise also gives TMPD the potential to address certain real-world applications, such as CT and MRI imaging with Gaussian noise (Sijbers & den Dekker, 2004; Song et al., 2021b). Despite having on-par restoration results with ΠGDM (Song et al., 2023), TMPD is slower, as each iteration costs more memory and compute due to the Jacobian over the score model. Even with a diagonal and row-sum approximation to the Jacobian, the method is around $1.5\times$ slower than DPS and ΠGDM. The row-sum approximation may not be suitable for inverse problems with more complicated or non-linear observation operators, therefore, it would be helpful to explore more efficient sampling techniques, such as ensemble methods or low rank approximations to the Jacobian.

On the positive side, our approach does not require any hyperparameter tuning and is more principled compared to existing approaches, as shown in Section 4. We show that our method, unlike DPS and ΠGDM, does not fail for cases where the additive Gaussian noise is large. We provided a way to analyse similar methods and our moment approximations can be analysed more rigorously to provide deeper theoretical results for these kinds of methods. Our future work plans include expanding the analysis we provided in this work.

## REPRODUCIBILITY STATEMENT

We have made the following efforts to facilitate reproducibility of our work. (i) Our experiments in Sec 6 are conducted on publicly available datasets and model checkpoints. (ii) We provide open-source code for our method and all of our experiments (iii) We include a detailed description of our

algorithm in Algorithm 1. (iv) We discuss all the key hyperparameters and evaluation metrics to reproduce our experiments in Sec. 6 and App. E.

## ETHICS STATEMENT

Our results do not increase or introduce new specific risks, but as a novel way to sample images and generate new data, usual risks of image generation models apply to our work. In particular, the methods in this paper can be used to generate faces of people who do not exist as well as edit existing images in certain ways that creates misinformation risk. However, our method as it is proposed here do not integrate text-based editing, therefore we think these risks are very low for the particular version of this method.

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

# A  Proofs for Section 3

## A.1  Proof of Proposition 1

In order to prove this result, we adapt Theorem 1 of Meng et al. (2021). We write the proof for generic exponential family which can be adapted to our case easily. Let us consider

$$p_{t|0}(\mathbf{x}_t|\mathbf{x}_0) = \mathcal{N}(\mathbf{x}_t; \sqrt{\alpha_t}\mathbf{x}_0, v_t \mathbf{I}_{d_x}).$$

and write $p_t(\mathbf{x}_t) = \int p_{t|0}(\mathbf{x}_t|\mathbf{x}_0)p_0(\mathbf{x}_0)\mathrm{d}\mathbf{x}_0$. We are interested in finding the moments of the posterior

$$p_{0|t}(\mathbf{x}_0|\mathbf{x}_t) = \frac{p_{t|0}(\mathbf{x}_t|\mathbf{x}_0)p_0(\mathbf{x}_0)}{p_t(\mathbf{x}_t)}.$$

Let us redefine the right handside in this Bayes' rule using the exponential family parameterisation of the Gaussian $p_{t|0}(\mathbf{x}_t|\mathbf{x}_0)$

$$p(\boldsymbol{\eta}_0|\mathbf{x}_t) = \frac{p_{t|0}(\mathbf{x}_t|\boldsymbol{\eta}_0)q_0(\boldsymbol{\eta}_0)}{p_t(\mathbf{x}_t)},$$

where $\boldsymbol{\eta}_0 = \mathbf{x}_0\sqrt{\alpha_t}/v_t$ and

$$p_{t|0}(\mathbf{x}_t|\boldsymbol{\eta}_0) = e^{\boldsymbol{\eta}_0^\top \mathbf{x}_t - \psi(\boldsymbol{\eta}_0)}q(\mathbf{x}_t),$$

where

$$q(\mathbf{x}_t) = ((2\pi)^d v^{2d})^{-1/2}e^{-\mathbf{x}_t^\top \mathbf{x}_t/2v_t}$$

Now let

$$\lambda(\mathbf{x}_t) = \log\frac{p_t(\mathbf{x}_t)}{q(\mathbf{x}_t)},$$

we can rewrite

$$p(\boldsymbol{\eta}_0|\mathbf{x}_t) = e^{\boldsymbol{\eta}_0^\top \mathbf{x}_t - \psi(\mathbf{x}_t) - \lambda(\mathbf{x}_t)}q_0(\boldsymbol{\eta}_0).$$

It is then easy to show that the moments of $\boldsymbol{\eta}_0$ are given by (Meng et al., 2021)

$$\mathbb{E}[\boldsymbol{\eta}_0|\mathbf{x}_t] = \mathbf{J}_\lambda(\mathbf{x}_t),$$
$$\mathbb{E}[\boldsymbol{\eta}_0^\top|\mathbf{x}_t] = \mathbf{J}_\lambda(\mathbf{x}_t)^\top,$$
$$\mathbb{E}[\boldsymbol{\eta}_0\boldsymbol{\eta}_0^\top|\mathbf{x}_t] = \mathbf{S}_\lambda(\mathbf{x}_t) + \mathbf{J}_\lambda(\mathbf{x}_t)\mathbf{J}_\lambda(\mathbf{x}_t)^\top,$$

where $\mathbf{J}_\lambda$ is the Jacobian of $\lambda$ w.r.t. $\mathbf{x}_t$ and $\mathbf{S}_\lambda$ is the Hessian of $\lambda$ w.r.t. $\mathbf{x}_t$. Recall now that $\boldsymbol{\eta}_0 = \mathbf{x}_0 \sqrt{\alpha_t}/v_t$ and

$$\lambda(\mathbf{x}_t) = \log p_t(\mathbf{x}_t) + \frac{\mathbf{x}_t^\top \mathbf{x}_t}{2v_t} + C$$

$$\mathbf{J}_\lambda(\mathbf{x}_t) = \nabla_{\mathbf{x}_t} \log p_t(\mathbf{x}_t) + \frac{\mathbf{x}_t}{v_t}$$

$$\mathbf{S}_\lambda(\mathbf{x}_t) = \nabla^2_{\mathbf{x}_t} \log p_t(\mathbf{x}_t) + \frac{1}{v_t}\mathbf{I}_{d_x}.$$

This implies that

$$\mathbb{E}[\mathbf{x}_0|\mathbf{x}_t] = \frac{1}{\sqrt{\alpha_t}}(\mathbf{x}_t + v_t \nabla_{\mathbf{x}_t} \log p_t(\mathbf{x}_t)),$$

which proves the Tweedie's formula for the mean. For the covariance, note that

$$\mathrm{Cov}(\boldsymbol{\eta_0}|\mathbf{x}_t) = \mathbb{E}[\boldsymbol{\eta_0}\boldsymbol{\eta}_0^\top|\mathbf{x}_t] - \mathbb{E}[\boldsymbol{\eta_0}|\mathbf{x}_t]\mathbb{E}[\boldsymbol{\eta_0}|\mathbf{x}_t]^\top$$

$$= \nabla^2 \log p_t(\mathbf{x}_t) + \frac{1}{v_t}\mathbf{I}_{d_x}.$$

Since

$$\mathrm{Cov}(\boldsymbol{\eta_0}|\mathbf{x}_t) = \frac{\alpha_t}{v_t^2}\mathrm{Cov}(\mathbf{x}_0|\mathbf{x}_t),$$

we conclude

$$\mathrm{Cov}(\mathbf{x}_0|\mathbf{x}_t) = \frac{v_t}{\alpha_t}(\mathbf{I}_{d_x} + v_t \nabla^2 \log p_t(\mathbf{x}_t)),$$

which concludes the proof. $\qquad\square$

## B  PROOFS FOR SECTION 4

### B.1  PROOF OF PROPOSITION 3

If $p_{\mathrm{data}}$ is Gaussian, then the full process $\mathbf{x}_t$ is a Gaussian process. In particular, $(\mathbf{x}_0, \mathbf{x}_t)$ are jointly Gaussian:

$$\mathbf{x}_0|\mathbf{x}_t \propto \mathcal{N}(\mathbb{E}[\mathbf{x}_0|\mathbf{x}_t], \mathbb{V}[\mathbf{x}_0|\mathbf{x}_t]),$$

where $\mathbb{V}$ denotes the covariance. However, the right-hand side is precisely the approximation we make, due to Proposition 1. Therefore, the approximation we make in equation 9 is correct. Adding this to the learned drift $\mathbf{s}_\theta(\mathbf{x}_t, t) = \nabla \log p_t(\mathbf{x}_t)$, we get an expression for $\nabla \log p_t(\mathbf{x}_t|\mathbf{y})$ by equation 4.

### B.2  PROOF OF THEOREM 1

We first introduce three SDEs.

**Conditioned SDEs** : The first pair of SDEs is given by an OU process, but started in the conditional distribution:

$$\mathrm{d}\mathbf{x}_t^c = -\frac{1}{2}\mathbf{x}_t^c\mathrm{d}t + \mathrm{d}\mathbf{w}_t, \quad \mathbf{x}_0^c \sim p_0^c := p_{\mathrm{data}}(\mathbf{x}_0|\mathbf{y}). \tag{15}$$

We denote the marginals of $\mathbf{x}_t^c$ by $p_t^c$ to differentiate them from the marginals $p_t$ of the OU process started in the correct distribution, defined in equation 1. Note that $p_t^c = p_t(\mathbf{x}_t|\mathbf{y})$. The time reversal $\mathbf{z}_t = \mathbf{x}_{T-t}$ then satisfies the SDE

$$\begin{aligned}\mathrm{d}\mathbf{z}_t = &\frac{1}{2}\mathbf{z}_t\mathrm{d}t + \nabla_{\mathbf{z}_t} \log p_{T-t}^c(\mathbf{z}_t)\mathrm{d}t + \mathrm{d}\mathbf{w}_t \\ = &\frac{1}{2}\mathbf{z}_t\mathrm{d}t + \nabla_{\mathbf{z}_t} \log p_{T-t}(\mathbf{z}_t)\mathrm{d}t + \nabla_{\mathbf{z}_t} \log p_{\mathbf{y}|T-t}(\mathbf{y}|\mathbf{z}_t)\mathrm{d}t + \mathrm{d}\mathbf{w}_t \\ \mathbf{z}_0 \sim &p_T^c,\end{aligned} \tag{16}$$

where we used equation 4.

Solutions to the above reverse SDE will sample our target measure $p_0^c = p_{\mathrm{data}}(\mathbf{x}_0|\mathbf{y})$ at final time. Therefore, we want to study how our algorithm approximates solutions of $\mathbf{x}^c$.

**Intermediate Gaussian SDE:** Instead of bounding the distance of solutions to the above conditioned reverse SDE to our algorithm, we will instead introduce an intermediate process, which we will later use in a triangle inequality. The process will be a Gaussian process. Therefore, we denote it with a superscript $G$.

The process $\mathbf{x}_t^G$ will be defined analogous to equation 1, but assuming that it is started in $\mathcal{N}(\mathbf{x}_0; \boldsymbol{\mu}_0, \boldsymbol{\Sigma}_0)$ instead of $p_{\text{data}}$. Since the forward SDE in equation 1 is linear, all of the marginals of $\mathbf{x}_t^G$, which we denote by $p_t^G$, will also be Gaussian.

Again, we define a conditioned version of $\mathbf{x}$, called $\mathbf{x}^{G,c}$ analogous to equation 15, i.e. $\mathbf{x}_0^{G,c}$ will be distributed as $\mathbf{x}_0^G$ conditioned on $\mathbf{y} = y$, i.e. $p^G(\mathbf{x}_0|\mathbf{y} = y)$. Since we have a linear observation model, $p^G(\mathbf{x}_0|\mathbf{y} = y)$ is still a Gaussian, and therefore $\mathbf{x}^{G,c}$ will also be a Gaussian process. Its reverse SDE $\mathbf{z}^{G,c}$ is defined through equation 16, just that every appearance of $p$ will also have a superscript $G$,

$$\begin{aligned} \mathrm{d}\mathbf{z}_t =& \frac{1}{2}\mathbf{z}_t\mathrm{d}t + \nabla_{\mathbf{z}_t}\log p_{T-t}^G(\mathbf{z}_t)\mathrm{d}t + \nabla_{\mathbf{z}_t}\log p_{T-t}^G(\mathbf{y}|\mathbf{z}_t)\mathrm{d}t + \mathrm{d}\mathbf{w}_t \\ \mathbf{z}_0 \sim& p_T^c, \end{aligned} \tag{17}$$

**Algorithm SDE:** Finally, we define a third reverse SDE, which is the SDE that we are discretizing when implementing our algorithm. It is given by

$$\begin{aligned} \mathrm{d}\mathbf{z}_t =& \frac{1}{2}\mathbf{z}_t\mathrm{d}t + \nabla_{\mathbf{z}_t}\log p_{T-t}(\mathbf{z}_t)\mathrm{d}t + f_{T-t}(\mathbf{z}_t)\mathrm{d}t + \mathrm{d}\mathbf{w}_t \\ \mathbf{z}_0 \sim& p_T^c, \end{aligned} \tag{18}$$

where

$$f_t(\mathbf{z}_t) = \nabla_{\mathbf{z}_t}\mathbb{E}[\mathbf{x}_0|\mathbf{x}_t = z_t]\mathbf{H}^T\mathbb{V}[\mathbf{x}_0|\mathbf{x}_t = z_t]^{-1}(\mathbf{y} - \mathbf{H}\mathbb{E}[\mathbf{x}_0|\mathbf{x}_t = z_t]), \tag{19}$$

and $\mathbb{V}$ denotes the conditional covariance. Except for approximation errors due to time discretization and in the initial conditions, this is the SDE we are sampling from in our algorithm. We denote the marginal of $\mathbf{z}_t$ by $q_t$. Now we are ready to write our proof.

*Proof of Theorem 1.* We have that $p_{\text{data}}(\cdot|\mathbf{y}) = p_0^c$

$$\|p_0^c - q_T\|_{\text{TV}} \le \|p_0^c - p_0^{G,c}\|_{\text{TV}} + \|p_0^{G,c} - q_T\|_{\text{TV}}$$

We will bound the second term using Pinsker's inequality to get a $KL$-term on the path space, as done in Chen et al. (2023). The proof then consists in bounding the first total variation term and the resulting KL term.

**Bounding the first term:** The first term is the total variation distance between $p_0(\mathbf{x}_0|\mathbf{y} = y)$ and $p_0^G(\mathbf{x}_0^G|\mathbf{y} = y)$. Now,

$$\begin{aligned} p_0(\mathbf{x}_0 = x_0|\mathbf{y} = y) &= \frac{p_{\mathbf{y}|0}(\mathbf{y} = y|\mathbf{x}_0 = x_0)p_0(\mathbf{x}_0 = x_0)}{p_{\mathbf{y}}(\mathbf{y} = y)} \\ &= \frac{p_{\mathbf{y}|0}^G(\mathbf{y} = y|\mathbf{x}_0 = x_0)p_0^G(\mathbf{x}_0 = x_0)}{p_{\mathbf{y}}^G(\mathbf{y} = y)} \frac{p_0(\mathbf{x}_0 = x_0)}{p_0^G(\mathbf{x}_0^G = x_0^G)} \frac{p_{\mathbf{y}}(\mathbf{y} = y)}{p_{\mathbf{y}}^G(\mathbf{y} = y)} \\ &= p_{\mathbf{y}|0}^G(\mathbf{x}_0 = x_0|\mathbf{y} = y)\exp(\Phi(x_0))\frac{p_{\mathbf{y}}(\mathbf{y} = y)}{p_{\mathbf{y}}^G(\mathbf{y} = y)}, \end{aligned}$$

where we used that the conditional distribution of $\mathbf{y}$ given $\mathbf{x}_0$ does not depend on the distribution of $\mathbf{x}_0$. For the last term we get

$$\begin{aligned} p_{\mathbf{y}}(\mathbf{y} = y) &= \int p_{\mathbf{y}|0}(\mathbf{y} = y|\mathbf{x}_0 = x_0)p_0(\mathbf{x}_0 = x_0)\mathrm{d}x_0 \\ &= \int p_{\mathbf{y}|0}^G(\mathbf{y} = y|\mathbf{x}_0 = x_0)p_0^G(\mathbf{x}_0 = x_0)\exp(\Phi(x_0))\mathrm{d}x_0 \\ &= Np_{\mathbf{y}}^G(\mathbf{y} = y), \end{aligned}$$

with $N \in [1/M, M]$. Therefore,

$$\|p_0^c - p_0^{G,c}\|_{\text{TV}} \le \int |\frac{p_0^c}{p_0^{G,c}}(x_0) - 1|p_0^{G,c}(x_0)\mathrm{d}x_0 \le M^2 - 1,$$

where we used that $M - 1$ is always greater or equal to $1 - \frac{1}{M}$, since $M \ge 1$.

**Bounding the second term:** For the second term we get that $p_0^{G,c}$ is the final marginal of equation 17, while $q_T$ is the final marginal of equation 18. We now use Pinsker's inequality,

$$\|p_0^{G,c} - q_T\|_{\text{TV}} \le \sqrt{\text{KL}(p_0^{G,c}\|q_T)}.$$

We denote the path measures induced by equation 17 and equation 18 by $\mathbb{P}^{G,c}$ and $\mathbb{Q}$ respectively. They have final time marginals $p_0^{G,c}$ and $q_T$. Therefore, we can bound the KL-Divergence of the marginals $p^{G,c}$ and $q_T$ by the KL-Divergence on the full path space:

$$\text{KL}(p_0^{G,c}\|q_T) \le \text{KL}(\mathbb{P}^{G,c}\|\mathbb{Q}).$$

We will assume that we can apply Girsanovs theorem and show later that this is justified. Using Girsanovs theorem, we can evaluate the Radon-Nikodym derivative $\frac{d\mathbb{Q}}{d\mathbb{P}^{G,c}}$ on the path space and therefore calculate the KL-divergence:

$$
\begin{aligned}
\text{KL}(\mathbb{P}^{G,c}\|\mathbb{Q}) =& \mathbb{E}_{\mathbb{P}^{G,c}}[\log \frac{d\mathbb{P}^{G,c}}{d\mathbb{Q}}] \\
=& \mathbb{E}_{\mathbb{P}^{G,c}}\left[-\int_0^t \nabla \log p_t(x_t) + f_{T-t}(\mathbf{x}_t) - \nabla \log p_{T-t}^{G,c}(\mathbf{x}_t)dw_t\right] \\
&+ \mathbb{E}_{\mathbb{P}^{G,c}}\left[\int_0^t \|\nabla \log p_t(x_t) + f_{T-t}(\mathbf{x}_t) - \nabla \log p_{T-t}^{G,c}(\mathbf{x}_t)\|^2 dt\right] \\
=& \int_0^t \mathbb{E}_{p_t^{G,c}}[\|\nabla \log p_t(x_t) + f_{T-t}(\mathbf{x}_t) - \nabla \log p_{T-t}^{G,c}(\mathbf{x}_t)\|^2]dt,
\end{aligned}
$$

where the term on the second line drops out because stochastic integrals have expectation 0. Now we have the drift of the Gaussian SDE given by

$$
\begin{aligned}
& \nabla \log p_t^{G,c}(x_t) \\
=& \nabla \log p_t^G(x_t) + \nabla \log p^G(\mathbf{y}|\mathbf{x}_t^G = x_t) \\
=& \nabla \log p_t^G(x_t) \\
& + \nabla_{x_t}\mathbb{E}[\mathbf{x}_0^G|\mathbf{x}_t^G = x_t]\mathbf{H}^\top(\mathbf{H}\mathbb{V}[\mathbf{x}_0^G|\mathbf{x}_t^G = x_t]\mathbf{H}^T + \sigma_y^2\mathbf{I}_{d_y})^{-1}(\mathbf{y} - \mathbf{H}\mathbb{E}[\mathbf{x}_0^G|\mathbf{x}_t^G = x_t]) \\
=:& \nabla \log p_t^G(x_t) + \tilde{f}_t(x_t),
\end{aligned}
\tag{20}
$$

see equation 43, while the drift of the algorithm SDE is given by

$$\nabla \log p_t(x_t) + f_t(x_t), \tag{21}$$

where $f_t$ is given by equation 19. We see that the difference in the SDE drifts mainly consists of differences between conditional moments of $p_t$ and $p_t^G$, as well as the derivatives of the conditional expectations. Therefore, the main difficulty of the proof is to bound these.

The density of the conditional distribution of $p_{0|t}$ is given by

$$
\begin{aligned}
p_{0|t}(x_0|x_t) =& \frac{p_{0,t}(x_0, x_t)}{p_t(x_t)} = \frac{p_{0,t}^G(x_0, x_t)\exp(\Phi(x_0))}{p(x_t)} = \frac{p_{0|t}^G(x_0|x_t)p^G(x_t)\exp(\Phi(x_0))}{p(x_t)} \\
=& p_{0|t}^G \exp(\Phi(x_0) - \Phi(x_t)),
\end{aligned}
\tag{22}
$$

where

$$\exp(\Phi_t(x_t)) = \frac{p_t(x_t)}{p_t^G(x_t)} = \mathbb{E}[\exp(\Phi(\mathbf{x}_0))|\mathbf{x}_t = x_t]. \tag{23}$$

In the last equality, we used that

$$\frac{d\mathbb{P}}{d\mathbb{P}^G}(x_{[0,t]}) = \exp(\Phi(x_0)),$$

where we denoted by $\mathbb{P}$ the path measure induced by equation 15. Therefore, also their marginals $p_t = \mathbb{P}_t^G$ and $p_t = \mathbb{P}_t$ have relative densities, which are given by integrating out the density to time $t$, as we did in equation 23.

By assumption $\exp(\Phi)$ is bounded from above and below by $M$ and $1/M$ respectively, and by equation 23 the same holds for $\exp(\Phi_t)$. Therefore, by equation 22, $p_{t|0}$ is absolutely continuous with respect to $p_{t|0}^G$ with a density that is bounded from above and below by $M^2$ and $1/M^2$ respectively. We now obtain

$$|\mathbb{E}[\mathbf{x}_0|\mathbf{x}_t = x_t] - \mathbb{E}[\mathbf{x}_0^G|\mathbf{x}_t^G = x_t]|$$

$$= \left|\int x_0[p(x_0|x_t) - p^G(x_0|x_t)]\mathrm{d}x_0\right| = \left|\int x_0 p^G(x_0|x_t)(\exp(\Phi(x_0) - \Phi(x_t)) - 1)\mathrm{d}x_0\right|$$

$$\leq (M^2 - 1)\mathbb{E}[|\mathbf{x}_0^G| | \mathbf{x}_t^G = x_t^G]$$

The same holds for every entry of the covariance matrix. We denote

$$V_{ij} := \mathbb{V}[\mathbf{x}_0|\mathbf{x}_t]_i j,$$
$$V_{ij}^G := \mathbb{V}[\mathbf{x}_0^G|\mathbf{x}_t^G]_{ij} = N_{ij},$$
$$|V_{ij} - V_{ij}^G| \leq (M^2 - 1)N_{ij}$$
$$N_{ij} := \mathbb{E}[|\mathbf{x}_{0,i}^G \mathbf{x}_{0,j}^G| \, |\mathbf{x}_t^G]$$

Now we get that

$$\|V - V^G\|_F = \left(\sum_{ij}(V_{ij} - V_{ij}^G)^2\right)^{1/2} \leq (M^2 - 1)\|V^G\|_F \max_{ij}(N_{ij}).$$

We can bound $N_{ij}$ by

$$N_{ij} \leq \mathbb{E}[(\mathbf{x}_{0,i}^G)^2 + (\mathbf{x}_{0,j}^G)^2|\mathbf{x}_t^G] \leq \mathrm{Tr}(\mathbb{V}[\mathbf{x}_0^G|\mathbf{x}_t^G = x_t]). \tag{24}$$

The latter term does not actually depend on $x_t$, but only on $t$. By using the formulas in equation 39 we see thta it can be bounded independently of $t$ (only depending on $m_0$ and $\Sigma_0$). Therefore, we can put it into a constant. We get that

$$\|V - V^G\|_F \lesssim (M^2 - 1)\|V^G\|_F,$$

where

$$a \lesssim b \quad \Leftrightarrow \quad a \leq Cb,$$

with a constant $C$ only depending on $m_0$, $\Sigma_0$, $H$, $\sigma^y$ and the observation $y$. Now let $v$ be a vector with $\|v\| = 1$. It holds that

$$v^T V_{ij} v = \int (v^T(x_0 - \mathbb{E}[\mathbf{x}_0|\mathbf{x}_t = x_t]))^2 p_{0|t}^G(x_0|x_t)\exp(\Phi(x_0) - \Phi_t(x_t))\mathrm{d}x_0$$

$$= \frac{1}{M}v^T V_{ij}^G v + N \int (v^T(\mathbb{E}[\mathbf{x}_0^G|\mathbf{x}_t^G = x_t] - \mathbb{E}[\mathbf{x}_0|\mathbf{x}_t = x_t]))^2 p_{0|t}(x_0|x_t)\mathrm{d}x_0 \tag{25}$$

$$\geq v^T V_{ij}^G v \frac{1}{M}$$

with $N \geq 1/M$. Since $V$ and $V^G$ are positive semidefinite, symmetric matrices, this implies that all eigenvalues of $V$ are bounded by the lowest eigenvalue of $V^G$ times $1/M$. We define

$$T_1 = HV^G H^T + \sigma_y^2 I_{d_y} \tag{26}$$
$$T_2 = HVH^T + \sigma_y^2 I_{d_y}. \tag{27}$$

We then need to bound

$$\|T_1^{-1} - T_2^{-1}\| \leq \|T_1 - T_2\|(\|T_1^{-1}\| + \|T_2^{-1}\|).$$

We start with

$$\|T_1 - T_2\|_{\mathrm{op}} \leq \|H\|^2 \|V - V^G\|_{\mathrm{op}} \lesssim \|V - V^G\|_F \lesssim (M^2 - 1)\|V^G\|_F.$$

We also used the equivalence of the Frobenius and operator norm here. Due to our eigenvalue bound (equation 25) on $V$, we also get an analogous bound on $V + \sigma_y I_{d_y}$. Therefore,

$$\|T_1^{-1} - T_2^{-1}\|_{\mathrm{op}} \lesssim (M^2 - 1)\|V^G\|_F\|V^G\|_F^{-1}(1 + M) \leq (M^2 - 1)(M + 1) \leq M^3 - 1$$

where we used that the operator norm of the inverse, is the inverse of the operator norm and the equivalence of the operator norm to the Frobenius norm.

Finally, we need to bound

$$\nabla_{x_t} \int x_0 p(x_0|x_t) \mathrm{d}x_0$$

$$= \int x_0 \nabla_{x_t} p^G(x_0|x_t) \exp(\Phi(x_0) - \Phi(x_t)) \mathrm{d}x_0$$

$$= \int x_0 \exp(\Phi(x_0) - \Phi(x_t)) \nabla_{x_t} p^G(x_0|x_t) \mathrm{d}x_0 + \int x_0 \nabla_{x_t} \Phi(x_t) \exp(\Phi(x_0) - \Phi(x_t)) p^G(x_0|x_t) \mathrm{d}x_0$$

$$= \int x_0 \exp(\Phi(x_0) - \Phi(x_t)) \nabla_{x_t} p^G(x_0|x_t) \mathrm{d}x_0 + \nabla_{x_t} \Phi(x_t) \mathbb{E}[\mathbf{x}_0|\mathbf{x}_t = x_t].$$

(28)

Also

$$\left\| \int x_0 \exp(\Phi(x_0) - \Phi(x_t)) \nabla_{x_t} p^G(x_0|x_t) \mathrm{d}x_0 - \nabla_{x_t} \mathbb{E}[\mathbf{x}_0^G|\mathbf{x}_t^G = x_t] \right\|$$

$$= \left\| \int \nabla_{x_t} p^G(x_0|x_t) x_0^T (\exp(\Phi(x_0) - \Phi(x_t)) - 1) \mathrm{d}x_0 \right\|$$

$$\leq (M^2 - 1) \int p^G(x_0|x_t) \|\nabla \log p^G(x_0|x_t) x_0^T\| \mathrm{d}x_0$$

$$\leq (M^2 - 1) (\int p^G(x_0|x_t) \|\nabla \log p^G(x_0|x_t) x_0^T\| \mathrm{d}x_0)^{1/2}$$

(29)

$$\leq (M^2 - 1) (\int p^G(x_0|x_t) x_0^T \Sigma_{0|t}^{-1} (x_0 - m_{0|t}^{x_t}) \mathrm{d}x_0)^{1/2}$$

$$\lesssim (M^2 - 1) (\int p^G(x_0|x_t) (x_0 - m_{0|t}^{x_t})^T \Sigma_{0|t}^{-1} (x_0 - m_{0|t}^{x_t}) \mathrm{d}x_0)^{1/2} \lesssim \|x_t\|(M^2 - 1)\sqrt{d_x}$$

$$\lesssim \|x_t\|(M^2 - 1),$$

here we used that we can upper bound the operator norm of a positive semidefinite matrix by its trace from the third to the fourth line. We denoted by $m_{0|t}^{x_t}$ and $\Sigma_{0|t}$ the mean and covariance of $p^G(\mathbf{x}_0^G|\mathbf{x}_t^G = x_t)$. From the second to last to the last line we used that the $m_{0|t}^{x_t}$ depends on $x_t$ linearly, and the magnitude of the linear dependence can be bounded uniformly in $t$ (see equation 39). The integral in the last line is the variance of a standard normal random variable, which evaluates to $d_x$. Furthermore,

$$\nabla \Phi_t(x_t) = \mathbb{E}[\nabla \Phi_0(\mathbf{x}_0)|\mathbf{x}_t = x_t] \leq L,$$

see for example the Proof of Theorem 1 in Pidstrigach et al. (2023). Therefore,

$$\|\nabla_{x_t} \mathbb{E}[\mathbf{x}_0^G|\mathbf{x}_t^G = x_t] - \nabla_{x_t} \mathbb{E}[\mathbf{x}_0|\mathbf{x}_t = x_t]\|$$

$$\lesssim \|x_t\|(M^2 - 1) + L\|\mathbb{E}[\mathbf{x}_0|\mathbf{x}_t = x_t]\|$$

$$\leq \|x_t\|(M^2 - 1) + L(\|\mathbb{E}[\mathbf{x}_0|\mathbf{x}_t = x_t] - \mathbb{E}[\mathbf{x}_0^G|\mathbf{x}_t^G = x_t]\| + \|\mathbb{E}[\mathbf{x}_0^G|\mathbf{x}_t^G = x_t]\|)$$

$$\lesssim \|x_t\|(M^2 - 1) + L(M^2 - 1 + \|x_t\|) = \|x_t\|(M^2 - 1 + L) + (M^2 - 1)L =: B_{x_t}$$

Furthermore, from equation 28 we see that

$$\|\nabla_{x_t} \mathbb{E}[\mathbf{x}_0|\mathbf{x}_t = x_t]\| \leq \|x_t\|(M^2 - 1) + L(M^2 - 1 + \|x_t\|) = B_{x_t}$$

too. We now subtract the full drifts in equation 20 and equation 21 and arrive at

$$\|\nabla \tilde{f}(x_t) - f_t(x_t)\|$$

$$\leq B_{x_t} \times \|T_1^{-1}(y - H\mathbb{E}[\mathbf{x}_0^G|\mathbf{x}_t^G = x_t])\|$$

$$\quad + B_{x_t} \|T_1^{-1} - T_2^{-1}\|_{\mathrm{op}} \|y - H\mathbb{E}[\mathbf{x}_0^G|\mathbf{x}_t^G = x_t]\|$$

$$\quad + B_{x_t} \|T_2^{-1}\|_{\mathrm{op}} \|H\mathbb{E}[\mathbf{x}_0^G|\mathbf{x}_t^G = z_t] - H\mathbb{E}[\mathbf{x}_0|\mathbf{x}_t = z_t]\|$$

$$\lesssim B_{x_t} + B_{x_t}(M^3 - 1) + B_{x_t}M(M^2 - 1)$$

$$\leq B_{x_t}(M^3 - 1) = (M^5 - 1)(\|x_t\| + L) + (M^3 - 1)L\|x_t\|.$$

Furthermore, we get that

$$\|\nabla \log p_t^G(x_t) - \nabla \log p_t(x_t)\| \le \|\nabla \log \frac{p_t^G(x_t)}{p_t(x_t)}\| \le \|\nabla_{x_t} \Phi_t(x_t)\| \le L.$$

Plugging in equation 20, we get that

$$\|\nabla \log p_t(x_t) + f_t(x_t) - p_t^{G,c}(x_t)\| \le (M^5 - 1)(\|x_t\| + L) + (M^3 - 1)L\|x_t\| + L.$$

Using this expression, we see that the drift change is linear in $\|x_t\|$. We now want to apply an iterated version of Novikov's condition to show that our application of Girsanov's Theorem is justified. To that end, we follow the argument Karatzas & Shreve (2012, Corollary 3.5.16). We repeat it here for completeness. We see that

$$\mathbb{E}_{\mathbb{P}^{G,c}} \left[ \exp \left( \frac{1}{2} \int_{t_i}^{t_{i+1}} \|\nabla \log p_t(x_t) + f_t(x_t) - \nabla \log p_t^{G,c}(x_t)\|^2 \mathrm{d}t \right) \right]$$

$$\le \mathbb{E}_{\mathbb{P}^{G,c}} \left[ \exp \left( c \int_{t_i}^{t_{i+1}} 1 + \|x_t\|^2 \mathrm{d}t \right) \right].$$

Since the expectation is regarding a Gaussian random variable $x_t$, we can make it finite as long as we pick $\Delta_i = t_{i+1} - t_i$ small enough. By setting $t_0 = 0$ and $t_1 > 0$, we can show equivalence on $[t_0, t_1]$. We can then iterate this procedure, to get equivalence on $[t_1, t_2]$ and so on. Since the lowest and highest eigenvalues of $\Sigma_t$ are bounded from below and above respectively, and $m_t$ is also bounded, the $\Delta_i$ can be bounded from below. Therefore, we get equivalence on $[0, T]$ this way in at most $\lceil T/\Delta \rceil$ steps.

Furthermore, we see that the likelihood KL-divergence can be bounded by

$$\mathbb{E}_{\mathbb{P}^{G,c}} \left[ \frac{1}{2} \int_0^T \|\nabla \log p_t(x_t) + f_t(x_t) - \nabla \log p_t^{G,c}(x_t)\|^2 \mathrm{d}t \right]$$

$$\lesssim T((M^5 - 1)(L+1) + (M^3 - 1)L + L) \le T((M^5 - 1)(L+1) + L),$$

where we used that the mean and covariance of the Gaussian process $x_t$ (under $\mathbb{P}^{G,c}$) can be bounded by a constant only depending on $y$, $\Sigma_t$ and $m_t$. Taking the square root proves our theorem. $\qquad \square$

## C   VARIANCE EXPLODING SDE (TIME RESCALED BROWNIAN MOTION)

The second main type of SDE used in denoising-diffusion is the Variance Exploding (Song et al., 2020) or time-rescaled Brownian motion, is

$$d\mathbf{x}_t = \sqrt{\frac{dv(t)}{dt}} d\mathbf{w}_t, \quad \mathbf{x}_0 \sim p_0 = p_{\text{data}} \tag{30}$$

where $(\mathbf{w}_t)_{t \in [0,T]}$ is a Brownian motion and $v : \mathbb{R}_{\ge 0} \to \mathbb{R}_{\ge 0}$ is an increasing function where $v(0) = 0$. The transition kernel of the time rescaled Brownian motion is

$$p_{t|0}(\mathbf{x}_t|\mathbf{x}_0) = \mathcal{N}(\mathbf{x}_0, v_t \mathbf{I}_{d_x}),$$

denoting $v(t) = v_t$. The signal $\mathbf{x}_0$ is sampled from a second SDE, the reverse process, given in forward time as

$$d\mathbf{x}_t = -\nabla_{\mathbf{x}_t} \log p_t(\mathbf{x}_t) dt - \sqrt{\frac{dv_t}{dt}} d\bar{\mathbf{w}}_t, \quad \mathbf{x}_T \sim p_T.$$

## D   TWEEDIE'S FORMULA

In this section, we give an alternative derivation of the Tweedie's formula for VPSDEs and VESDEs.

Tweedie's formula (Robbins, 1992; Efron, 2011) gives the minimum mean squared error (MMSE) estimator of $\mathbf{x}_0|\mathbf{x}_t$ when $\mathbf{x}_t|\mathbf{x}_0$ is Gaussian and the score $\nabla_{\mathbf{x}_t} \log p_t(\mathbf{x}_t)$ is available. Tweedie's formula applied to the time rescaled SDEs is given below.

### D.1 VARIANCE PRESERVING SDE (TIME RESCALED ORNSTEIN-UHLENBECK PROCESS)

Using the Variance Preserving SDE or time-rescaled Ornstein-Uhlenbeck process in equation 2, given the random variable

$$\sqrt{\frac{\alpha_t}{v_t}}\mathbf{x}_0 := \mathbf{v}_0 \sim p_{\mathbf{v}_0},$$

where, we now define $p_{\mathbf{v}_0}(\mathbf{v}_0)$ as the probability density of the random variable $\mathbf{v}_0$ evaluated at the realization $\mathbf{v}_0$, then the random variable

$$\frac{\mathbf{x}_t}{\sqrt{v_t}} := \mathbf{v}_t \sim \mathcal{N}(\mathbf{v}_0, \mathbf{I}_{d_x}),$$

has a marginal probability density evaluated at the realization $\mathbf{v}_t$ is the convolution of the random variable $\mathbf{v}_0$ with the Gaussian kernel

$$p_{\mathbf{v}_t}(\mathbf{v}_t) = \int \phi(\mathbf{v}_t - \mathbf{v}_0)p_{\mathbf{v}_0}(\mathbf{v}_0)d\mathbf{v}_0.$$

Then, Tweedie's formula (Robbins, 1992; Efron, 2011) gives

$$\mathbb{E}_{\mathbf{v}_0 \sim p_{\mathbf{v}_0|\mathbf{v}_t}}[\mathbf{v}_0] = \mathbf{v}_t + \nabla_{\mathbf{v}_t} \log p_{\mathbf{v}_t}(\mathbf{v}_t)$$

From continuous change of random variables, $p_{\mathbf{v}_t}(\mathbf{v}_t) = p_{\mathbf{x}_t}(\mathbf{v}_t\sqrt{v_t})\left|\frac{d\mathbf{x}_t}{d\mathbf{v}_t}\right| = p_{\mathbf{x}_t}(\mathbf{x}_t)\sqrt{v_t}$, giving $\log p_{\mathbf{v}_t}(\mathbf{v}_t) = \log p_{\mathbf{x}_t}(\mathbf{x}_t) + \log \sqrt{v_t}$, which yields

$$\nabla_{\mathbf{v}_t} \log p_{\mathbf{v}_t}(\mathbf{v}_t) = \sqrt{v_t}\nabla_{\mathbf{x}_t} \log p_{\mathbf{x}_t}(\mathbf{x}_t).$$

Now, substituting into Tweedie's formula,

$$\mathbb{E}_{\mathbf{v}_0 \sim p_{\mathbf{v}_0|\mathbf{v}_t}}[\mathbf{v}_0] = \frac{\mathbf{x}_t}{\sqrt{v_t}} + \sqrt{v_t}\nabla_{\mathbf{x}_t} \log p_{\mathbf{x}_t}(\mathbf{x}_t),$$

giving

$$\mathbb{E}_{\mathbf{x}_0 \sim p_{\mathbf{x}_0|\mathbf{x}_t}}[\mathbf{x}_0] = \frac{1}{\sqrt{\alpha_t}}\mathbf{x}_t + \frac{1}{\sqrt{\alpha_t}}v_t\nabla_{\mathbf{x}_t} \log p_{\mathbf{x}_t}(\mathbf{x}_t).$$

The covariance is given as (Robbins, 1992; Efron, 2011), $\mathbb{V}_{\mathbf{v}_0 \sim p_{\mathbf{v}_0|\mathbf{v}_t}}[\mathbf{v}_0|\mathbf{v}_t] = I + \nabla_{\mathbf{v}_t}\nabla_{\mathbf{v}_t} \log p_{\mathbf{v}_t}(\mathbf{v}_t)$, where

$$\nabla_{\mathbf{v}_t}\nabla_{\mathbf{v}_t} \log p_{\mathbf{v}_t}(\mathbf{v}_t) = \nabla_{\mathbf{v}_t}(\sqrt{v_t}\nabla_{\mathbf{x}_t} \log p_{\mathbf{x}_t}(\mathbf{x}_t)) \tag{31}$$

$$= v_t\nabla^2_{\mathbf{x}_t} \log p_{\mathbf{x}_t}(\mathbf{x}_t) \tag{32}$$

$$\tag{33}$$

and so

$$\mathbb{V}_{\mathbf{v}_0 \sim p_{\mathbf{v}_0|\mathbf{v}_t}}[\mathbf{v}_0] = \mathbf{I}_{d_x} + v_t\nabla^2_{\mathbf{x}_t} \log p_{\mathbf{x}_t}(\mathbf{x}_t)$$

giving

$$\mathbb{V}_{\mathbf{x}_0 \sim p_{\mathbf{x}_0|\mathbf{x}_t}}[\mathbf{x}_0] = \frac{v_t}{\alpha_t}(\mathbf{I}_{d_x} + v_t\nabla_{\mathbf{x}_t} \log p_{\mathbf{x}_t}(\mathbf{x}_t)).$$

In practice, we make the approximation $\nabla_{\mathbf{x}_t} \log p_{\mathbf{x}_t}(\mathbf{x}_t) \approx \mathsf{s}_\theta(\mathbf{x}_t, t)$, giving

$$\mathbb{E}_{\mathbf{x}_0 \sim p_{\mathbf{x}_0|\mathbf{x}_t}}[\mathbf{x}_0] \approx \mathbf{m}_{0|t} := \frac{1}{\sqrt{\alpha_t}}\mathbf{x}_t + \frac{1}{\sqrt{\alpha_t}}v_t\mathsf{s}_\theta(\mathbf{x}_t, t)$$

and

$$\mathbb{V}_{\mathbf{x}_0 \sim p_{\mathbf{x}_0|\mathbf{x}_t}}[\mathbf{x}_0] \approx \frac{v_t}{\alpha_t}(\mathbf{I}_{d_x} + v_t\nabla_{\mathbf{x}_t}\mathsf{s}_\theta(\mathbf{x}_t, t)),$$

which yields a Gaussian approximation that matches the first and second moments,

$$p_{\mathbf{x}_0|\mathbf{x}_t}(\mathbf{x}_0|\mathbf{x}_t) \approx \mathcal{N}\left(\mathbf{m}_{0|t}, \frac{v_t}{\alpha_t}(\mathbf{I}_{d_x} + v_t\nabla_{\mathbf{x}_t}\mathsf{s}_\theta(\mathbf{x}_t, t))\right) \tag{34}$$

$$= \mathcal{N}\left(\mathbf{m}_{0|t}, \frac{v_t}{\sqrt{\alpha_t}}\nabla_{\mathbf{x}_t}\mathbf{m}_{0|t}\right). \tag{35}$$

The matrix $\frac{v_t}{\alpha_t}(\mathbf{I}_{d_x} + v_t\nabla_{\mathbf{x}_t}\mathsf{s}_\theta(\mathbf{x}_t, t))$ needs to be inverted to calculate the log likelihood, and therefore must be both symmetric and positive definite for all time, which puts the requirement that $\mathsf{s}_\theta(\mathbf{x}_t, t)$ can be written as the negative gradient of a potential, however it is only approximated as such $\mathsf{s}_\theta(\mathbf{x}_t, t) \approx \nabla \log p_{\mathbf{x}_t}(\mathbf{x}_t)$.

## D.2 VARIANCE EXPLODING SDE (TIME RESCALED BROWNIAN MOTION)

Using the Variance Exploding SDE (Song et al., 2020) or time rescaled Brownian motion, given the random variable

$$\sqrt{\frac{1}{v_t}}\mathbf{x}_0 := \mathbf{v}_0 \sim p_{\mathbf{z}_0}, \tag{36}$$

a similar calculation as in D.1 gives

$$\mathbb{E}_{\mathbf{x}_0 \sim p_{\mathbf{x}_0|\mathbf{x}_t}}[\mathbf{x}_0] = \mathbf{x}_t + v_t \nabla_{\mathbf{x}_t} \log p_{\mathbf{x}_t}(\mathbf{x}_t),$$

and

$$\mathbb{V}_{\mathbf{x}_0 \sim p_{\mathbf{x}_0|\mathbf{x}_t}}[\mathbf{x}_0] = v_t(\mathbf{I}_{d_x} + v_t \nabla_{\mathbf{x}_t} \log p_{\mathbf{x}_t}(\mathbf{x}_t)).$$

Again, in practice, we make the approximation $\nabla_{\mathbf{x}_t} \log p_{\mathbf{x}_t}(\mathbf{x}_t) = \mathsf{s}_\theta(\mathbf{x}_t, t)$, giving

$$\mathbb{E}_{\mathbf{x}_0 \sim p_{\mathbf{x}_0|\mathbf{x}_t}}[\mathbf{x}_0] \approx \mathbf{m}_{0|t} := \mathbf{x}_t + v_t \mathsf{s}_\theta(\mathbf{x}_t, t)$$

and

$$\mathbb{V}_{\mathbf{x}_0 \sim p_{\mathbf{x}_0|\mathbf{x}_t}}[\mathbf{x}_0] \approx v_t(\mathbf{I}_{d_x} + v_t \nabla_{\mathbf{x}_t} \mathsf{s}_\theta(\mathbf{x}_t, t)),$$

which yields a Gaussian approximation that matches the first and second moments,

$$p_{\mathbf{x}_0|\mathbf{x}_t}(\mathbf{x}_0|\mathbf{x}_t) \approx \mathcal{N}\left(\mathbf{m}_{0|t}, v_t(\mathbf{I}_{d_x} + v_t \nabla_{\mathbf{x}_t} \mathsf{s}_\theta(\mathbf{x}_t, t))\right) \tag{37}$$

$$= \mathcal{N}\left(\mathbf{m}_{0|t}, v_t \nabla_{\mathbf{x}_t} \mathbf{m}_{0|t}\right). \tag{38}$$

## E ALGORITHMIC DETAILS AND NUMERICS

The code for all of the experiments and instructions to run them are available at X.

---

**Algorithm 1** TMPD-D (Ancestral sampling, VP)

---

$\mathbf{x}_N \sim \mathcal{N}(0, \mathbf{I}_{d_\mathbf{x}})$
**for** $n = N-1, \ldots, 0$ **do**
  $\mathbf{m}_{0|t} = \frac{1}{\sqrt{\alpha_n}}(\mathbf{x}_n + v_n \mathsf{s}_\theta(\hat{\mathbf{x}}_n, t_n))$
  $\mathbf{m}_{0|t}^{\mathbf{y}} = \mathbf{m}_{0|t} + \frac{v_n}{\sqrt{\alpha_n}}\nabla_{\mathbf{x}_n}\mathbf{m}_{0|t}\mathbf{H}^\top(\mathbf{H}\frac{v_n}{\sqrt{\alpha_n}}\nabla_{\mathbf{x}_n}\mathbf{m}_{0|t}\mathbf{H}^\top + \sigma_y^2\mathbf{I}_{d_\mathbf{y}})^{-1}(\mathbf{y} - \mathbf{H}\mathbf{m}_{0|t})$
  $\mathbf{z}_n \sim \mathcal{N}(0, \mathbf{I}_{d_\mathbf{x}})$
  $\sigma_n = \sqrt{(1 - \alpha_{n-1})\beta_n/(1 - \alpha_n)}$
  $\mathbf{x}_{n-1} \leftarrow \frac{\sqrt{1-\beta_n}(1-\alpha_{n-1})}{1-\alpha_n}\mathbf{x}_n + \frac{\sqrt{\alpha_{n-1}}\beta_n}{1-\alpha_n}\mathbf{m}_{0|t}^{\mathbf{y}} + \sigma_n\mathbf{z}_n$
**end for**
**return** $\hat{\mathbf{x}}_0$

---

### E.1 COMPUTATIONAL COMPLEXITY

Let $N$ is the number of noise scales, and let $d_\mathbf{y}$ is the dimensions of the observation, $d_\mathbf{x}$ are the dimension of the image and $T_s$ is the time complexity of evaluating the score network. Computing a jacobian-vector product has time complexity $T_s$. Then, the time complexity of TMPD (not including fast matrix-vector products) is $\mathcal{O}(N(d_\mathbf{y}^3 + T_s d_\mathbf{y} + T_s + T_s))$. ΠGDM comes at a smaller computational cost due to needing only $\mathcal{O}(1)$ vector-jacobian-products, instead of $\mathcal{O}(d_\mathbf{y})$, resulting in a time complexity of $\mathcal{O}(N(T_s + T_s))$. DPS comes at the same complexity as ΠGDM.

Whereas TMPD requires calculating the Jacobian which has memory complexity of atleast $\mathcal{O}(d_\mathbf{x}d_\mathbf{y})$. This is too large for high resolution image problems where the dimension of the observation is large. In comparison, the memory complexity of ΠGDM depends on the observation operator, but for the class of problems that are explored in Song et al. (2023) is $\mathcal{O}(d_\mathbf{x})$. DPS comes at the same memory complexity as ΠGDM.

As mentioned in the text, we make the following approximation

DTMPD   $\nabla_{\mathbf{x}_t} \log p_{\mathbf{y}|t}(\mathbf{y}|\mathbf{x}_t) \approx \nabla_{\mathbf{x}_t}\mathbf{m}_{0|t}\mathbf{H}^\top(\mathbf{H}\frac{v_t}{\sqrt{\alpha_t}}\mathrm{diag}(\nabla_{\mathbf{x}_t}\mathbf{m}_{0|t})\mathbf{H}^\top + \sigma_\mathbf{y}^2\mathbf{I}_{d_\mathbf{y}})^{-1}(\mathbf{y} - \mathbf{H}\mathbf{m}_{0|t}).$

Whilst this approximation does not require a linear solve, taking out the $\mathcal{O}(Nd_\mathbf{y}^3)$ time complexity term, we would still like to take out the $\mathcal{O}(NT_s d_\mathrm{y})$ time complexity and $\mathcal{O}(d_\mathrm{x} d_\mathrm{y})$ memory complexity term from calculating and storing the Jacobian, respectively, since this is too large for solving high resolution image applications. A further approximation approximates the diagonal of the Jacobian by the row sum of the Jacobian which only requires one vector jacobian product and brings the memory and time complexity of DTMPD down to that of $\Pi$GDM. We use this approximation in the image experiments and find that in practice it is only $(1.5 \pm 0.1)\times$ slower than $\Pi$GDM and DPS across all of our experiments. The row sum will be a good approximation of the diagonal when the Jacobian is approximately diagonal, which happens when there is small linear correlation between observation pixels, which we found to work well for super-resolution and inpainting. A fruitful direction of research would be to explore low rank approximations of the Jacobian matrix.

### E.2 GAUSSIAN

When the data distribution $p_0(\mathbf{x}_0)$ is a (multivariate) Gaussian, then the reverse SDE is a linear SDE and we can calculate all of the terms needed to sample from the target posterior using diffusion explicitly. Moreover, we can sample from the target posterior using a direct or implicit method such as Cholesky decomposition. In this simple example, we compare samples from the direct method to various conditional diffusion methods (TMPD, $\Pi$GDM, DPM), by plotting a sample estimate of the $L^2$ Wasserstein distance between the sample and the target Gaussian measures (Givens & Shortt, 1984) by using the analytical mean and covariance of the target distribution and empirical estimate of the mean and covariance of the sample distribution. To generate $p_0(\mathbf{x}_0)$, we use an equally spaced grid of vectors $\mathbf{u}_i \in [-5.0, 5.0]^2$ for $i \in \{1, 2, ..., 32\}^2$, pick a Matern 5/2 kernel for the covariance function $k(\mathbf{u}_i, \mathbf{u}_j) = \left(1 + \sqrt{5}|\mathbf{u}_i - \mathbf{u}_j| + \frac{5}{3}|\mathbf{u}_i - \mathbf{u}_j|^2\right) \exp\left(-\sqrt{3}|\mathbf{u}_i - \mathbf{u}_j|\right)$ which defines the prior $p_0(\mathbf{x}_0) = \mathcal{N}(\mathbf{m}_0, \mathbf{C}_0)$ covariance $\mathbf{C}_0 \in \mathbb{R}^{1024 \times 1024}$ where $\mathbf{C}_{0ij} = k(\mathbf{u}_i, \mathbf{u}_j)$ and we define the mean as a zero vector $\mathbf{m}_0 = \mathbf{0} \in \mathbb{R}^{1024}$. To compute analytically the distribution of $p_{0|\mathbf{y}}(\mathbf{x}_0|\mathbf{y})$, we sample $\mathbf{y} = \mathbf{H}\mathbf{x}_0 + \mathbf{z}, \mathbf{z} \sim \mathcal{N}(\mathbf{0}, \sigma_\mathrm{y}^2 \mathbf{I}_{d_y})$ and use the standard Gaussian formula to calculate the mean and covariance of $\mathbf{x}_0|\mathbf{y}$ which in this case are a complete description of $p_{0|\mathbf{y}}(\mathbf{x}_0|\mathbf{y})$. The $L^2$ Wasserstein estimate is plotted over an increasing sample size $N \in [9, 1500]$ and for $\sigma_\mathrm{y} = 0.1$ in Figure 1.

We provide an illustration of the mean and uncertainty captured by the diffusion samples using 1500 samples of each diffusion model to produce a Monte-Carlo estimate of the mean and diagonal variance vector, and compare these to the exact mean and diagonal variance.

Below, we provide a calculation comparing the exact diffusion posterior for the linear diffusion posterior sde to the approximations used in $\Pi$GDM (Song et al., 2023) and TMPD. Let us assume that the target distribution be known $p_0(\mathbf{x}_0) = \mathcal{N}(\mathbf{m}_0, \mathbf{C}_0)$. Then, via Bayes' rule,

$$
\begin{aligned}
\log p_{0|t}(\mathbf{x}_0|\mathbf{x}_t) &= \log p_{t|0}(\mathbf{x}_t|\mathbf{x}_0) + \log p_0(\mathbf{x}_0) + \text{constant} \\
&= -\frac{1}{2}(\mathbf{x}_t - \sqrt{\alpha_t}\mathbf{x}_0)^\top (v_t \mathbf{I}_{d_\mathrm{x}})^{-1}(\mathbf{x}_t - \sqrt{\alpha_t}\mathbf{x}_0) \\
&\quad -\frac{1}{2}(\mathbf{x}_0 - \mathbf{m}_0)^\top \mathbf{C}_0^{-1}(\mathbf{x}_0 - \mathbf{m}_0) + \text{constant} \\
&= -\frac{1}{2}(\sqrt{\alpha_t}\mathbf{x}_0)^\top (v_t \mathbf{I}_{d_\mathrm{x}})^{-1}(\sqrt{\alpha_t}\mathbf{x}_0) + \mathbf{x}_t^\top (v_t \mathbf{I}_{d_\mathrm{x}})^{-1}(\sqrt{\alpha_t}\mathbf{x}_0) \\
&\quad -\frac{1}{2}\mathbf{x}_0^\top \mathbf{C}_0^{-1}\mathbf{x}_0 + \mathbf{m}_0^\top \mathbf{C}_0^{-1}\mathbf{x}_0 + \text{constant} \\
&= -\frac{1}{2}(\mathbf{x}_0 - \mathbf{m}_t)^\top \mathbf{\Sigma}_t^{-1}(\mathbf{x}_0 - \mathbf{m}_t),
\end{aligned}
$$

so

$$
p_{0|t}(\mathbf{x}_0|\mathbf{x}_t) = \mathcal{N}(\mathbf{m}_t, \mathbf{\Sigma}_t) \tag{39}
$$

where

$$
\mathbf{\Sigma}_t = \left( (\frac{v_t}{\alpha_t} \mathbf{I}_{d_\mathrm{x}})^{-1} + \mathbf{C}_0^{-1} \right)^{-1} \tag{40}
$$

and

$$
\mathbf{m}_t = \mathbf{\Sigma}_t \left( \frac{\sqrt{\alpha_t}}{v_t} \mathbf{x}_t + \mathbf{C}_0^{-1} \mathbf{m}_0 \right).
$$

We also have that,

$$p_t(\mathbf{x}_t) = \mathcal{N}(\sqrt{\alpha_t}\mathbf{m}_0, \mathbf{C}_t)$$

where $\mathbf{C}_t = \alpha_t\mathbf{C}_0 + v_t\mathbf{I}_{d_x}$, and thus

$$\log p_t(\mathbf{x}_t) = -\frac{1}{2}(\mathbf{x}_t - \mathbf{m}_0\sqrt{\alpha_t})^\top \mathbf{C}_t^{-1}(\mathbf{x}_t - \mathbf{m}_0\sqrt{\alpha_t}).$$

From

$$p_{\mathbf{y}|t}(\mathbf{y}|\mathbf{x}_t) = \int p_{\mathbf{y}|0}(\mathbf{y}|\mathbf{x}_0)p_{0|t}(\mathbf{x}_0|\mathbf{x}_t)d\mathbf{x}_0 = \mathcal{N}(\mathbf{H}\mathbf{m}_t, \mathbf{H}\mathbf{\Sigma}_t\mathbf{H}^\top + \sigma_{\mathbf{y}}^2\mathbf{I}_{d_y}),$$

we have that

$$\log p_t(\mathbf{y}|\mathbf{x}_t) = -\frac{1}{2}(\mathbf{y} - \mathbf{H}\mathbf{m}_t)^\top (\mathbf{H}\mathbf{\Sigma}_t\mathbf{H}^T + \sigma_{\mathbf{y}}^2\mathbf{I}_{d_y})^{-1}(\mathbf{y} - \mathbf{H}\mathbf{m}_t).$$

The posterior score can be calculated (see Cardoso et al. (2023)) from applying Bayes' theorem,

$$\log p_{t|\mathbf{y}}(\mathbf{x}_t|\mathbf{y}) = \log p_t(\mathbf{x}_t) + \log p_{\mathbf{y}|t}(\mathbf{y}|\mathbf{x}_t) + \text{Constant},$$

and taking gradients with respect to the state $\mathbf{x}_t$ gives

$$\nabla_{\mathbf{x}_t} \log p_{t|y}(\mathbf{x}_t|\mathbf{y}) = \nabla_{\mathbf{x}_t} \log p_t(\mathbf{x}_t) + \nabla_{\mathbf{x}_t} \log p_{\mathbf{y}|t}(\mathbf{y}|\mathbf{x}_t) \tag{41}$$

$$= -(\alpha_t\mathbf{C}_0 + v_t\mathbf{I}_{d_x})^{-1}(\mathbf{x}_t - \mathbf{m}_0\sqrt{\alpha_t}) \tag{42}$$

$$+ \frac{\sqrt{\alpha_t}}{v_t}\mathbf{\Sigma}_t^\top \mathbf{H}^\top (\mathbf{H}\mathbf{\Sigma}_t\mathbf{H}^\top + \sigma_{\mathbf{y}}^2\mathbf{I}_{d_y})^{-1}(\mathbf{y} - \mathbf{H}\mathbf{\Sigma}_t(\frac{\sqrt{\alpha_t}}{v_t}\mathbf{x}_t + \mathbf{C}_0^{-1}\mathbf{m}_0)). \tag{43}$$

Setting $\mathbf{m}_0 = \mathbf{0}$ in (43) gives,

$$\nabla_{\mathbf{x}_t} \log p_{t|\mathbf{y}}(\mathbf{x}_t|\mathbf{y}) = -(\alpha_t\mathbf{C}_0 + v_t\mathbf{I}_{d_x})^{-1}\mathbf{x}_t \tag{44}$$

$$+ \frac{\sqrt{\alpha_t}}{v_t}\mathbf{\Sigma}_t^\top \mathbf{H}^\top (\mathbf{H}\mathbf{\Sigma}_t\mathbf{H}^\top + \sigma_{\mathbf{y}}^2\mathbf{I}_{d_y})^{-1}(\mathbf{y} - \frac{\sqrt{\alpha_t}}{v_t}\mathbf{H}\mathbf{\Sigma}_t\mathbf{x}_t) \tag{45}$$

### E.2.1 COMPARISON OF THE EXACT POSTERIOR TO THE APPROXIMATION MADE IN PSEUDO-INVERSE-GUIDANCE

We aim to compare (45), to the approximation used in $\Pi$GDM (Song et al., 2023);

$$p_{0|t}(\mathbf{x}_0|\mathbf{x}_t) \approx \mathcal{N}(\mathbf{m}_{0|t}, r_t^2\mathbf{I}_{d_x}),$$

where $\mathbf{m}_{0|t} = \mathbf{x}_t + v_t\nabla_{\mathbf{x}_t}\log p_t(\mathbf{x}_t)$, which is the MMSE estimate of $\mathbf{x}_0|\mathbf{x}_t$ and $r_t$ is chosen empirically. This gives $p_{\mathbf{y}|t}(\mathbf{y}|\mathbf{x}_t) \approx \mathcal{N}(\mathbf{H}\mathbf{m}_{0|t}, r_t^2\mathbf{H}\mathbf{H}^T + \sigma_{\mathbf{y}}^2\mathbf{I}_{d_y})$, which in turn gives

$$\nabla_{\mathbf{x}_t} \log p_{t|y}(\mathbf{x}_t|\mathbf{y}) \approx \nabla_{\mathbf{x}_t} \log p_t(\mathbf{x}_t) \tag{46}$$

$$+ \nabla_{\mathbf{x}_t}\mathbf{m}_{0|t}\mathbf{H}^\top (r_t^2\mathbf{H}\mathbf{H}^\top + \sigma_{\mathbf{y}}^2\mathbf{I}_{d_y})^{-1}(\mathbf{y} - \mathbf{H}\mathbf{m}_{0|t}) \tag{47}$$

which is computationally tractable in the standard diffusion model setting where the score is non-linear and approximated as $\nabla_{\mathbf{x}_t} \log p_t(\mathbf{x}_t) \approx \mathbf{s}_\theta(\mathbf{x}_t, t)$. Substituting in the known score and again setting $\mathbf{m}_0 = \mathbf{0}$ score to compare to the linear case (45) gives,

$$\nabla_{\mathbf{x}_t} \log p_{t|\mathbf{y}}(\mathbf{x}_t|\mathbf{y}) \approx \nabla_{\mathbf{x}_t} \log p_t(\mathbf{x}_t) \tag{48}$$

$$+ \nabla_{\mathbf{x}_t}\mathbf{m}_{0|t}\mathbf{H}^\top (r_t^2\mathbf{H}\mathbf{H}^\top + \sigma_{\mathbf{y}}^2I_{d_y})^{-1}(y - \mathbf{H}\mathbf{m}_{0|t}) \tag{49}$$

$$= -(\alpha_t\mathbf{C}_0 + v_t\mathbf{I}_{d_x})^{-1}\mathbf{x}_t \tag{50}$$

$$+ \frac{1}{\sqrt{\alpha_t}}(\mathbf{I}_{d_x} - v_t(\alpha_t\mathbf{C}_0 + v_t\mathbf{I}_{d_x})^{-1})\mathbf{H}^\top (r_t^2\mathbf{H}\mathbf{H}^T + \sigma_{\mathbf{y}}^2\mathbf{I}_{d_y})^{-1} \tag{51}$$

$$(\mathbf{y} - \frac{1}{\sqrt{\alpha_t}}\mathbf{H}(\mathbf{I}_{d_x} - v_t(\alpha_t\mathbf{C}_0 + v_t\mathbf{I}_{d_x})^{-1})\mathbf{x}_t) \tag{52}$$

comparing terms, $\Pi$GDM is making the approximation

$$\nabla_{\mathbf{x}_t}\mathbf{m}_{0|t} = \frac{1}{\sqrt{\alpha_t}}\mathbf{I}_{d_x} - \frac{v_t}{\sqrt{\alpha_t}}(\alpha_t\mathbf{C}_0 + v_t\mathbf{I}_{d_x})^{-1} \approx \frac{\sqrt{\alpha_t}}{v_t}\mathbf{\Sigma}_t.$$

Note that since,

$$\frac{\sqrt{\alpha_t}}{v_t}\boldsymbol{\Sigma}_t = \frac{\sqrt{\alpha_t}}{v_t}(\mathbf{C}_0^{-1} + (\frac{v_t}{\alpha_t}I_{d_x})^{-1})^{-1} \tag{53}$$

$$= \frac{1}{v_t\sqrt{\alpha_t}}((\alpha_t\mathbf{C}_0)^{-1} + (v_t\mathbf{I}_{d_x})^{-1})^{-1} \tag{54}$$

$$= \frac{1}{v_t\sqrt{\alpha_t}}(v_t\mathbf{I} - v_t^2(\alpha_t\mathbf{C}_0 + v_t\mathbf{I}_{d_x})^{-1}) \quad \text{Woodbury identity} \tag{55}$$

$$= \frac{1}{\sqrt{\alpha_t}}(\mathbf{I} - v_t(\alpha_t\mathbf{C}_0 + v_t\mathbf{I}_{d_x})^{-1}) \tag{56}$$

$$= \nabla_{\mathbf{x}_t}\mathbf{m}_{0|t}, \tag{57}$$

this is exact. The other approximation ΠGDM is making is $r_t^2\mathbf{I} \approx \boldsymbol{\Sigma}_t$, and note that this approximation is accurate with $r_t^2 = v_t$ as $t \to 0$ but inaccurate as $t \to 1$. But the exact linear SDE can be recovered by instead using TMPD,

$$\nabla_{\mathbf{x}_t}\log p_{t|y}(\mathbf{x}_t|\mathbf{y}) = \nabla_{\mathbf{x}_t}\log p_t(\mathbf{x}_t) \tag{58}$$

$$+ \nabla_{\mathbf{x}_t}\mathbf{m}_{0|t}\mathbf{H}^\top(\mathbf{H}\frac{v_t}{\sqrt{\alpha_t}}\nabla_{\mathbf{x}_t}\mathbf{m}_{0|t}\mathbf{H}^\top + \sigma_y^2 I_{d_y})^{-1}(\mathbf{y} - \mathbf{H}\mathbf{m}_{0|t}) \tag{59}$$

$$= \nabla_{\mathbf{x}_t}\log p_t(\mathbf{x}_t) \tag{60}$$

$$+ \frac{\sqrt{\alpha_t}}{v_t}\boldsymbol{\Sigma}_t\mathbf{H}^\top(\mathbf{H}\boldsymbol{\Sigma}_t\mathbf{H}^\top + \sigma_y^2 I_{d_y})^{-1}(\mathbf{y} - \frac{\sqrt{\alpha_t}}{v_t}\mathbf{H}\boldsymbol{\Sigma}_t\mathbf{x}_t) \tag{61}$$

### E.3   GMM

For a given dimension $d_x$, we consider $p_0$ a mixture of 25 Gaussian random variables. The components have mean $\mu_{i,j} := (8i, 8j, ..., 8i, 8j) \in \mathbb{R}^{d_x}$ for $(i,j) \in \{-2, -1, 0, 1, 2\}^2$ and unit variance. We have set the associated unnormalized weights $\omega_{i,j} = 1.0$. We have set $\sigma_\delta^2 = 10^{-4}$.

Note that $p_t(\mathbf{x}_t) = \int p_{t|0}(\mathbf{x}_t|\mathbf{x}_0)p_0(\mathbf{x}_0)d\mathbf{x}_0$. As $p_0(\mathbf{x}_0)$ is a mixture of Gaussians, $p_t(\mathbf{x}_t)$ is also a mixture of Gaussians with means $\sqrt{\alpha_t}\mu_{i,j}$ and unitary variances. Therefore, using automatic differentiation libraries, we can calculate $\nabla_{\mathbf{x}_t}\log p_t(\mathbf{x}_t)$. We chose $\beta_{\max} = 500.0$ and $\beta_{\min} = 0.1$. We use 1000 timesteps for the time-discretization. For the pair of dimensions and chosen observation noise standard deviation $(d_x, d_y, \sigma_y)$ the measurement model $(y, \mathbf{H})$ is drawn as follows:

- **H**: We first draw $\tilde{\mathbf{H}} \sim \mathcal{N}(\mathbf{0}_{d_y \times d_x}, \mathbf{I}_{d_y \times d_x})$ and compute the SVD decomposition of $\tilde{\mathbf{H}} = \boldsymbol{U}\boldsymbol{S}\boldsymbol{V}^\top$. Then, we sample for $(i,j) \in \{-2, -1, 0, 1, 2\}^2$, $s_{i,j}$ according to a uniform in $[0, 1]$. Finally, we set $\mathbf{H} = \boldsymbol{U}\text{diag}(\{s_{i,j}\}_{(i,j)\in\{-2,-1,0,1,2\}^2})\boldsymbol{V}^\top$.

- **y**: We then draw $\mathbf{x}_* \sim p_0$ and set $\mathbf{y} := \mathbf{H}x_* + \mathbf{z}$ where $\mathbf{z} \sim \mathcal{N}(\mathbf{0}, \sigma_y^2\mathbf{I}_{d_y})$.

Once we have drawn both $\mathbf{x}_* \sim p_0$ and $(\mathbf{y}, \mathbf{H}, \sigma_y)$, the posterior can be exactly calculated using Bayes formula and gives a mixture of Gaussians with mixture components $c_{i,j}$ and associated weights $\tilde{\omega}_{i,j}$,

$$c_{i,j} := \mathcal{N}(\boldsymbol{\Sigma}(\mathbf{H}^\top\mathbf{y}/\sigma_y^2 + \mu_{i,j}), \boldsymbol{\Sigma}), \tag{62}$$

$$\tilde{\omega}_i := \omega_i\mathcal{N}(\mathbf{y}; \mathbf{H}\mu_{i,j}, \sigma_\delta^2\mathbf{I}_{d_x} + \mathbf{H}\mathbf{H}^\top), \tag{63}$$

where $\boldsymbol{\Sigma} = (\mathbf{I}_{d_x} + \frac{1}{\sigma_\delta^2}\mathbf{H}^\top\mathbf{H})^{-1}$.

#### E.3.1   EULER-MARUYAMA SOLVER

To compare the posterior distribution estimated by each algorithm with the target posterior distribution, we use $10^4$ slices for the SW distance and compare 1000 samples of the continuous SDEs defined by the TMPD, DTMPD, Song et al. (2023) and Chung et al. (2022a) approximations obtained using 1000 Euler-Maruyama time-steps with 1000 samples of the true posterior distribution.

Table 4 indicate the Central Limit Theorem (CLT) 95% confidence intervals obtained by considering 20 randomly selected measurement models (**H**) for each setting $(d_x, d_x, \sigma_y)$. Figure 2 shows the first

Table 3: Sliced Wasserstein for the GMM case for VE DDPM.

| | | $\sigma_{\mathbf{y}} = 0.01$ | | | | $\sigma_{\mathbf{y}} = 0.1$ | | | | $\sigma_{\mathbf{y}} = 1.0$ | | | |
|---|---|---|---|---|---|---|---|---|---|---|---|---|---|
| $d_x$ | $d_y$ | TMPD-D | DTMPD-D | ΠGDM-D | DPS-D | TMPD-D | DTMPD-D | ΠGDM-D | DPS-D | TMPD-D | DTMPD-D | ΠGDM-D | DPS-D |
| 8 | 1 | $1.6 \pm 0.5$ | $1.8 \pm 0.6$ | $2.6 \pm 0.9$ | $4.7 \pm 1.5$ | $\mathbf{1.4 \pm 0.5}$ | $1.8 \pm 0.7$ | $2.2 \pm 0.9$ | $4.7 \pm 1.6$ | $\mathbf{0.9 \pm 0.3}$ | $\mathbf{0.9 \pm 0.2}$ | $1.5 \pm 0.4$ | $5.2 \pm 1.3$ |
| 8 | 2 | $\mathbf{0.7 \pm 0.3}$ | $3.3 \pm 1.5$ | $2.1 \pm 1.0$ | $1.8 \pm 1.5$ | $\mathbf{0.9 \pm 0.3}$ | $2.7 \pm 1.1$ | $1.6 \pm 0.6$ | $1.5 \pm 0.9$ | $\mathbf{0.9 \pm 0.2}$ | $1.7 \pm 0.8$ | $2.3 \pm 0.4$ | $3.5 \pm 1.2$ |
| 8 | 4 | $\mathbf{0.3 \pm 0.3}$ | $0.4 \pm 0.2$ | $3.8 \pm 2.3$ | $0.7 \pm 0.6$ | $\mathbf{0.3 \pm 0.2}$ | $0.5 \pm 0.2$ | $3.8 \pm 2.2$ | $0.8 \pm 0.6$ | $\mathbf{0.6 \pm 0.2}$ | $0.9 \pm 0.5$ | $1.8 \pm 0.3$ | $2.5 \pm 0.9$ |
| 80 | 1 | $2.7 \pm 0.7$ | $2.8 \pm 0.9$ | $3.2 \pm 1.0$ | $3.2 \pm 1.9$ | $\mathbf{2.3 \pm 0.7}$ | $2.6 \pm 0.9$ | $2.9 \pm 0.8$ | $5.1 \pm 1.8$ | $1.5 \pm 0.7$ | $1.4 \pm 0.6$ | $1.6 \pm 0.5$ | $6.9 \pm 1.8$ |
| 80 | 2 | $\mathbf{1.0 \pm 0.5}$ | $3.2 \pm 1.1$ | $2.8 \pm 1.3$ | $3.2 \pm 1.9$ | $\mathbf{1.2 \pm 0.5}$ | $3.2 \pm 1.1$ | $2.7 \pm 1.2$ | $3.1 \pm 1.9$ | $\mathbf{1.1 \pm 0.2}$ | $2.1 \pm 1.0$ | $1.4 \pm 0.2$ | $3.9 \pm 1.2$ |
| 80 | 4 | $\mathbf{0.3 \pm 0.1}$ | $0.7 \pm 0.4$ | $0.6 \pm 0.4$ | $1.2 \pm 1.1$ | $\mathbf{0.4 \pm 0.2}$ | $0.8 \pm 0.4$ | $0.6 \pm 0.4$ | $1.0 \pm 1.1$ | $0.9 \pm 0.3$ | $0.9 \pm 0.4$ | $0.9 \pm 0.2$ | $1.7 \pm 0.6$ |
| 800 | 1 | $3.1 \pm 0.7$ | $3.7 \pm 0.7$ | $3.5 \pm 1.1$ | $5.8 \pm 1.6$ | $\mathbf{2.9 \pm 0.6}$ | $3.4 \pm 0.7$ | $3.3 \pm 0.9$ | $5.7 \pm 1.6$ | $\mathbf{1.5 \pm 0.4}$ | $1.4 \pm 0.4$ | $2.0 \pm 0.4$ | $6.8 \pm 1.0$ |
| 800 | 2 | $\mathbf{1.4 \pm 0.4}$ | $3.5 \pm 0.7$ | $3.1 \pm 1.1$ | $3.5 \pm 1.7$ | $\mathbf{1.3 \pm 0.3}$ | $3.4 \pm 0.7$ | $2.7 \pm 0.9$ | $3.1 \pm 1.4$ | $\mathbf{1.2 \pm 0.3}$ | $2.0 \pm 0.4$ | $2.0 \pm 0.5$ | $4.7 \pm 1.3$ |
| 800 | 4 | $\mathbf{0.4 \pm 0.2}$ | $0.7 \pm 0.5$ | $\mathbf{0.4 \pm 0.2}$ | $1.4 \pm 1.0$ | $\mathbf{0.4 \pm 0.2}$ | $0.8 \pm 0.5$ | $\mathbf{0.4 \pm 0.2}$ | $1.3 \pm 0.9$ | $0.9 \pm 0.2$ | $1.1 \pm 0.5$ | $\mathbf{0.6 \pm 0.2}$ | $0.9 \pm 0.4$ |

Table 4: Sliced Wasserstein for the GMM case for the reverse VE SDEs discretized with Euler-Maruyama.

| | | $\sigma_{\mathbf{y}} = 0.01$ | | | | $\sigma_{\mathbf{y}} = 0.1$ | | | | $\sigma_{\mathbf{y}} = 1.0$ | | | |
|---|---|---|---|---|---|---|---|---|---|---|---|---|---|
| $d_x$ | $d_y$ | TMPD | DTMPD | ΠGDM | DPS | TMPD | DTMPD | ΠGDM | DPS | TMPD | DTMPD | ΠGDM | DPS |
| 8 | 1 | $1.5 \pm 0.5$ | $1.5 \pm 0.5$ | $1.5 \pm 0.4$ | $5.7 \pm 2.2$ | $1.4 \pm 0.5$ | $1.4 \pm 0.5$ | $1.2 \pm 0.4$ | $5.6 \pm 2.1$ | $0.9 \pm 0.3$ | $0.9 \pm 0.3$ | $0.9 \pm 0.3$ | $0.9 \pm 0.3$ |
| 8 | 2 | $0.7 \pm 0.3$ | $3.2 \pm 1.4$ | $0.4 \pm 0.3$ | $6.2 \pm 0.8$ | $0.9 \pm 0.3$ | $2.7 \pm 1.1$ | $0.5 \pm 0.3$ | $6.2 \pm 2.4$ | $0.9 \pm 0.2$ | $1.8 \pm 0.8$ | $1.0 \pm 0.3$ | $1.2 \pm 0.4$ |
| 8 | 4 | $0.3 \pm 0.3$ | $0.6 \pm 0.4$ | $0.1 \pm 0.1$ | - | $0.3 \pm 0.2$ | $0.7 \pm 0.4$ | $0.1 \pm 0.0$ | $8.4 \pm 3.1$ | $0.6 \pm 0.2$ | $0.9 \pm 0.5$ | $0.2 \pm 0.1$ | $0.3 \pm 0.2$ |
| 80 | 1 | $2.7 \pm 0.7$ | $2.7 \pm 0.7$ | $2.9 \pm 1.4$ | $9.1 \pm 1.3$ | $2.3 \pm 0.7$ | $2.3 \pm 0.7$ | $2.1 \pm 1.1$ | $4.7 \pm 1.8$ | $1.5 \pm 0.7$ | $1.5 \pm 0.7$ | $1.8 \pm 0.8$ | $1.9 \pm 0.9$ |
| 80 | 2 | $1.0 \pm 0.5$ | $3.3 \pm 1.0$ | $0.8 \pm 0.7$ | $2.2 \pm 0.9$ | $1.2 \pm 0.5$ | $3.3 \pm 1.0$ | $0.8 \pm 0.7$ | $6.0 \pm 2.1$ | $1.1 \pm 0.2$ | $2.2 \pm 1.0$ | $1.3 \pm 0.5$ | $1.5 \pm 0.5$ |
| 80 | 4 | $0.3 \pm 0.1$ | $0.9 \pm 0.5$ | $0.1 \pm 0.0$ | - | $0.4 \pm 0.2$ | $1.0 \pm 0.5$ | $0.1 \pm 0.1$ | $4.4 \pm 1.6$ | $0.9 \pm 0.2$ | $1.0 \pm 0.4$ | $0.4 \pm 0.2$ | $0.5 \pm 0.3$ |
| 800 | 1 | $3.1 \pm 0.7$ | $3.1 \pm 0.7$ | $3.2 \pm 1.0$ | $6.8 \pm 1.2$ | $2.9 \pm 0.6$ | $2.9 \pm 0.6$ | $2.8 \pm 0.7$ | $6.4 \pm 1.5$ | $1.5 \pm 0.4$ | $1.5 \pm 0.4$ | $1.3 \pm 0.3$ | $1.3 \pm 0.3$ |
| 800 | 2 | $1.3 \pm 0.4$ | $3.6 \pm 1.2$ | $0.8 \pm 0.5$ | $7.4 \pm 0.9$ | $1.3 \pm 0.3$ | $3.2 \pm 1.1$ | $0.8 \pm 0.4$ | $6.4 \pm 1.9$ | $1.2 \pm 0.3$ | $1.9 \pm 0.5$ | $1.1 \pm 0.3$ | $1.1 \pm 0.3$ |
| 800 | 4 | $0.3 \pm 0.2$ | $0.9 \pm 0.6$ | $0.6 \pm 0.5$ | - | $0.4 \pm 0.2$ | $0.9 \pm 0.6$ | $0.1 \pm 0.0$ | $5.8 \pm 1.4$ | $0.9 \pm 0.2$ | $1.1 \pm 0.5$ | $0.4 \pm 0.2$ | $0.4 \pm 0.2$ |

two dimensions of the estimated posterior distributions corresponding to the configurations $(80, 1)$ and $(800, 1)$ from Table 1 for one of the randomly generated measurement model ($\mathbf{H}$). Illustration of other settings are given in Appendix E.3. These illustrations give us insight into the behaviour of the algorithms and their ability to accurately estimate the posterior distribution. We see that no one method estimates the posterior distribution well.

### E.4 INPAINTING AND SUPER-RESOLUTION

Our experiments compare TMPD to DPS and ΠGDM, and also compare score-based methods (discretized with Euler-Maruyama) in Table 6 to diffusion methods (DDPM) in Table 5. In each comparison, we use the same score network for each method and a comparable numerical method. All methods are discretized using 1000 denoising steps. For the Markov chain methods we use DDPM and for the SDE methods we use an Euler-Maruyama discretization. We show a large sample (121) of the images used to calculate the metrics in Tables 5 and 6 in Fig. 5.

We found the hyperparameter suggested for the DDIM method for the VP SDE in Song et al. (2023) $r_t = v_t$ to give unstable solutions for the Algorithm given in Song et al. (2023). To make the method stable, we instead plug the ΠGDM posterior score approximation into a DDIM sampler in a similar way to Algorithm 1, which, for the VPSDE, brings the algorithm ΠGDM-D closer to our method; we are then able to choose $r_t^2 = v_t$ for both VP DDIM and VP SDE methods. We are able to use the hyperparameter $r_t^2 = v_t/(1 + v_t)$ as suggested by Song et al. (2023) for both VE DDIM and VE SDE methods, but note some instability for the ΠGDM VE SDE method for small noise, as shown in Fig 5. For DDPM we use the step-size constant suggested in Chung et al. (2022a) for inpainting, $\zeta_i = \zeta' / \|\mathbf{y} - \mathbf{H}\mathbf{m}_{0|t}\|$, where we tune $\zeta'$ over the suggested range of $\zeta' \in [0.1, 1.0]$ in Chung et al. (2022a) across LPIPS, MSE, PSNR and SSIM, as shown in Fig. 4 for each individual inverse problem (each line in the Tables 5 and 6).

Table 5: Noisy observation inpainting and super-resolution for VE DDPM on CIFAR-10 1k validation set. KID reported in units $\times 10^{-3}$.

| Problem | Method | FID ↓ | KID ↓ | LPIPS ↓ | MSE ↓ | PSNR ↑ | SSIM ↑ |
|---|---|---|---|---|---|---|---|
| $\sigma_y = 0.01$ | TMPD-D | **34.2** | **3.99** | $0.096 \pm 0.059$ | $0.005 \pm 0.004$ | $23.4 \pm 5.5$ | $0.778 \pm 0.075$ |
| 'box' mask | DPS-D | 39.9 | 9.1 | $0.085 \pm 0.036$ | $0.005 \pm 0.004$ | $24.2 \pm 3.0$ | $0.800 \pm 0.060$ |
| inpainting | ΠGDM-D | 40.5 | 9.5 | $0.087 \pm 0.037$ | $0.005 \pm 0.003$ | $24.2 \pm 2.9$ | $0.804 \pm 0.063$ |
| $\sigma_y = 0.01$ | TMPD-D | **38.5** | **8.32** | $0.276 \pm 0.080$ | $0.030 \pm 0.030$ | $16.0 \pm 4.4$ | $0.577 \pm 0.080$ |
| 'half' mask | DPS-D | 43.6 | 12.0 | $0.248 \pm 0.060$ | $0.031 \pm 0.020$ | $16.2 \pm 3.2$ | $0.595 \pm 0.080$ |
| inpainting | ΠGDM-D | 43.1 | 11.3 | $0.246 \pm 0.060$ | $0.030 \pm 0.020$ | $16.2 \pm 3.0$ | $0.604 \pm 0.080$ |
| $\sigma_y = 0.05$ | TMPD-D | **36.5** | **5.89** | $0.165 \pm 0.080$ | $0.007 \pm 0.014$ | $22.2 \pm 5.8$ | $0.722 \pm 0.094$ |
| 'box' mask | DPS-D | 101 | 74.2 | $0.227 \pm 0.075$ | $0.008 \pm 0.004$ | $21.7 \pm 2.1$ | $0.681 \pm 0.087$ |
| inpainting | ΠGDM-D | 90.7 | 62.2 | $0.232 \pm 0.075$ | $0.008 \pm 0.004$ | $21.4 \pm 1.7$ | $0.673 \pm 0.088$ |
| $\sigma_y = 0.05$ | TMPD-D | **41.9** | **10.6** | $0.322 \pm 0.085$ | $0.030 \pm 0.021$ | $15.3 \pm 6.7$ | $0.532 \pm 0.085$ |
| 'half' mask | DPS-D | 107 | 78.9 | $0.359 \pm 0.075$ | $0.031 \pm 0.019$ | $15.9 \pm 6.8$ | $0.507 \pm 0.082$ |
| inpainting | ΠGDM-D | 93.1 | 62.6 | $0.366 \pm 0.076$ | $0.033 \pm 0.019$ | $15.4 \pm 2.6$ | $0.506 \pm 0.081$ |
| $\sigma_y = 0.1$ | TMPD-D | **38.1** | **7.09** | $0.234 \pm 0.095$ | $0.010 \pm 0.021$ | $21.1 \pm 4.4$ | $0.650 \pm 0.107$ |
| 'box' mask | DPS-D | 86.4 | 50.8 | $0.323 \pm 0.081$ | $0.012 \pm 0.006$ | $19.6 \pm 2.0$ | $0.579 \pm 0.093$ |
| inpainting | ΠGDM-D | 130 | 99.5 | $0.383 \pm 0.087$ | $0.015 \pm 0.004$ | $18.3 \pm 1.0$ | $0.524 \pm 0.109$ |
| $\sigma_y = 0.1$ | TMPD-D | **41.1** | **9.58** | $0.369 \pm 0.091$ | $0.032 \pm 0.019$ | $15.2 \pm 6.0$ | $0.485 \pm 0.092$ |
| 'half' mask | DPS-D | 93.2 | 61.3 | $0.440 \pm 0.074$ | $0.038 \pm 0.021$ | $14.7 \pm 2.3$ | $0.398 \pm 0.080$ |
| inpainting | ΠGDM-D | 128 | 100 | $0.474 \pm 0.081$ | $0.042 \pm 0.019$ | $14.1 \pm 1.9$ | $0.397 \pm 0.087$ |
| $\sigma_y = 0.01$ | TMPD-D | **34.5** | **1.58** | $0.140 \pm 0.092$ | $0.006 \pm 0.019$ | $23.7 \pm 5.7$ | $0.806 \pm 0.121$ |
| 2× 'nearest' | DPS-D | 43.4 | 9.00 | $0.136 \pm 0.052$ | $0.004 \pm 0.002$ | $24.7 \pm 2.5$ | $0.832 \pm 0.066$ |
| super-resolution | ΠGDM-D | 38.7 | 7.58 | $0.126 \pm 0.049$ | $0.004 \pm 0.003$ | $24.3 \pm 2.7$ | $0.819 \pm 0.075$ |
| $\sigma_y = 0.01$ | TMPD-D | **38.2** | **5.68** | $0.293 \pm 0.089$ | $0.011 \pm 0.006$ | $19.9 \pm 4.0$ | $0.541 \pm 0.117$ |
| 4× 'bicubic' | DPS-D | 54.5 | 19.9 | $0.303 \pm 0.073$ | $0.011 \pm 0.005$ | $20.0 \pm 2.2$ | $0.541 \pm 0.094$ |
| super-resolution | ΠGDM-D | 42.7 | 9.21 | $0.275 \pm 0.080$ | $0.012 \pm 0.006$ | $19.6 \pm 2.4$ | $0.529 \pm 0.119$ |
| $\sigma_y = 0.05$ | TMPD-D | **37.0** | **1.10** | $0.223 \pm 0.012$ | $0.007 \pm 0.028$ | $21.9 \pm 7.7$ | $0.737 \pm 0.150$ |
| 2× 'nearest' | DPS-D | 101 | 71.6 | $0.286 \pm 0.079$ | $0.007 \pm 0.003$ | $21.8 \pm 1.6$ | $0.704 \pm 0.083$ |
| super-resolution | ΠGDM-D | 109 | 81.8 | $0.313 \pm 0.086$ | $0.009 \pm 0.003$ | $20.7 \pm 1.5$ | $0.657 \pm 0.094$ |
| $\sigma_y = 0.05$ | TMPD-D | **36.2** | **3.62** | $0.379 \pm 0.009$ | $0.016 \pm 0.028$ | $18.4 \pm 4.7$ | $0.432 \pm 0.119$ |
| 4× 'bicubic' | DPS-D | 126 | 98.0 | $0.469 \pm 0.075$ | $0.017 \pm 0.007$ | $18.0 \pm 1.7$ | $0.422 \pm 0.095$ |
| super-resolution | ΠGDM-D | 120 | 85.4 | $0.452 \pm 0.086$ | $0.020 \pm 0.007$ | $17.3 \pm 1.5$ | $0.402 \pm 0.106$ |
| $\sigma_y = 0.1$ | TMPD-D | **37.2** | **4.29** | $\mathbf{0.296} \pm \mathbf{0.141}$ | $0.011 \pm 0.034$ | $20.6 \pm 5.4$ | $0.642 \pm 0.139$ |
| 2× 'nearest' | DPS-D | 140 | 112 | $0.420 \pm 0.078$ | $0.013 \pm 0.004$ | $19.1 \pm 1.1$ | $0.561 \pm 0.093$ |
| super-resolution | ΠGDM-D | 161 | 135 | $0.451 \pm 0.087$ | $0.019 \pm 0.004$ | $17.4 \pm 1.0$ | $0.499 \pm 0.111$ |
| $\sigma_y = 0.1$ | TMPD-D | **35.9** | **2.32** | $\mathbf{0.441} \pm \mathbf{0.101}$ | $0.020 \pm 0.008$ | $17.2 \pm 4.7$ | $0.346 \pm 0.119$ |
| 4× 'bicubic' | DPS-D | 155 | 135 | $0.537 \pm 0.070$ | $0.022 \pm 0.008$ | $16.7 \pm 1.4$ | $0.350 \pm 0.092$ |
| super-resolution | ΠGDM-D | 217 | 209 | $0.564 \pm 0.075$ | $0.019 \pm 0.004$ | $14.5 \pm 0.9$ | $0.281 \pm 0.104$ |

Table 6: Noisy observation inpainting and super-resolution for the reverse VE SDEs on CIFAR-10 1k validation set. KID reported in units $\times 10^{-3}$.

| Problem | Method | FID ↓ | KID ↓ | LPIPS ↓ | MSE ↓ | PSNR ↑ | SSIM ↑ |
|---|---|---|---|---|---|---|---|
| $\sigma_y = 0.01$ | TMPD | **40.7** | 9.84 | $0.107_{\pm 0.060}$ | $0.006_{\pm 0.019}$ | $23.1_{\pm 4.4}$ | $0.767_{\pm 0.074}$ |
| 'box' mask | DPS | 48.5 | 16.6 | $0.305_{\pm 0.099}$ | $0.010_{\pm 0.043}$ | $20.1_{\pm 1.8}$ | $0.552_{\pm 0.132}$ |
| inpainting | ΠGDM | 77.5 | 8.61 | $0.099_{\pm 0.046}$ | $0.005_{\pm 0.003}$ | $24.1_{\pm 2.9}$ | $0.781_{\pm 0.068}$ |
| $\sigma_y = 0.01$ | TMPD | 45.4 | **12.3** | $0.279_{\pm 0.078}$ | $0.031_{\pm 0.029}$ | $16.2_{\pm 3.0}$ | $0.572_{\pm 0.071}$ |
| 'half' mask | DPS | 45.9 | 14.0 | $0.385_{\pm 0.095}$ | $0.032_{\pm 0.021}$ | $15.7_{\pm 5.7}$ | $0.569_{\pm 0.077}$ |
| inpainting | ΠGDM | 51.4 | 62.4 | $0.265_{\pm 0.069}$ | $0.032_{\pm 0.021}$ | $16.4_{\pm 3.1}$ | $0.429_{\pm 0.077}$ |
| $\sigma_y = 0.05$ | TMPD | **45.9** | **14.4** | $0.167_{\pm 0.075}$ | $0.008_{\pm 0.004}$ | $22.2_{\pm 5.8}$ | $0.714_{\pm 0.092}$ |
| 'box' mask | DPS | 51.3 | 19.8 | $0.370_{\pm 0.010}$ | $0.015_{\pm 0.006}$ | $18.6_{\pm 1.7}$ | $0.457_{\pm 0.014}$ |
| inpainting | ΠGDM | 84.3 | 16.1 | $0.157_{\pm 0.058}$ | $0.006_{\pm 0.004}$ | $23.0_{\pm 1.7}$ | $0.731_{\pm 0.080}$ |
| $\sigma_y = 0.05$ | TMPD | 52.4 | 18.9 | $0.328_{\pm 0.091}$ | $0.031_{\pm 0.036}$ | $15.1_{\pm 7.7}$ | $0.527_{\pm 0.091}$ |
| 'half' mask | DPS | **46.1** | **15.9** | $0.347_{\pm 0.092}$ | $0.031_{\pm 0.020}$ | $16.0_{\pm 3.0}$ | $0.493_{\pm 0.112}$ |
| inpainting | ΠGDM | 56.4 | 16.6 | $0.307_{\pm 0.080}$ | $0.028_{\pm 0.019}$ | $16.4_{\pm 3.0}$ | $0.546_{\pm 0.087}$ |
| $\sigma_y = 0.1$ | TMPD | 50.2 | 16.7 | $0.241_{\pm 0.096}$ | $0.009_{\pm 0.027}$ | $20.4_{\pm 8.1}$ | $0.637_{\pm 0.116}$ |
| 'box' mask | DPS | 49.0 | 18.5 | $0.543_{\pm 0.095}$ | $0.038_{\pm 0.013}$ | $14.5_{\pm 1.7}$ | $0.176_{\pm 0.100}$ |
| inpainting | ΠGDM | 88.8 | 15.9 | $0.220_{\pm 0.079}$ | $0.008_{\pm 0.004}$ | $21.8_{\pm 2.3}$ | $0.668_{\pm 0.101}$ |
| $\sigma_y = 0.1$ | TMPD | 54.4 | 21.2 | $\mathbf{0.373}_{\pm \mathbf{0.089}}$ | $0.032_{\pm 0.022}$ | $15.1_{\pm 6.3}$ | $0.481_{\pm 0.093}$ |
| 'half' mask | DPS | **48.0** | **15.9** | $0.480_{\pm 0.090}$ | $0.038_{\pm 0.021}$ | $14.7_{\pm 3.0}$ | $\mathbf{0.298}_{\pm \mathbf{0.113}}$ |
| inpainting | ΠGDM | 59.4 | 17.0 | $0.350_{\pm 0.082}$ | $0.031_{\pm 0.021}$ | $16.0_{\pm 2.9}$ | $0.505_{\pm 0.091}$ |
| $\sigma_y = 0.01$ | TMPD | 47.1 | **13.9** | $\mathbf{0.149}_{\pm \mathbf{0.086}}$ | $0.009_{\pm 0.045}$ | $23.1_{\pm 7.2}$ | $\mathbf{0.789}_{\pm \mathbf{0.111}}$ |
| 2× 'nearest' | DPS | 48.2 | 16.2 | $0.303_{\pm 0.098}$ | $0.009_{\pm 0.003}$ | $20.1_{\pm 1.8}$ | $0.616_{\pm 0.127}$ |
| super-resolution | ΠGDM | 62.2 | 20.1 | $0.261_{\pm 0.084}$ | $0.007_{\pm 0.003}$ | $22.0_{\pm 2.2}$ | $0.711_{\pm 0.076}$ |
| $\sigma_y = 0.01$ | TMPD | 53.4 | 18.1 | $\mathbf{0.297}_{\pm \mathbf{0.082}}$ | $\mathbf{0.017}_{\pm \mathbf{0.004}}$ | $19.6_{\pm 5.3}$ | $\mathbf{0.525}_{\pm \mathbf{0.119}}$ |
| 4× 'bicubic' | DPS | 47.0 | 16.5 | $0.481_{\pm 0.097}$ | $0.026_{\pm 0.010}$ | $16.1_{\pm 1.7}$ | $0.273_{\pm 0.118}$ |
| super-resolution | ΠGDM | 54.8 | 24.6 | $0.527_{\pm 0.075}$ | $0.032_{\pm 0.011}$ | $15.2_{\pm 1.7}$ | $0.224_{\pm 0.087}$ |
| $\sigma_y = 0.05$ | TMPD | 49.1 | **14.9** | $\mathbf{0.219}_{\pm \mathbf{0.091}}$ | $\mathbf{0.007}_{\pm \mathbf{0.019}}$ | $21.8_{\pm 8.1}$ | $\mathbf{0.730}_{\pm \mathbf{0.139}}$ |
| 2× 'nearest' | DPS | 49.8 | 17.5 | $0.370_{\pm 0.107}$ | $0.013_{\pm 0.005}$ | $19.1_{\pm 2.0}$ | $0.508_{\pm 0.141}$ |
| super-resolution | ΠGDM | 60.7 | 21.4 | $0.333_{\pm 0.094}$ | $0.009_{\pm 0.004}$ | $20.7_{\pm 1.9}$ | $0.625_{\pm 0.099}$ |
| $\sigma_y = 0.05$ | TMPD | 52.9 | 19.2 | $\mathbf{0.379}_{\pm \mathbf{0.009}}$ | $\mathbf{0.016}_{\pm \mathbf{0.028}}$ | $\mathbf{18.4}_{\pm \mathbf{4.7}}$ | $\mathbf{0.430}_{\pm \mathbf{0.117}}$ |
| 4× 'bicubic' | DPS | 47.1 | **16.3** | $0.538_{\pm 0.091}$ | $0.036_{\pm 0.013}$ | $14.6_{\pm 1.7}$ | $0.188_{\pm 0.101}$ |
| super-resolution | ΠGDM | 51.1 | 19.8 | $0.548_{\pm 0.082}$ | $0.037_{\pm 0.013}$ | $14.6_{\pm 1.6}$ | $0.188_{\pm 0.163}$ |
| $\sigma_y = 0.1$ | TMPD | 50.2 | **15.8** | $\mathbf{0.302}_{\pm \mathbf{0.115}}$ | $\mathbf{0.009}_{\pm \mathbf{0.008}}$ | $\mathbf{20.2}_{\pm \mathbf{7.0}}$ | $\mathbf{0.633}_{\pm \mathbf{0.144}}$ |
| 2× 'nearest' | DPS | 50.1 | 17.7 | $0.486_{\pm 0.102}$ | $0.025_{\pm 0.009}$ | $16.4_{\pm 1.6}$ | $0.301_{\pm 0.133}$ |
| super-resolution | ΠGDM | 60.9 | 23.6 | $0.391_{\pm 0.100}$ | $0.013_{\pm 0.005}$ | $19.3_{\pm 1.8}$ | $0.519_{\pm 0.113}$ |
| $\sigma_y = 0.1$ | TMPD | 51.6 | 19.9 | $\mathbf{0.441}_{\pm \mathbf{0.934}}$ | $0.020_{\pm 0.012}$ | $17.3_{\pm 3.4}$ | $\mathbf{0.343}_{\pm \mathbf{0.113}}$ |
| 4× 'bicubic' | DPS | 47.4 | **16.7** | $0.547_{\pm 0.094}$ | $0.039_{\pm 0.014}$ | $14.3_{\pm 1.7}$ | $0.174_{\pm 0.097}$ |
| super-resolution | ΠGDM | 50.1 | 18.4 | $0.560_{\pm 0.082}$ | $0.041_{\pm 0.015}$ | $14.2_{\pm 1.7}$ | $0.168_{\pm 0.085}$ |

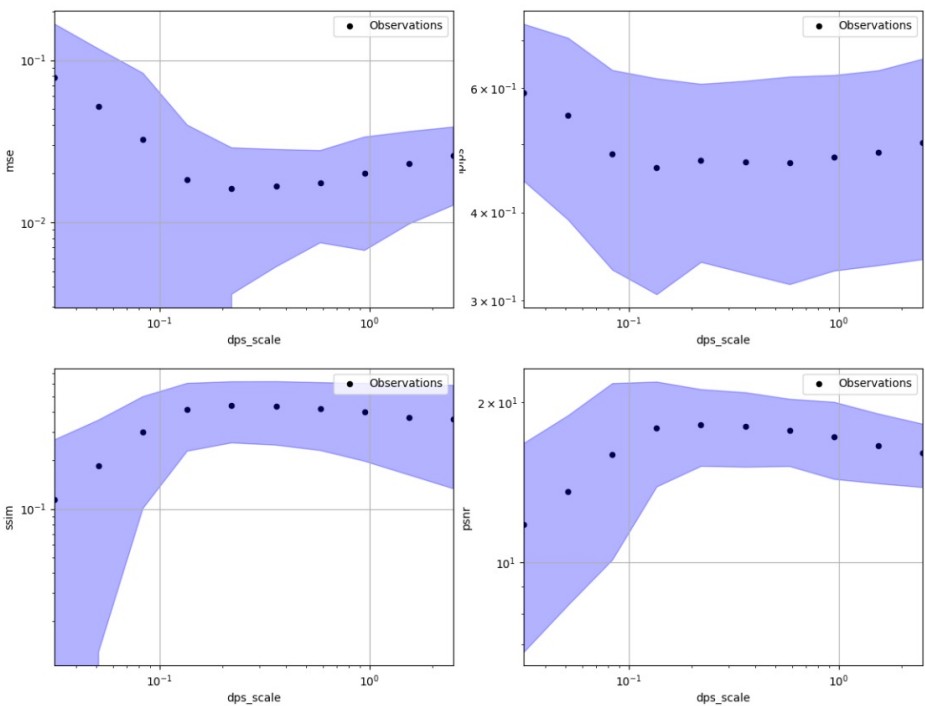

Figure 4: DPS scale hyperparameter search across LPIPS, MSE, PSNR and SSIM for CIFAR-10 $4\times$ bicubic interpolation super-resolution with $\sigma = 0.05$. Plotted are the mean values $\pm$ 2 standard deviations over a 128 sample/batch size, repeated 10 times to calculate the mean and spread. We chose an optimal value of 0.15 for the DPS scale hyperparameter in this case.

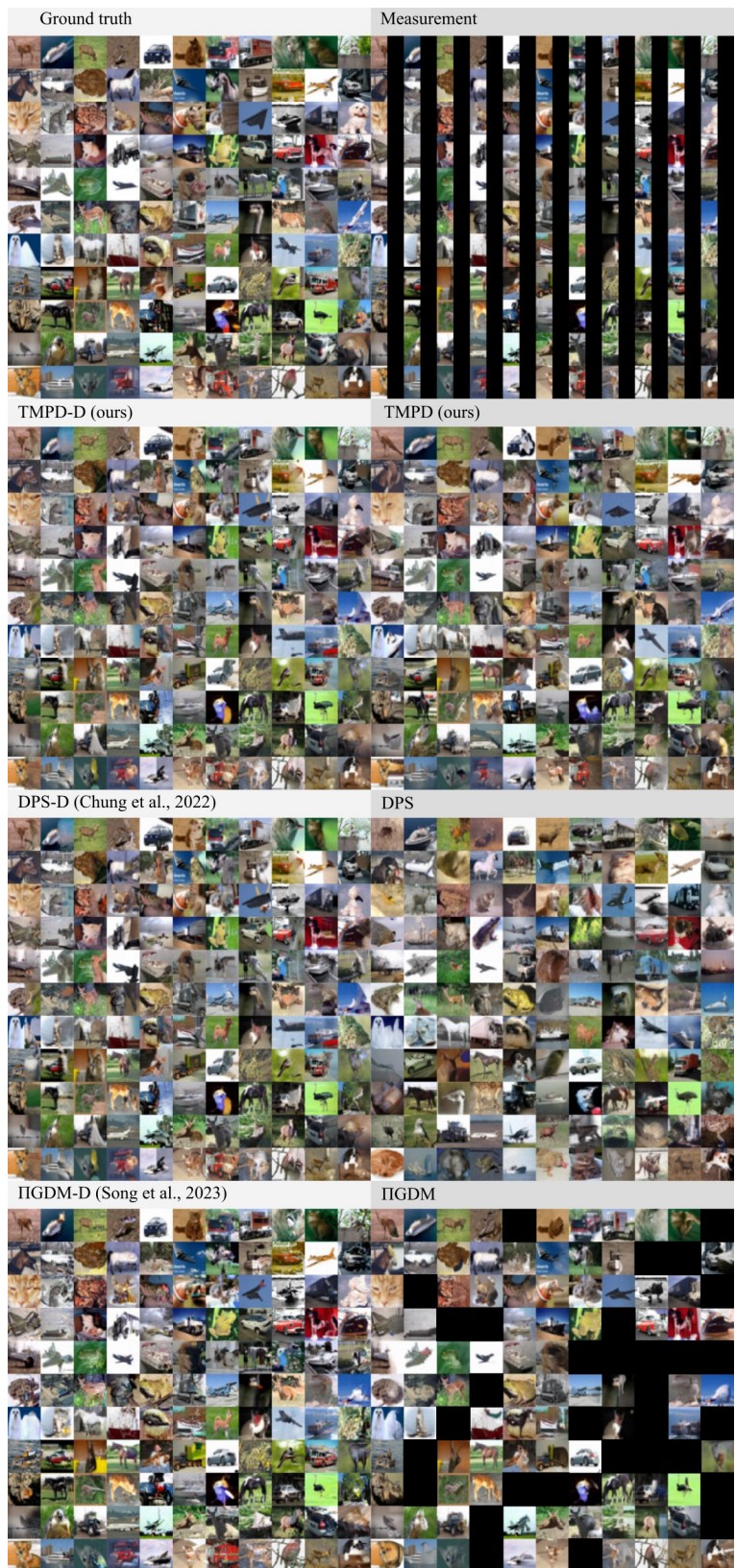

Figure 5: Inpainting samples from the VE SDE on CIFAR-10. The observation model was 'half' mask with Gaussian ($\sigma_y = 0.001$) noise.

