# OpenReview forum: "Tweedie Moment Projected Diffusions for Inverse Problems"
_ICLR.cc/2024/Conference — Submitted to ICLR 2024_

### Official Review · Reviewer_3wW3 · 2023-10-20

**Soundness:** 2 fair
**Presentation:** 2 fair
**Contribution:** 2 fair
**Rating:** 6
**Confidence:** 4

**Summary:**

This paper proposes a novel approach for inverse problem solving via diffusion models via a finer approximation of the posterior density than in prior work. The approximation leverages Tweedie's formula to estimate the posterior mean and covariance and use it find a Gaussian distribution that is close (in the sense of KL-divergence). The resulting algorithm is optimal for Gaussian data distributions, which is further supported by synthetic experiments.

**Strengths:**

- Leveraging higher order information from Tweedie's formula is an original contribution which comes with clear improvements in the accuracy of the posterior approximation.
- Tuning the weight/step size of the conditional score is a notoriously difficult problem that can easily lead to instability when done wrong. This paper obviates the need for such tuning, which has potential positive impact on designing more reliable solvers with less tuning.
- Overall, I find the paper well-written and fairly easy to follow.
- The technique has excellent performance on the syntethic experiments, providing support for the theoretical claims.

**Weaknesses:**

- The experiments are insufficient to verify the practicality of the method. Beyond the synthetic examples, no quantitative comparison is provided for actual image reconstruction problems (inpainting, SR). I recommend performing experiments in the same setting as DPS or $\Pi GDM$ in the original papers and compare the following metrics: PSNR, SSIM, LPIPS, FID. The proposed method has greatly increased compute cost, and therefore it would only make practical sense if sample quality is demonstrated to be significantly better than in competing techniques.
- Qualitative comparison seem to show that reconstruction quality is poor. There are many inconsistencies and artifacts on generated samples both for inpainting and SR (Figures 3 and 4). State-of-the-art diffusion samplers should be able to provide higher quality reconstructions visually, for instance take a look at samples generated by DPS in the original paper. I believe DPS is not set up properly for the experiments as the reconstructions are clearly not consistent with the observations. I recommend tuning the step size for DPS on a small validation set to optimize LPIPS/PSNR.
- Claims such as "having on-par restoration results with $\Pi GDM$" and "we found our approach more robust across various problems" are not supported by the paper in its current form.

**Questions:**

- How does the proposed technique compare quantitatively (both perception and distortion metrics)  to prior work on the image datasets?
- What has been done to ensure fair comparison with competing techniques in terms of hyperparameter tuning?
- What exactly justifies the increased compute cost of the proposed sampler beyond the theoretical justification? In other words, in what setting would one choose the proposed sampler over others, such as DPS?

---

> ### Author Response · Authors · 2023-11-18
> **Our response to reviewer 3wW3 (first part)**
>
> We thank the reviewer for their insightful comments. We are indeed happy to see that the reviewer observed the main strengths of our method, namely removing hyperparameter tuning requirement, as well as having excellent performance. We are also happy to see that the reviewer found our paper well-written, thank you for these comments.
>
> We address your concerns below point-by-point.
>
> >Weakness 1. The experiments are insufficient to verify the practicality of the method. Beyond the synthetic examples, no quantitative comparison is provided for actual image reconstruction problems (inpainting, SR). I recommend performing experiments in the same setting as DPS or in the original papers and compare the following metrics: PSNR, SSIM, LPIPS, FID. The proposed method has greatly increased compute cost, and therefore it would only make practical sense if sample quality is demonstrated to be significantly better than in competing techniques.
>
> We have now a comprehensively updated experimental results section with the improved code and clarity (see our "General response to reviewers"). In particular, we now provide quantitative comparison for the inpainting and super-resolution experiments on CIFAR10 (which is the same setting as in Song et. al "Score-Based Generative Modeling through Stochastic Differential Equations" https://arxiv.org/abs/2011.13456), comparing to DPS and PiGDM benchmarks using perception and distance metrics in Tables 2 and 5, which shows increased performance compared to the alternative methods for problems with large Gaussian observation noise problems, and comparitively good performance in low noise problems. To expand on this point, Table 6 presents a comparison to DPS and PiGDM baselines on CIFAR-10, which demonstrates better consistency with the whole dataset through FID and KID scores, and better consistency with the data through LPIPS, MSE, PSNR and SSIM scores.
>
> For small (\sigma_y=0.01) noise on 2 times 'nearest neighbour' super-resolution:
> FID 34.5 TMPD-D is better compared to 43.4 DPS-D and 38.7 PiGDM-D.
> LPIPS 0.140 TMPD-D is comparible compared to 0.136 DPS-D and 0.126 PiGDM-D.
>
> For medium (\sigma_y=0.05) noise on 2 times 'nearest neighbour' super-resolution:
> FID 37.0 TMPD-D is better compared to 101 DPS-D and 109 PiGDM-D.
> LPIPS 0.223 TMPD-D is better compared to 0.286 DPS-D and 0.313 PiGDM-D.
>
> For large (\sigma_y=0.1) noise on 2 times 'nearest neighbour' super-resolution:
> FID 37.2 TMPD-D is better compared to 140 DPS-D and 161 PiGDM-D.
> LPIPS 0.296 TMPD-D is better compared to 0.420 DPS-D and 0.451 PiGDM-D.
>
> The numerical results for all of the other inverse problems (4 times 'bicubic' super-resolution, ‘half’ mask inpainting and ‘half’ mask inpainting) are similar, showing TMPD has strong performance for large noise compared to DPS and PiGDM that are less (numerically and visually) consistent with the model or data for large noise. We have made sure to select optimal DPS hyperparameters by cross-validating over LPIPS, MSE, PSNR and SSIM distance metrics (see, for example, Fig. 4 for a demonstration) and use we the hyperparameters and algorithm suggested in the original paper for PiGDM. We also would like to clarify that TMPD does not require hyperparameter tuning (as also pointed out by other reviewers) which is a big plus in terms of computational cost.
>
> Furthermore, we would like to note that we also have a very practical advantage as we remove the requirement of hyperparameter tuning (as noted by other reviewers as "strengths"). This means that our method removes a very costly step that is mostly hidden in papers - but very crucial in practice. Also, our inpainting and super-resolution sampler is only 1.4-1.6 times slower when compared to DPS and PiGDM samplers across all of our image experiments.

---

> ### Author Response · Authors · 2023-11-18
> **Our response to reviewer 3wW3 (second part)**
>
> >Weakness 2. Qualitative comparison seem to show that reconstruction quality is poor. There are many inconsistencies and artifacts on generated samples both for inpainting and SR (Figures 3 and 4). State-of-the-art diffusion samplers should be able to provide higher quality reconstructions visually, for instance take a look at samples generated by DPS in the original paper. I believe DPS is not set up properly for the experiments as the reconstructions are clearly not consistent with the observations. I recommend tuning the step size for DPS on a small validation set to optimize LPIPS/PSNR.
>
> As we noted in our "general response to reviewers", this caused by an error on our side. We have now comprehensively updated these results, please see our new experimental results.
>
> We have now set up DPS properly and have, for each inverse problem, tuned the step size for DPS on a 128 size validation set to optimize LPIPS, PSNR, MSE and SSIM, please see Fig. 4 for an example.
>
> We illustrate the higher quality reconstructions visually in Figure 3 and Figure 5, and are happy to provide visuals of the full set of samples used to calculate Tables 2, 5 and 6 upon request (as the file size is quite large). In our newly shown experiments, (see Tables 2 and 5), we indeed demonstrate that our method does increase quantitative metrics significantly w.r.t. state-of-the-art.
>
> >Weakness 3. Claims such as "having on-par restoration results with PiGDM" and "we found our approach more robust across various problems" are not supported by the paper in its current form.
>
> Thank you for this comment. We agree in the first version of our paper, this claim was not supported well but after fixing our code (see "general response to reviewers") we have been able to make an explicit conclusion that our method does not fail for the problems where the additive Gaussian observation noise is large, and demonstrate on-par restoration results in terms of FID, KID, LPIPS, MSE, PSNR and SSIM in Table 2 and 5 for the low (\sigma_y = 0.01) observation noise problems and better restoration results in Table 2 and 5 for noisy (\sigma = 0.05, 0.1) problems.
>
> >Q1. How does the proposed technique compare quantitatively (both perception and distortion metrics) to prior work on the image datasets?
>
> Thank you for this suggestion. We have now compared quantitatively on both perception (FID, KID, LPIPS, SSIM) and distortion (MSE, PSNR) metrics. Table 2 shows that our method is on-par with Pi-GDM and DPS for low noise (\sigma_y = 0.01) problems and significantly better in terms of model consistency (FID and KID) for noisy (\sigma_y = 0.05, \sigma_y = 0.1) problems, and Table 5 shows that our method is on-par or better in terms of observed data consistency (LPIPS, MSE, PSNR, SSIM).
>
> >Q2. What has been done to ensure fair comparison with competing techniques in terms of hyperparameter tuning?
>
> Before our update to the paper, the paper uses the hyperparameters suggested by the authors, but since there was a bug adding much more observation noise than the hyperparameters were set to, the experimental setup was not correct. This negatively affected the performance of TMPD as well as the benchmarks DPS and PiGDM, since TMPD and PiGDM use the noise standard deviation in the algorithms, and the DPS hyperparameter was not optimal. Please see our "General response to reviewers" for a full explanation.
>
> After correcting the experimental setup, the noise standard deviation is the correct level for the observations. We have made sure to select optimal DPS hyperparameters by cross-validating over LPIPS, MSE, PSNR and SSIM distance metrics (see, for example, Fig. 4 for a demonstration). We use we the hyperparameters and algorithm suggested in the original paper for PiGDM and can motivate this choice, since the ablation studies of Table 6, Table 10, and Table 11 in Song et al., 2021a show that the use of $\eta=1.0$ in combination with a large number of noise-levels (we use 1000) gives optimal performance, and stronger performance than DPS in for low observation noise problems. The PiGDM methodology is similar to ours in the derivation of step-size but makes an impractical assumption that the image distribution is standard normal distribution, and so there is no need to select a step-size hyperparameter for the PiGDM method.

---

> ### Author Response · Authors · 2023-11-18
> **Our response to reviewer 3wW3 (third (and last) part)**
>
> >Q3. What exactly justifies the increased compute cost of the proposed sampler beyond the theoretical justification? In other words, in what setting would one choose the proposed sampler over others, such as DPS?
>
> As demonstrated in the new PDF, the cost increase is 1.4-1.6x, while this also comes with significant performance increase. We do believe that our method can be further engineered in practical settings to have even higher quality - which means that it could be a practical choice when the sample quality is the ultimate goal.
>
> Another important aspect is that we do not have hyperparameters to search for, therefore, our sampler is much more generic. We also have shown in the synthetic experiments that it produces more diverse samples, which is also important in statistical settings, e.g., the use of our method in scientific applications (as opposed to image generation).
>
> **Conclusion**
>
> We sincerely thank the reviewer for insightful and thoughtful comments and expressing their concerns. We followed their advice to improve the weak points of our paper and we believe that our updated manuscript has a much better standing as a result. We now think that we have experimental evidence that shows our method has strong qualities (increased performance as well as removal of hyperparameter tuning which is a significant cost) and obtained better theoretical understanding. We kindly request the reviewer to reevaluate the paper in the light of these changes and increase the score if the reviewer agrees that our paper now meets the standards of ICLR. Thank you.

---

> > ### Comment · Reviewer_3wW3 · 2023-11-21
> > **Response to author feedback**
> >
> > I thank the authors for the effort addressing my concerns. Please find below how my opinion has changed on each of my points after the response.
> >
> > Question 1/Weakness 1: I appreciate the effort for adding experiments on CIFAR-10, however this is a very low dimensional dataset and the findings may not extend to practical image resolutions (256x256 and above) that are investigated in recent competing methods. Thus, I am still unable to definitively verify the practical contribution and significance of the work. I would also recommend including some distortion metrics in the main part of the paper. FID alone is not a good metric to evaluate reconstructions, as data consistency is a crucial requirement when evaluating inverse problem solvers. One could achieve extremely good FID by unconditional generation that has nothing to do with the observation.
> >
> > Question 2/Weakness 2: Thank you for fixing the experiments and making sure that competing methods are tuned fairly.
> >
> > Question 3/Weakness 3: I understand that the key benefit is the lack of hyperparameter tuning at the cost of a 50% increase in compute cost in case of a low-dimensional image dataset. How does the compute overhead scale with image dimension? Can we expect similar overhead at higher image resolutions?
> >
> > Overall, I am still not convinced by the practicality of the method, but I acknowledge that this work opens up a new direction in diffusion-based solvers via leveraging second-order information. Also, the lack of hyperparameters is a neat property. Thus, I raise my score to 6 as my final decision.

---

> > > ### Author Response · Authors · 2023-11-22
> > > **Our response to reviewer 3wW3 "Response to author feedback"**
> > >
> > > > Question 1/Weakness 1: I appreciate the effort for adding experiments on CIFAR-10, however this is a very low dimensional dataset and the findings may not extend to practical image resolutions (256x256 and above) that are investigated in recent competing methods. Thus, I am still unable to definitively verify the practical contribution and significance of the work. I would also recommend including some distortion metrics in the main part of the paper. FID alone is not a good metric to evaluate reconstructions, as data consistency is a crucial requirement when evaluating inverse problem solvers. One could achieve extremely good FID by unconditional generation that has nothing to do with the observation.
> > >
> > > Thank you for your suggestion to do numerical experiments on 256x256 and above images, we are working on these experiments. We hope that the practical contribution can be determined from the paper as it stands via a combination 1) the visual results on FFHQ (256x256), 2) the fact that the method does indeed scale up to larger dimensions, since the difference in computational expense difference to PiGDM is that DTMPD with the rowsum approximation (which was used for all image experiments, as mentioned in Section 3.4) takes two vector Jacobian products compared to PiGDM which takes one vector Jacobian product. As a result the 1.4-1.6x slower claim is correct for higher dimensional (256x256 +) images also. Finally 3) the FID, PSNR and SSIM metrics reported for CIFAR10 when compared to the baselines. Thank you for your suggestion to include distortion (PSNR and SSIM) metrics in the main Table 2. We agree with the insight that FID, KID and LPIPS do not measure solving the inverse problem which is a difficult thing to evaluate without knowing or having samples from a ground truth posterior. However, solving the inverse problem requires data consistency which can be indirectly measured by using distortion metrics such as PSNR and SSIM. We have replaced the KID and LPIPS metrics with PSNR and SSIM metrics, and the KID and LPIPS metrics are still visible in Table 5 in the appendix.
> > >
> > > > Question 2/Weakness 2: Thank you for fixing the experiments and making sure that competing methods are tuned fairly.
> > >
> > > Thank you for this comment.
> > >
> > > > Question 3/Weakness 3: I understand that the key benefit is the lack of hyperparameter tuning at the cost of a 50% increase in compute cost in case of a low-dimensional image dataset. How does the compute overhead scale with image dimension? Can we expect similar overhead at higher image resolutions?
> > >
> > > Yes, we can expect similar overhead at the higher image resolutions. The computational cost per sample scales in the same manner as DPS and PiGDM with image dimension. The method does indeed scale up to larger dimensions, since the only difference in computational expense to PiGDM is that DTMPD with the rowsum approximation (which was used for all image experiments, as mentioned in Section 3.4) takes two vector Jacobian products compared to PiGDM which takes just one vector Jacobian product. As a result the 'TMPD-D is 1.4-1.6x slower than PiGDM and DPS' claim is correct for higher dimensional (256x256 +) images also.
> > >
> > > > Overall, I am still not convinced by the practicality of the method, but I acknowledge that this work opens up a new direction in diffusion-based solvers via leveraging second-order information. Also, the lack of hyperparameters is a neat property. Thus, I raise my score to 6 as my final decision.
> > >
> > > We acknowledge that more image experiments can be done, and we will be working on those. We thank the reviewer for reevaluating and acknowledging this work and would finally like to express that we have hoped to develop a method and research direction that is not limited to solving inverse problems on image datasets alone, but can be extended for use on in scientific applications or on other modalities of data.

---

### Official Review · Reviewer_ixot · 2023-10-30

**Soundness:** 2 fair
**Presentation:** 3 good
**Contribution:** 2 fair
**Rating:** 3
**Confidence:** 5

**Summary:**

In this paper, the authors proposed a new method to solve the posterior sampling by the means of diffusion models. Compared to prior works like DPS, the authors also tried to approximate the posterior score function which is theoretically intractable, but they considered a higher order moment and approximated p(x_0 | x_t) as a Gaussian distribution, instead of just the posterior mean, with statistical guarantee. Just like DPS, the technique proposed by the authors can be plugged into several types of diffusion models such as DDPM, DDIM. Finally, the authors did lots of experiments including toy cases and image restoration problems based on pre-trained score estimator, and showed better image quality.

**Strengths:**

1. The presentation of this paper is very clear and the reference list is also very complete. The writing quality is high.
2. The experiments in this paper is quite solid. The presentation of the generated images by the proposed model as well as several baseline models is very clear.

**Weaknesses:**

1. It would be better to show us some numerical results under some image metrics like PSNR or FID of the generated images, with the results of baseline methods added, so that the audience will know whether the image quality really increases numerically.
2. In Proposition 3, the authors assumed that the initial distribution p_{data} is Gaussian, which is understandable since p(x_0 | x_t) is approximated by using a posterior Gaussian. However, I would say the assumption is very strong and impractical. It would be better if the authors can figure out a way to propose a better theoretical result with weaker assumptions.
3. Compared with DPS, the authors added a posterior covariance matrix, which makes the contribution quite incremental. Unless the authors can show that the added covariance can greatly improve the performance compared to DPS, or I afraid the contribution is not enough for an "accept" to a more and more competitive ML conference.

**Questions:**

1. Can you explain in which aspects, the added posterior covariance matrix is important for the conditional diffusion models?
2. Can you add some numerical results on the image restoration tasks as I said in the previous section? Thanks very much.

---

> ### Author Response · Authors · 2023-11-18
> **Our response to reviewer ixot (first part)**
>
> We thank the reviewer for their insightful and helpful comments. We are happy to see that the reviewer found our writing quality high and our experimental setting solid. We aim to address their points below point-by-point, especially their concerns about quantitative metrics and theoretical results.
>
> >Weakness 1. It would be better to show us some numerical results under some image metrics like PSNR or FID of the generated images, with the results of baseline methods added, so that the audience will know whether the image quality really increases numerically.
>
> We have now comprehensively updated experimental results section with the improved code and clarity (see our "General response to reviewers"). In particular, we have added results comparing to DPS and PiGDM benchmarks using perception and distance metrics in Tables 2, 5 and 6, which shows increased performance compared to the alternative methods for problems with large Gaussian observation noise problems, and comparitively good performance in low noise problems. To expand on this point, Table 6 presents a comparison to DPS and PiGDM baselines on CIFAR-10, which demonstrates better consistency with the whole dataset through FID and KID scores, and better consistency with the observation data through LPIPS, MSE, PSNR and SSIM scores.
>
> For small (\sigma_y=0.01) noise on 2 times 'nearest neighbour' super-resolution:
> FID 34.5 TMPD-D is better compared to 43.4 DPS-D and 38.7 PiGDM-D.
> LPIPS 0.140 TMPD-D is comparible compared to 0.136 DPS-D and 0.126 PiGDM-D.
>
> For medium (\sigma_y=0.05) noise on 2 times 'nearest neighbour' super-resolution:
> FID 37.0 TMPD-D is better compared to 101 DPS-D and 109 PiGDM-D.
> LPIPS 0.223 TMPD-D is better compared to 0.286 DPS-D and 0.313 PiGDM-D.
>
> For large (\sigma_y=0.1) noise on 2 times 'nearest neighbour' super-resolution:
> FID 37.2 TMPD-D is better compared to 140 DPS-D and 161 PiGDM-D.
> LPIPS 0.296 TMPD-D is better compared to 0.420 DPS-D and 0.451 PiGDM-D.
>
> The numerical results for all of the other inverse problems (4 times 'bicubic' super-resolution, ‘half’ mask inpainting and ‘half’ mask inpainting) are similar, showing TMPD has strong performance for large noise compared to DPS and PiGDM that are less (numerically and visually) consistent with the model or data for large noise. We have made sure to select optimal DPS hyperparameters by cross-validating over LPIPS, MSE, PSNR and SSIM distance metrics (see, for example, Fig. 4 for a demonstration) and use we the hyperparameters and algorithm suggested in the original paper for PiGDM. We also would like to clarify that TMPD does not require hyperparameter tuning (as also pointed out by other reviewers) which is a big plus in terms of computational cost.
>
> >Weakness 2. In Proposition 3, the authors assumed that the initial distribution $p_{data}$ is Gaussian, which is understandable since $p(x_0 | x_t)$ is approximated by using a posterior Gaussian. However, I would say the assumption is very strong and impractical. It would be better if the authors can figure out a way to propose a better theoretical result with weaker assumptions.
>
> Thank you for this comment. We would like to note Proposition 3 only outlined a specific case to show our approximation's validity in the Gaussian data case. We extended this result in Theorem 1 to general non-Gaussian distributions. To clarify this, we have now added (Gaussian data distribution) note to Proposition 3 and (General data distribution) note to Theorem 1.
>
> In Theorem 1, we assume that if the distribution is non-Gaussian, we still get an upper bound on the distance to the true posterior. How far the distance to the true posterior is, depends on the non-Gaussianity of the distribution. This is however an inherent property of our method, since we made a Gaussian approximation. Note, that none of the methods that we compare to can give similar guarantees to neither Proposition 3 or Theorem 1, since their approximations are not exact in the Gaussian setting.
>
> >Weakness 3. Compared with DPS, the authors added a posterior covariance matrix, which makes the contribution quite incremental. Unless the authors can show that the added covariance can greatly improve the performance compared to DPS, or I afraid the contribution is not enough for an "accept" to a more and more competitive ML conference.
>
> In our newly shown experiments, (see Tables 2 and 5), we indeed demonstrate that our method does increase quantitative metrics significantly w.r.t. state-of-the-art. Furthermore, we would like to note that we also have a very practical advantage as we remove the requirement of hyperparameter tuning (as noted by other reviewers as "strengths"). This means that our method removes a very costly step that is mostly hidden in papers which is crucial in practice. Also, our inpainting and super-resolution sampler is only 1.4-1.6 times slower when compared to DPS and PiGDM samplers across all of our image experiments.

---

> ### Author Response · Authors · 2023-11-18
> **Our response to reviewer ixot (second (and last) part)**
>
> >Q1. Can you explain in which aspects, the added posterior covariance matrix is important for the conditional diffusion models?
>
> The covariance approximation in the Tweedie moment projection leads to the formula in Eq. (8) where we obtain a full covariance in our approximate likelihood, rather than a tunable scalar one as used in PiGDM. Algorithmically, this acts similar to a preconditioner, scaling each dimension appropriately with respect to the uncertainty, improving the conditioning of the drift. Even though we end up approximating this matrix with a diagonal approximation, the effect of this adaptive scaling is still apparent and results in improved performance.
>
> >Q2. Can you add some numerical results on the image restoration tasks as I said in the previous section? Thanks very much.
>
> Thanks for this comment. We have now added numerical results on the image restoration tasks, and refer the reviewer to our comment for "weakness 1".
>
> **Conclusion**
>
> We thank the reviewer for their comments which, in our opinion, significantly improved the quality of our paper and made it more comprehensive. We believe that with the added experimental results as well as clarifications and improvements on theory, our paper is much higher quality, therefore also deserves another evaluation. We would be very happy if the reviewer can reevaluate our paper and increase the score if the updated paper meets the standards of ICLR in their view. Thank you.

---

### Official Review · Reviewer_uiGE · 2023-10-30

**Soundness:** 3 good
**Presentation:** 3 good
**Contribution:** 3 good
**Rating:** 8
**Confidence:** 4

**Summary:**

This paper proposes a conditional sampler of diffusion models for solving Bayesian linear problems. This conditional sampler explores the second-order Tweedies formula, which extends beyond the prior works that only make use of the first order information. The proposed approach essentially forms a Gaussian approximation to the intermediate posterior distributions (along the generation steps), and eliminates the need of choosing a time-dependent variance schedule required by prior works. This approximation is exact when the true data density is indeed Gaussian and the score function is learned perfectly (and when taking the full gradient); when it not the case, the authors prove an error bound.
The method is evaluated and compared to prior works on a synthetic example (which reveals a superior performance in terms of the wasserstein distance metric), and real image restoration tasks.

**Strengths:**

I've updated my score from 5 to 8 after the rebuttal.

------
The paper is very well-written and easy to follow. The proposed method is a very elegant and simple way to extend and improve upon the prior works (namely DPS and $\Pi$GDM)  along this line. The most practical point is that it (potentially) removes the procedure  to select a time-dependent step-size hyperparameters. The synthetic study also demonstrates that the proposed method can better capture the true posterior. Lastly, the theoretical analysis should be helpful for analyzing other similar algorithms in this space.

**Weaknesses:**

I have identified two significant limitations:

1. Practical Usability and Compute Cost:
   One major limitation I've identified is the insufficient discussion of practical usability, particularly in terms of the associated compute cost. Although the authors briefly mentioned the drawback that "TMPD is slower, as each iteration costs more memory and compute..." in the concluding section, I believe that this point requires more emphasis to address its impact on real-world applications.

To improve the clarify of writing, I recommend moving a summary of the discussion on computation cost comparison and cheaper approximations from Appendix E.1 into the main text. This change would provide readers with a clearer understanding of the computational trade-offs and make the paper more accessible to a broader audience.

On the practical side, I doubt if this second-order approximation can be practical without resorting to low-rank approximation or other proxy approaches (while the authors assert the computation is not expensive for inpainting but that cannot be generalized to other tasks).

2. Limited Empirical Evaluation:
   Another significant limitation is the limited empirical evaluation presented in the paper. While the paper showcases samples generated by the proposed method and other competing methods for inpainting and super-resolution tasks, it is not immediately evident how the performance of the proposed method differs from the alternatives. The paper lacks a comprehensive summary of the qualitative differences observed in these samples. To address this limitation, I suggest incorporating quantitative metrics, such as FID or IS.

**Questions:**

- Page 3, bottom part, "the central to our method" is this a typo? this phrase is not clear to me.

- Theorem 1: can yo unpack how "we can bound the approximation error of our SDE solely due to approximations made for the likelihood."? The bound in the theorem depends on H, and the property of Phi (which characterizes the behavior of how true data density deviates from a Gaussian). Where does the approximation to the likelihood enter in to this upper bound?

- On page 8 "our chosen hyperparameter η = 1.0 (Song et al., 2021a)". Have you defined $\eta$?

- On page 9: "explore more efficient sampling techniques, such as ensemble methods or low rank approximations" can you expand on how ensemble methods can be useful here?

- On page 9 "On the positive side, we found our approach more robust across various problems due to the stabilising effect of better approximations.", similar to my comment in Weakness 2, I don't see how this point is supported other than the synthetic results.

 - On page 18 the compute cost: what is N? The number of data points to generate?

---

> ### Author Response · Authors · 2023-11-18
> **Our response to reviewer uiGE (first part)**
>
> We thank the reviewer for their careful and informative review. We followed most of their suggestions and addressed questions, which significantly improved the quality of our work. We are happy to see that the reviewer found our work well written and observed that our theoretical results could indeed help analysing similar algorithms for inverse problems. We address weaknesses and questions below.
>
> >I have identified two significant limitations:
>
> >Weakness 1. Practical Usability and Compute Cost: One major limitation I've identified is the insufficient discussion of practical usability, particularly in terms of the associated compute cost. Although the authors briefly mentioned the drawback that "TMPD is slower, as each iteration costs more memory and compute..." in the concluding section, I believe that this point requires more emphasis to address its impact on real-world applications.
>
> We found that, for real world applications, sampling is 1.4-1.6 times slower than DPS and PiGDM for inpainting and superresolution. However, we would like to point out, with our updated results comparing to DPS and PiGDM benchmarks using perception and distance metrics in Tables 2, 5 and 6, this comes with an increased performance. We also would like to clarify the removal of hyperparameter tuning (as also pointed out by other reviewers) is a big plus in terms of computational cost.
>
> To make these points clear in our updated manuscript, we have added a Section 3.4 that discusses the computational cost of the additional calculation, and suggests a computationally cheaper approximation of the Gaussian projection.
>
> Considering now the practical limitations of our method, for high dimensional problems where the row sum of the jacobian is not approximately equal to the diagonal of the jacobian, which occurs for non-linear problems or problems with lots of mixing components in the obervation matrix H, then computational cost of calculating and inverting the full jacobian is much higher. To treat this case, one could explore more advanced low rank or ensemble approximations to the Jacobian, which we leave to future work. For low dimensional problems this is not an issue, and our method can be used as an efficient monte-carlo sampler, as demonstrated in the synthetic examples, where our method clearly outperforms DPS and PiGDM.
>
> >To improve the clarify of writing, I recommend moving a summary of the discussion on computation cost comparison and cheaper approximations from Appendix E.1 into the main text. This change would provide readers with a clearer understanding of the computational trade-offs and make the paper more accessible to a broader audience.
>
> Thank you for this comment. We have summarised Appendix E.1 in the main text, which is now Section 3.4 in the updated manuscript, and improved the clarity in the appendix.
>
> >On the practical side, I doubt if this second-order approximation can be practical without resorting to low-rank approximation or other proxy approaches (while the authors assert the computation is not expensive for inpainting but that cannot be generalized to other tasks).
>
> We agree with the reviewer a full second-order approximation comes with a significant cost - but we would like to make a few arguments why this cost might be justified even in the setting of diagonal/low-rank approximations. In low dimensions, of course our method has a significant advantage of having a full-preconditioner in effect. In approximate settings (where this second-order approximation is replaced by a diagonal or low-rank matrices), we still argue that this brings a dimension-wise scaling that adapts the diffusion sampler and improves the condition number during sampling. In fact, low rank or diagonal preconditioning approaches are very popular and successful in their own right (e.g. Adam and RMSprop in stochastic optimisation), therefore we think that developing a principled way to obtain these matrices are invaluable.

---

> > ### Comment · Reviewer_uiGE · 2023-11-18
> > **Further questions regarding complexity and experiment results**
> >
> > I would like to thank the authors for their detailed response regarding complexity and the added discussion. I have the following further questions:
> >
> > 1. While the diagonal approximation to the Jacobian matrix (termed as DTMPD) seems reasonable in high-dimensional setting, are the experiment results presented in the main paper (mainly Table1, Figure 2, 3, Table 2) from the non-diagonal, i.e. full-rank Jacobian method (TMPD)? If so, can you list the practical memory and time comparison of TMPD(-D) to DPS(-D) and ΠGDM(-D)? I think that would be a more comprehensive comparison for practical reference. In addition, do you have results (generated samples and test statistics e.g. FID) for the DTMPD variant?
> >
> > 2. I wonder why the diagonal approximation with row sum of the Jacobian matrix is a reasonbale approximation? I see that when the matrix is almost diagonal the row sum is closed to the diagonal elements, but in the case why don't we just use the diagonal elements? Or do you have any external references that using row sum might be a good practical choice?
> >
> > 3. While I agree that low-rank approximation is a common practice in other fields, I don't see in the setting of this paper how to directly derive a low-rank approximation (or develop a good preconditioner for the matrix solve). Can the authors shed some light on this point?

---

> > > ### Author Response · Authors · 2023-11-20
> > > **Our response to reviewer uiGE (Further questions regarding complexity and experiment results, second part)**
> > >
> > > >  I wonder why the diagonal approximation with row sum of the Jacobian matrix is a reasonbal approximation? I see that when the matrix is almost diagonal the row sum is closed to the diagonal elements, but in the case why don't we just use the diagonal elements? Or do you have any external references that using row sum might be a good practical choice?
> > >
> > > We thank the reviewer for this observant question. We think that the rowsum of the Jacobian matrix is a reasonable approximation to the diagonal since pixels far away in the image are not (linearly) correlated to each other, so those values in the covariance matrix/ Jacobian are zero. So the row sum should well approximate the diagonal of the jacobian for images with inpainting and superresolution operators that do not have many mixing components in their observation matrix H. The approximation error between the rowsum and diagonal will be due to the fact that neighbouring pixels are (linearly) correlated. We don't just use the diagonal elements because there is currently no natural way to compute the diagonal of the Jacobian via JAX's transformations, as we explained in our previous comment. It turns out that the row sum approximation is computationally cheaper than the diagonal since the row sum approximation requires only one vector jacobian product for inpainting, super-resolution and deblur, whereas the diagonal requires min(d_y, d_x) vector Jacobian products, since, in general, each element of the jacobian evaluated at x depends on every element of x. We don't have any external references that suggest using a row sum might be a good practical choice, but we instead point to our strong empirical results on CIFAR-10 and FFHQ. We would be happy to receive any such external references.

---

> ### Author Response · Authors · 2023-11-18
> **Our response to reviewer uiGE (second part)**
>
> >Weakness 2. Limited Empirical Evaluation: Another significant limitation is the limited empirical evaluation presented in the paper. While the paper showcases samples generated by the proposed method and other competing methods for inpainting and super-resolution tasks, it is not immediately evident how the performance of the proposed method differs from the alternatives. The paper lacks a comprehensive summary of the qualitative differences observed in these samples. To address this limitation, I suggest incorporating quantitative metrics, such as FID or IS.
>
> We have now a comprehensively updated experimental results section with the improved code and clarity (see our "General response to reviewers"). In particular, we have added results comparing to DPS and PiGDM benchmarks using perception and distance metrics in Tables 2 and 5, which shows increased performance compared to the alternative methods for problems with large Gaussian observation noise problems, and comparitively good performance in low noise problems. To expand on this point, Table 2 presents a comparison to DPS and PiGDM baselines on CIFAR-10, which demonstrates better consistency with the whole dataset through FID scores, and better consistency with the data through LPIPS.
>
> For small (\sigma_y=0.01) noise on 2 times 'nearest neighbour' super-resolution:
> FID 34.5 TMPD-D is better compared to 43.4 DPS-D and 38.7 PiGDM-D.
> LPIPS 0.140 TMPD-D is comparible compared to 0.136 DPS-D and 0.126 PiGDM-D.
>
> For medium (\sigma_y=0.05) noise on 2 times 'nearest neighbour' super-resolution:
> FID 37.0 TMPD-D is better compared to 101 DPS-D and 109 PiGDM-D.
> LPIPS 0.223 TMPD-D is better compared to 0.286 DPS-D and 0.313 PiGDM-D.
>
> For large (\sigma_y=0.1) noise on 2 times 'nearest neighbour' super-resolution:
> FID 37.2 TMPD-D is better compared to 140 DPS-D and 161 PiGDM-D.
> LPIPS 0.296 TMPD-D is better compared to 0.420 DPS-D and 0.451 PiGDM-D.
>
> The results for all of the other inverse problems (4 times 'bicubic' super-resolution, ‘half’ mask inpainting and ‘box’ mask inpainting) are similar, showing TMPD has strong performance for large noise compared to DPS and PiGDM that are less (numerically and visually) consistent with the model or data for large noise. We have made sure to select optimal DPS hyperparameters by cross-validating over LPIPS, MSE, PSNR and SSIM distance metrics (see, for example, Fig. 4 for a demonstration) and we then use the hyperparameters and algorithm suggested in the original paper for PiGDM. We also would like to clarify that TMPD does not require hyperparameter tuning (as also pointed out by other reviewers) which is a big plus in terms of computational cost.
>
> >Q1. Page 3, bottom part, "the central to our method" is this a typo? this phrase is not clear to me.
>
> Thank you for pointing this out, we have fixed this. This sentence now reads:
>
> Our method relies on the approximation $p_{0|t}(\mathbf{x}\_{0}|\mathbf{x}\_{t}) \approx \mathcal{N}\left(\mathbf{x}\_0; \mathbf{x}\_{0|t}(\mathbf{x}\_t), \mathbf{C}\_{0 | t}(\mathbf{x}\_t)\right)$ to make the sampling process tractable.
>
> >Q2. Theorem 1: can yo unpack how "we can bound the approximation error of our SDE solely due to approximations made for the likelihood."? The bound in the theorem depends on H, and the property of Phi (which characterizes the behavior of how true data density deviates from a Gaussian). Where does the approximation to the likelihood enter in to this upper bound?
>
> Thank you, the passage was indeed confusing and we removed it. $H$ and the noise-level do enter the bounds, but we have not made that dependence explicit. We did so, since we were more interested in the dependence of the approximation error due to our distribution being non-Gaussian (as measured in $M$ and $L$), instead of the dependence on the problem at hand. We leave adding the explicit dependence on $\|H\|$ and $\sigma_y$ for future work.
>
> >Q3. On page 8 "our chosen hyperparameter η = 1.0 (Song et al., 2021a)". Have you defined?
>
> Thanks for this comment. We have now made it clear to refer to Algorithm 1 in Song et al., 2021a for the definition of this hyperparameter. We chose not to elaborate further since we only consider DDPM and so the hyperparameter is not of use after setting it to $\eta=1.0$: whilst Song et al., 2021a use a DDIM sampler, they show that by setting $\eta=1.0$, this is equivalent to a DDPM-type sampler. We can motivate this choice, since the ablation studies of Table 6, Table 10, and Table 11 in Song et al., 2021a show that the use of $\eta=1.0$ in combination with a large number of noise-levels (we use 1000) gives optimal performance, and stronger performance than DPS for low observation noise inverse problems.

---

> ### Author Response · Authors · 2023-11-18
> **Our response to reviewer uiGE (third (last) part)**
>
> >Q4. On page 9: "explore more efficient sampling techniques, such as ensemble methods or low rank approximations" can you expand on how ensemble methods can be useful here?
>
> Ensemble methods can be used to make an approximation to the covariance with a low rank equal to the number of particles in the ensemble. The covariance we calculate via the jacobian is the covariance of the ensemble cloud of particles at the current time. We think this could be useful but will leave this direction to future work.
>
> >Q5. On page 9 "On the positive side, we found our approach more robust across various problems due to the stabilising effect of better approximations.", similar to my comment in Weakness 2, I don't see how this point is supported other than the synthetic results.
>
> We agree with the reviewer that the original submission did not support this comment, but after fixing our code (see "general response to reviewers") we have been able to make an explicit conclusion that our method does not fail for the problems where the additive Gaussian observation noise is large.
>
> >Q6. On page 18 the compute cost: what is N? The number of data points to generate?
>
> We apologise for the confusion, $N$ is the number of noise-scales (equivalently the number of steps) the diffusion model is run for. We have now clarified this in Section E.1.
>
> **Conclusion**
>
> We thank the reviewer for their thoughtful comments once again which helped to increase the quality of our manuscript significantly. We sincerely believe that our paper is now much higher quality. We therefore kindly remind the reviewer to reevaluate our paper, if they agree with us, and update their score if they are happy with our rebuttal. Thank you.

---

> ### Author Response · Authors · 2023-11-20
> **Our response to reviewer uiGE (Further questions regarding complexity and experiment results, first part)**
>
> > While the diagonal approximation to the Jacobian matrix (termed as DTMPD) seems reasonable in high-dimensional setting, are the experiment results presented in the main paper (mainly Table1, Figure 2, 3, Table 2) from the non-diagonal, i.e. full-rank Jacobian method (TMPD)? If so, can you list the practical memory and time comparison of TMPD(-D) to DPS(-D) and ΠGDM(-D)? I think that would be a more comprehensive comparison for practical reference. In addition, do you have results (generated samples and test statistics e.g. FID) for the DTMPD variant?
>
> The experimental results presented in the synthetic experiments, Table 1, and Figure 2, are using the full-rank jacobian matrix. On the other hand, all of the image experiments, Figure 3 and Table 2, use the row sum approximation.
> We have listed the theoretical memory and time complexity of TMPD-D compared to DPS-D and PiGDM-D, and, in practice, TMPD-D with the row sum approximation is 1.4-1.6x slower than DPS-D and PiGDM-D and has the same memory costs as DPS-D and PiGDM-D. For the diagonal approximation and the full Jacobian, TMPD-D can be 10x slower than DPS-D and Pi-GDM-D, but the memory cost of computing the full Jacobian is the limiting factor, since it means that only a small batch size can be used for large images.
>
> Thanks for your suggestion to do experiments for the DTMPD variant. In the time period of the review discussion we can only do synthetic experiments for this, and we are now running the synthetic experiments that we will add to Table 1 and we will upload tomorrow. It would indeed be interesting to see how this diagonal approximation performs on the synthetic experiments, and would bring Table 1 in line with Table 4, that does already compare the DTMPD variant, but for the score-based reverse-SDE.
>
> As mentioned above, performing the diagonal approximation variant on the image experiments is computationally infeasible for the timeframe of the review discussion because the best method that is currently possible to compute the diagonal of the jacobian using JAX's transformations involves infact computing the full Jacobian, and then taking the diagonal. We expand on this point by providing commented pseudo-code, below. Our rowsum approximation was used in all of the image experiments, as we noted in section 3.4 of the paper (and also Appendix E.1 in the original submission). The diagonal approximation serves as a stepping stone to give both inspiration and justification for the rowsum approximation.
>
> ```
> > def get_grad_estimate_x_0_vmap(observation_map):
> >     """Method as suggested in https://stackoverflow.com/questions/70956578/jacobian-diagonal-computation-in-jax """
> >
> >   def estimate_x_0_single(val, i, x, t, timestep):
> >         x_shape = x.shape
> >         x = x.flatten()
> >         x = x.at[i].set(val)
> >         x = x.reshape(x_shape)
> >         x = jnp.expand_dims(x, axis=0)
> >         t = jnp.expand_dims(t, axis=0)
> >         v = discrete_sigmas[timestep]**2
> >         s = score(x, t)
> >         x_0 = x + v * s
> >         h_x_0 = observation_map(x_0)
> >         return h_x_0[i]
> >     return vmap(value_and_grad(estimate_x_0_single), in_axes=(0, 0, None, None, None))
> >
> > grad_estimate_x_0_vmap = get_grad_estimate_x_0_vmap(observation_map)
> >
> > def analysis(y, x, t, timestep, ratio):
> >     h_x_0, vjp_h_x_0, (_, x_0) = vjp(
> >     lambda x: estimate_h_x_0_vmap(x, t, timestep), x, has_aux=True)
> >
> >     # There is no natural way to get the diagonal of a Jacobian with JAX's transforms:
> >     # you cannot map the input, because in general each diagonal entry of the jacobian depends on all inputs.
> >
> >     # Calculating the full Jacobian seems like the best method, but the batch size cannot be large due to memory constraints, and so numerical evaluation is slow
> >     vec_vjp_h_x_0 = vmap(vjp_h_x_0)
> >     diag = jnp.diag(batch_observation_map(vec_vjp_h_x_0(jnp.eye(y.shape[0]))[0]))
> >
> >     # # This method gives OutOfMemory error
> >     # idx = jnp.arange(len(y))
> >     # h_x_0, diag = grad_estimate_x_0_vmap(x.flatten(), idx, x, t, timestep)
> >     # diag = self.observation_map(diag)
> >
> >     # # This method cannot be XLA compiled and is too slow for numerical evaluation experiments
> >     # diag = jnp.empty(y.shape[0])
> >     # for i in range(y.shape[0]):
> >     #   eye = jnp.zeros(y.shape[0])
> >     #   diag_i = jnp.dot(self.observation_map(vjp_h_x_0(eye)[0]), eye)
> >     #   diag = diag.at[i].set(diag_i)
> >
> >     C_yy = diag + self.noise_std**2 / ratio
> >     ls = vjp_h_x_0((y - h_x_0) / C_yy)[0]
> >     return x_0.squeeze(axis=0) + ls
> ```

---

> ### Author Response · Authors · 2023-11-20
> **Our response to reviewer uiGE (Further questions regarding complexity and experiment results, second (and last) part)**
>
> > While I agree that low-rank approximation is a common practice in other fields, I don't see in the setting of this paper how to directly derive a low-rank approximation (or develop a good preconditioner for the matrix solve). Can the authors shed some light on this point?
>
> The ensemble Kalman filter provides methodology to do a low rank approximation using J particles, where J is the rank of the approximation, by computing an empirical covariance matrix with J particles. The algorithm will only operate on the J particles, so should have only O(J d_y) memory complexity. We leave this extension to future work. We are happy to provide further references for this methodology upon request.
>
> Thank you again to the reviewer for these questions on the numerical implementation of our method, and we will be updating the synthetic experiments section tomorrow as a result of the suggestions given by the reviewer.

---

> > ### Comment · Reviewer_uiGE · 2023-11-20
> >
> > Thank you for your thoughtful response! I appreciate attaching the code snippet and the detailed reply. My questions have now been addressed and I believe the proposed approach would be a promising solution to a more refined variance schedule. I will definitely appreciate updating the synthetic experiments, and if possible, more empirical results on established benchmark tasks.

---

> ### Author Response · Authors · 2023-11-21
> **Rebuttal revision PDF includes new Table 3, please note Table3->Table4, Table4->Table5, Table5->Table6**
>
> Thank you for your review and discussion. We have updated the synthetic experiments to include the diagonal approximation the full Jacobian with TMPD for DDPM (denoted DTMPD-D in the updated summary Table 2) and we are working on more empirical results on established benchmark tasks, which will be uploaded before the end of the discussion period. All reviewers should note that this revision adds a Table 3 that in turn renames the tables that were part of the discussion, we have already made the edits which correct for this, and all of our comments should have the correct table numbers.

---

### Official Review · Reviewer_NB1r · 2023-11-01

**Soundness:** 2 fair
**Presentation:** 3 good
**Contribution:** 2 fair
**Rating:** 3
**Confidence:** 4

**Summary:**

This paper focuses on developing a sampler for solving inverse problems. While prior samplers use a dirac delta function to approximate the posterior, the authors use a Gaussian distribution with mean and covariance estimated using Tweedie’s formula. The key idea is to take the gradients of the score in estimating the covariance. This approximation is shown to be optimal when the data distribution is Gaussian. Experiments on denoising, inpainting, and super-resolution are conducted to support the claims. The results are compared with two baselines: DPS and piGDM.

**Strengths:**

1. While most of the prior works assume zero covariance while approximating $p_{0|t}$, the authors use Gaussian distribution to better approximate the posterior with a way to compute the covariance matrix.
2. The proposed pre-conditioning helps remove the time dependent step size in prior works.

**Weaknesses:**

1. The theoretical analysis is a simple extension of Chen et al. [2].
2. The authors use the results from Girsanov’s theorem. But there is no proof whether the condition to apply Girsanov holds in this case.
3. The experimental results do not seem correct. In fact, I suspect the baseline results, e.g. DPS results are wrong. I have tested many of the baselines myself and the results are not as bad as what the authors have shown in this paper, zoom in Fig 3, Fig 7 for instance to see the artifacts.
4. Page 9: “The ability to handle measurement noise also gives TMPD the potential to address certain real-world applications, such as CT and MRI imaging with Gaussian noise” I am not sure how TMPD handles the noise. The results are not as good as the results reported in the baseline papers, piGDM and DPS in a similar setting. Also, it is slower than piGDM.

**Questions:**

1. Missing citation: The score is trained using a denoising score-matching objective [1].
2. What is $C_{0|n}$ in the last paragraph on Page 4?
3. In equation 9, it seems like the first gradient in the right hand side is over the entire function, not just the mean, which is not correct. I ask the authors to put the brackets appropriately to make it clear.
4. What is the intuition behind the specific family of data distribution considered in Theorem 1? Is it the setting where the authors could prove something? Or does it capture any meaningful data distributions?
5. Section 6.2: The largest standard deviation of the added noise is 0.01 and the baselines still perform poorly as per the experiments reported in this paper. However, baselines such as DPS perform much better even with higher noise level (sigma = 0.05).
6. Super-resolution results of the proposed method TMPD in Fig 4 are equally bad with the unwanted color artifacts. Maybe take more measurements and compare.
7. Typo in the first equation on Page 18. Missing gradient
8. Typo on pp 14 in KL divergence. Missing gradient
9. What is the scale of M in Theorem 1? Is the dependence on M optimal?

**Reproducibility**

There are enough details present in the paper to reproduce the results.

**References**

[1] Pascal Vincent. “A connection between score matching and denoising autoencoders”. In: Neural computation 23.7 (2011), pp. 1661–1674

[2] Chen et al. “Sampling is as easy as learning the score”

---

> ### Author Response · Authors · 2023-11-18
> **Our response to Reviewer NB1r (first part)**
>
> We thank the reviewer for their careful review. The reviewer's comments resulted in significant increase in the quality of our results and we are grateful for this. We are also happy that the reviewers observed the strengths of our paper, and we reply below to both "weaknesses" and "questions" that come after.
>
> >Weakness 1. The theoretical analysis is a simple extension of Chen et al. [2].
>
> We bound the TV-Distance to the target via a triangle inequality and then bound one of the terms using Pinskers-Inequality, to get a KL-Term.  This is an attractive argument in this setting, introduced in the compelling work [2]. However, the major part of the proofs (from the conceptual difficulties as well as length-wise) consists in bounding the resulting terms, and in their case showing that one can apply Girsanov's theorem, for them as well as for us. However, these steps are inherently different:
>
> In our case, the approximation in our analytical extra drift comes from the fact that our distribution is not Gaussian (in which case the approximation would be exact). Therefore, the core of the proof consists in bounding these approximations under our assumptions on $p_\text{data}$. We do not treat time discretization. On the other hand, in [2], the errors stem from the drift being approximated by a neural network and the time-discretization of the SDE. The inherent different setting makes the contents of the proofs very different.
>
> We already cite [2], but we agree that another citation, directly at the beginning of our proof where we use the argument, is beneficial to the reader. We added the citation to the uploaded PDF. Furthermore, we hope that the preceding discussion highlighted that the proofs are distinct otherwise.
>
> >Weakness 2. The authors use the results from Girsanov’s theorem. But there is no proof whether the condition to apply Girsanov holds in this case.
>
> Thank you for reading the proof so carefully. You are right, that we should justify our use of Girsanov. We now do so by using Novikov's condition. Again, we can make use of the Gaussianity, and basically just apply the bounds on the differences in the drift that we have proven before: Since we have bounded $|\nabla \log p_t + f_t - p_t|$ by a quantity depending only on $M$, $L$ and $\|x\|$, Novikovs condition reduces to showing that
>
> $$\mathbb{E}[\exp(\frac{1}{2} c \int_0^T \|x_t\|^2 \mathrm{d}t)] < \infty.$$
>
> However, the expectation is taken with respect to the Gaussian reference process. We want to apply Novikov's condition iteratively and therefore prove that
>
> $$\mathbb{E}[\exp(\frac{1}{2} c \int_{t_i}^{t_i+1} \|x_t\|^2 \mathrm{d}t)] < \infty$$
>
> for $\Delta_i = t_{i+1} - t_i$ small enough. Since our process is well behaved (Gaussian process with bounded mean and lower/upper bounded covariance) we can show that we can lower bound the size of $\Delta_i$ and iterate this procedure. This argument is taken from Corollary 3.5.16 in [B].
>
> [B] Karatzas and Shreve, "Brownian motion and stochastic calculus"
>
> >Weakness 3. The experimental results do not seem correct. In fact, I suspect the baseline results, e.g. DPS results are wrong. I have tested many of the baselines myself and the results are not as bad as what the authors have shown in this paper, zoom in Fig 3, Fig 7 for instance to see the artifacts.
>
> The experimental results for the image experiments were indeed not correct due to a typo/bug in the code (please see our "general response to reviewers").
>
> The colour artifacts in Fig 3 and Fig 7 were due to the noise_std being very small compared to the observation noise (due to a typo in the code), as a result the DPS hyperparameter was not set correctly. We have now corrected the bug and have also optimised over LPIPS, SSIM, MSE and PSNR to choose the DPS scale hyperparameter in all image experiments. This is the only part of the code that has changed in our updated experiments. We believe the DPS implementation is correct as it closely follows the original code by Chung et al. Although the code has not been released for \PiGDM, we believe our code is correct as it follows their algorithm exactly.

---

> ### Author Response · Authors · 2023-11-18
> **Our response to Reviewer NB1r (second part)**
>
> >Weakness 4. Page 9: “The ability to handle measurement noise also gives TMPD the potential to address certain real-world applications, such as CT and MRI imaging with Gaussian noise” I am not sure how TMPD handles the noise. The results are not as good as the results reported in the baseline papers, piGDM and DPS in a similar setting. Also, it is slower than piGDM.
>
> As we have noted above, and in our general comment, the comparisons in the original PDF were not correct due to a bug. As many reviewers rightly pointed out, we investigated this issue and now we are confident that our results reflect the improved quality of our samples.
>
> Table 2 presents a comparison to DPS and PiGDM baselines on CIFAR-10, which demonstrates better consistency with the whole dataset through FID scores, and better consistency with the data through LPIPS.
>
> For small ($\sigma_y=0.01$) noise on 2 times 'nearest neighbour' super-resolution:
> FID 34.5 TMPD-D is better - compared to 43.4 DPS-D and 38.7 PiGDM-D.
> LPIPS 0.140 TMPD-D is competitive - compared to 0.136 DPS-D and 0.126 PiGDM-D.
>
> For medium ($\sigma_y=0.05$) noise on 2 times 'nearest neighbour' super-resolution:
> FID 37.0 TMPD-D is better compared to 101 DPS-D and 109 PiGDM-D.
> LPIPS 0.223 TMPD-D is better compared to 0.286 DPS-D and 0.313 PiGDM-D.
>
> For large ($\sigma_y=0.1$) noise on 2 times 'nearest neighbour' super-resolution:
> FID 37.2 TMPD-D is better compared to 140 DPS-D and 161 PiGDM-D.
> LPIPS 0.296 TMPD-D is better compared to 0.420 DPS-D and 0.451 PiGDM-D.
>
> The results for all of the other inverse problems (4 times 'bicubic' super-resolution, ‘half’ mask inpainting and ‘box’ mask inpainting) are similar, showing TMPD  has strong performance for large noise compared to DPS and PiGDM that are less (numerically and visually) consistent with the model or data for large noise. We have made sure to select optimal DPS hyperparameters by cross-validating over LPIPS, MSE, PSNR and SSIM distance metrics (see Fig. 4 for a demonstration of our hyperparameters search for DPS) and use we the hyperparameters and algorithm suggested in the original paper for PiGDM.
>
> As for the computational cost, we agree that our method is 1.4-1.6x slower for a single run, this is the cost (in this case) for increased quality, coming from more accurate approximations. But we would like to point out that we have no hyperparameters in our setting (as the reviewer rightly observed in their 2nd point in "Strengths"), this means we have to factor in the cost of hyperparameter search. We claim that our method could be the method of choice due to removing this requirement (which is a necessity every time there is a new inverse problem to solve), despite the increase of the sampling cost. Furthermore, we present a mature and efficient codebase that can be used to reproduce all of our results and will be made public after the review period.
>
> >Q1. Missing citation: The score is trained using a denoising score-matching objective [1].
>
> Thank you. We have added this citation in Section 2, technical background "This can be circumvented by approximating the drift using score matching techniques...".
>
> >Q2. What is C_{0|n} in the last paragraph on Page 4?
>
> Thanks for pointing out. It's meant to read $C_{0|t}$. This is the same quantity given in Prop. 1, which is the posterior covariance of $x_0|x_t$.
>
> >Q3. In equation 9, it seems like the first gradient in the right hand side is over the entire function, not just the mean, which is not correct. I ask the authors to put the brackets appropriately to make it clear.
>
> While putting brackets in the newly updated version posed a challenge of equation being too long, we have now added a sentence right after this equation that reads "where $\nabla_{{\mathbf{x}\_t}}$ only operates on $\mathbf{m}\_{0|t}$ in the above equation."

---

> ### Author Response · Authors · 2023-11-18
> **Our response to Reviewer NB1r (third (and last) part)**
>
> >Q4. What is the intuition behind the specific family of data distribution considered in Theorem 1? Is it the setting where the authors could prove something? Or does it capture any meaningful data distributions?
>
> The goal of the Theorem is to quantify the error to the posterior. Our method is exact in the Gaussian case (Proposition 3). Due to the derivation, we cannot prove (or expect) exact posteriors in other cases. What we do in Theorem 1 is to quantify the distance of our distribution to the Gaussians in some way, and then give quantitative guarantees on the distance of our generative model to the true posterior.
>
> In general, any distribution that can be written as $\exp(f(x))$ with a $f \in C^1$ satisfies the assumptions of the Theorem, since we can just use
>
> $$\exp(f(x)) = \exp(f(x) - \frac{1}{2} \|x - \mu_0\|^2_{\Sigma_0})\exp(\frac{1}{2} \|x - \mu_0\|^2_{\Sigma_0}) = \exp(f(x) - \frac{1}{2} \|x - \mu_0\|^2_{\Sigma_0})\mathcal{N}(x;\mu_0, \Sigma_0).$$
>
> Then, the theorem gives theoretical guarantees on the performance of our method, depending on the properties of $f(x) - \frac{1}{2}\|x - \mu_0\|^2$, or equivalently, how similar $\exp(f(x))$ is to a Gaussian.
>
> The argument could be extended to more general distributions by first convolving them with a Gaussian. One then gets a representation as above. The authors in [2] and [C] used similar arguments.
>
> Summarizing, we extend Proposition 3, where we say, the error of our model on Gaussian distributions is 0, to a statement quantifying that if our distribution has distance $\text{dist}$ (measured via $M$ and $L$), then our approximation error can be bounded by an explicit formula in terms of $\text{dist}$. We hope this demonstrates that the choice of distributions is not arbitrary, but natural due to the derivation of our method. Note, that none of the baseline methods that we compare have similar guarantees, also not for strictly Gaussian measures.
>
> [C] Chen, Lee, Lu, "Improved Analysis of Score-based Generative Modeling: User-Friendly Bounds under Minimal Smoothness Assumptions"
>
> >Q5. Section 6.2: The largest standard deviation of the added noise is 0.01 and the baselines still perform poorly as per the experiments reported in this paper. However, baselines such as DPS perform much better even with higher noise level (sigma = 0.05).
>
> As we noted in our "general response to reviewers", this caused by an error on our side. We have now comprehensively updated these results, please see our new experimental results.
>
> >Q6. Super-resolution results of the proposed method TMPD in Fig 4 are equally bad with the unwanted color artifacts. Maybe take more measurements and compare.
>
> We have repeated the experiment, this time correcting the noise standard deviation parameter in the code (please see our "general response to reviewers") and we have compared FID, KID and LPIPS metrics for 2 times super-resolution and 4 times super-resolution over a 1k CIFAR-10 validation set. We can report significantly better results when compared to DPS and PiGDM, especially when the Gaussian noise is large. TPDM is sucessful where DPS and PiGDM methods fail when the noise is large. We are happy to provide the samples used to generate the Tables upon request (as the files are quite large), and our code has been updated with this experiment that can be run to reproduce all of the tables.
>
> >Q7. Typo in the first equation on Page 18. Missing gradient
>
> Thank you - we have fixed this. This is now below Eq. (33) in the new manuscript.
>
> >Q8. Typo on pp 14 in KL divergence. Missing gradient
>
> Thank you! This is fixed, this is now on page 15 in the new PDF.
>
> >Q9. What is the scale of M in Theorem 1? Is the dependence on M optimal?
>
> The scale of M is determined by the proximity of $p_{\text{data}}$ to a Gaussian. An interesting sidenote might be that the assumption implies that
>
> $$\|p_\text{data} - \mathcal{N}(\mu_0, \Sigma_0)\|_{\text{TV}} \le M -1$$
>
> Therefore, it can be interpreted as bound on the total variation distance to the Gaussians. We do not know if the $M$ dependence is optimal but happy to report that in the updated proof, we have significantly improved dependence on $M$. We think that this would be an exciting direction for future research.
>
> **Conclusion**
>
> We sincerely believe that, with the added experimental results, as well as our updated theoretical section in response to the reviewer's comment, our paper has now a much better standing. We also kindly ask reviewer to increase their score if they are satisfied by the response to acknowledge that the points are addressed and that the paper is now in good standing from a scientific perspective. Thank you.

---

### Author Response · Authors · 2023-11-18
**General response to reviewers**

We would first of all like to sincerely thank the reviewers for the comments that have, in our opinion, helped improve the paper significantly. We apologise for our late response as we have been updating our work to address all comments comprehensively.

Before posting specific replies to each reviewer, which we will do shortly, we would like to post a general comment that address the most significant concerns raised by reviewers. In what follows, we will address the concerns raised by reviewers NB1r, uiGE, ixot, and 3wW3 (i) on the validity of the image experiments and the strength of the visual results and (ii) the lack of image perception and distance metrics comparing against benchmarks. We have completed a comprehensive update to address both concerns, reporting significantly improved results, please see details below. To address the concerns regarding theoretical results raised by the reviewer NB1r, we would like to note that we have updated our theorem statement and proof and will be posting a specific reply shortly. Please also note that we have updated the PDF and the code in the Supplementary Material and will refer to the tables in the updated paper.

(i) Validity of the image experiments: We would like to alert the reviewers to a mistake in the code of the image experiments as originally submitted, in tmpd/run_lib.py on lines 108 :meth:tmpd.run_lib.get_inpainting_observationand 115 in :meth:tmpd.run_lib.get_superresolution_observation: the noise that was added to obtain the inpainting and superresolution observation was incorrect for the image experiments only (not the synthetic experiments), resulting in the actual noise_std of the observed image data being much larger (by its square root) than as reported. In other words, for the image experiments, one needs to take the square root of all of the standard deviations reported in the original submission of the paper to get the standard deviation of the actual noise that was added to the images. For example, the largest noise level we applied to the images in the original submission was in Fig. 11 $\sigma_y=\sqrt{0.1}=0.316$, Fig. 12 $\sigma_y=\sqrt{0.05}=0.22$, and Fig.6 $\sqrt{0.01}=0.1$. This mistake is obvious when comparing the DPS [A] to our work visually, since our observed images have much more noise than those in [A] with the same reported noise standard deviation.

We are pleased to have found this typo in the code since the small mistake had detrimental impact on the visual quality of the samples of all of the presented methods, including the baselines. Since the true noise standard deviation was larger than the parameter set as a variable in the code, all of the algorithms, including TMPD and the baselines of DPS and PiDGM, were run with the incorrect noise_std variable for the inverse problem, which greatly worsened the visual quality of the samples, resulting in visually poor samples as the reviewers rightly pointed out. With this mistake corrected, we remade the visual figures, which show greatly improved visual samples.

(ii) the lack of image perception and distance metrics: We have taken on board the advice from the reviewers which is to run perception, distortion and distance metrics experiments in a similar setting to the other baselines. We are happy to report that TMPD is statistically significantly better on FID, KID and better or on par in LPIPS, MSE, PSNR and SSIM than the baselines PiGDM and DPS. We report these results in Tables 2 and 5 in an update to the paper pdf file. We will upload the generated images used to calculate the metrics in these tables upon request, since the file size is so large.

We would also like to thank the reviewers for pointing out the typos which have now been fixed. We kindly ask the reviewers to review our response in detail and reconsider their score if they are happy with the new results.

[A] Chung, Hyungjin and Kim, Jeongsol and Mccann, Michael Thompson and Klasky, Marc Louis and Ye, Jong Chul, Diffusion Posterior Sampling for General Noisy Inverse Problems, The Eleventh International Conference on Learning Representations, 2022

---

> ### Author Response · Authors · 2023-11-21
> **Dear reviewers, please review our rebuttal before discussion ends tomorrow**
>
> Dear reviewers,
>
> We thank you for your initial reviews, and thank Reviewer uiGE for their timely response and discussion to our rebuttal.
>
> We kindly ask all other reviewers (NB1r, ixot, 3wW3) to read our rebuttal (general and specific responses) and please reevaluate our paper. We genuinely believe that we have addressed all main concerns, with a significant update to empirical results (including all quantitative metrics requested) as well as mathematical clarification of theoretical results. We would really appreciate if you could take a look and reevaluate your scores as we think the current form of paper deserves a reevaluation.
>
> Thank you for your service!

---

### Meta-Review · Area_Chair_g8x1 · 2023-12-07

**Metareview:**

The paper proposed an improved technique to leverage diffusion model to solve inverse problems. The method is based on the "classifier-guided conditional generation" formulation of diffusion models, but (1) assumes a linear Gaussian emission/observation model (as the "classifier" or "likelihood") and (2) uses a Gaussian approximation to the denoising distribution $p(x_0 | x_t)$ (as the "prior"), so that the classifier-guided score (as the score of "posterior") can be derived in closed form. The correct "classifiers" for intermediate diffusion time steps are respected, which are otherwise hard to estimate. The key technical contribution of the method is the estimation of the covariance matrix of the denoising distribution, which is taken for the Gaussian approximation. An analysis on the bias due to the Gaussian approximation is also provided as a bound in terms of the non-Gaussianity of the data distribution in a certain sense. These contributions are clear and reasonable.

Nevertheless, a significant amount of critical experimental results are provided only in the rebuttal phase, including quantitative evaluations, and a correct comparison with baselines. The theoretical analysis is also substantially revised during the period. There also remain concerns on the technical novelty and the significance of the experimental results (e.g., relatively low dimensional problems, some unconventional comparison settings).

**Justification For Why Not Higher Score:**

As the initial review opinions are quite diverse, I called for discussions. All reviewers replied. I think it is appropriate to re-post their further comments (cropped some due to length limit), for the transparency and informativeness to the authors.

* Reviewer NB1r: "After reading all the reviews and response, I am still not convinced about the technical contribution and experimental results. Therefore, I am leaning towards "reject" unless I am missing something.\
  1. The main Theorem 1 has been completed changed during the rebuttal. The proof reads as a completely new paper. There was a weak dependence on the constant M, i.e., the previous upper bound on the divergence was $O(M^{12})$ where $M > 1$. This bound is to loose to convey any meaningful insights on the divergence between true posterior and learned posterior.\
  2. The new Theorem 1 has a better dependence on $M$, but the assumptions are too strong to provide any new insights that was not presented in prior works [1,2].\
  3. The experimental results are still not clear to me. The results of the baselines DPS and $\Pi$GDM don't look as good as other papers have reported under the same noise level, i.e., $\sigma = 0.05$ or even higher noise level $\sigma = 0.5$ [3,4,5]. See for example Fig 1 in [3].\
  4. The experimental setup is not consistent with prior inverse problem solvers, which makes it difficult to compare.\
  ...the paper presents a new algorithm which might be useful in practice, but the lack of consistent evaluation and clear technical contributions outweigh the merits.\
[1] Chen... Sampling is as easy...\
[2] Li... Towards Faster Non-Asymptotic... \
[3] Meng... Diffusion Model Based Posterior ...\
[4] Rout... Solving linear inverse problems provably...\
[5] Chung... Prompt-tuning latent diffusion..."

* Reviewer ixot: "I still choose to insist "reject". From the aspect of the overall idea and contributions, I think this paper is quite incremental, at least lower than the average quality of accepted papers. I believe the model proposed by the authors is no more than a DPS with covariance matrix added, with only the diagonal elements are taken into account. Of course, the additional technique is novel and the authors also studied some theory based on existing proving techniques and strong assumptions, the novelty of this work does not reach the borderline."

* Reviewer uiGE: "I have read the reviews from other reviewers. While I agree that the work is indeed incremental..., however, I still think this contribution is valuable. ...simplicity of the extension that could potentially make a large impact in practical applications, and hence I don't think "incremental contribution" alone should lead to a rejection of a paper. ...different scaling factors for DPS-like methods would have tremendous impacts on the results as shown by many prior works...mainly using a hyper-parameter grid search. I think a more principled approach...can have valuable impact to the development of this field.\
However, I agree with the concern on empirical evaluation by Reviewer NB1r. While the paper shows improvements over DPS and IIGDM, the quality of DPS and IIGDM samples do look inferior to the previously reported ones. The reproducibility of results remains my main concern of the paper getting accepted.\
Overall, I will recommend accept (somewhere between 6 and 8)."

* Reviewer 3wW3: "After reading other reviewers' response and the authors rebuttal, I still have a borderline opinion.\
  - Pros: the proposed approach is a novel look on the problem, even though I believe the covariance approximation could be greatly improved in the future. It is a more principled approach than most prior work and the lack of hyperparameter tuning is a significant benefit in my opinion having worked with other diffusion solvers.\
  - Cons: The experimental results are on CIFAR-10, which cannot be taken too seriously in this context as all competing methods typically provide results on datasets such as FFHQ, CelebA-HQ, ImageNet etc. with higher resolution images. Another issue is the very significant changes in the manuscript, which somewhat questions the reliability of the results: the source code needed to be fixed, experiments performed from scratch, and...the theoretical results have also been updated.\
I would be hesitant to strongly advise either accept or reject for this paper. ...this paper could be a great starting point for a new line of work efficiently leveraging higher-order information. ...the presented results are not sufficiently convincing, and the work appears somewhat of a work in progress."

I found the remaining concerns on the mentioned weaknesses in Metareview are valid, and that many vital theoretical and empirical results are provided only in the rebuttal stage leads to significant change hence may require further reviews for consistency and reproducibility. I posted the thoughts and recommending reject to reviewers. No one further argued for accept.

**Justification For Why Not Lower Score:**

N/A

---

### Decision · Program_Chairs · 2024-01-16

Reject